# MOMENTUM AND ERROR FEEDBACK FOR CLIPPING WITH FAST RATES AND DIFFERENTIAL PRIVACY

## ABSTRACT

Strong Differential Privacy (DP) and Optimization guarantees are two desirable properties for a method in Federated Learning (FL). However, existing algorithms do not achieve both properties at once: they either have optimal DP guarantees but rely on restrictive assumptions such as bounded gradients/bounded data heterogeneity, or they ensure strong optimization performance but lack DP guarantees. To address this gap in the literature, we propose and analyze a new method called Clip21-SGDM based on a novel combination of clipping, heavy-ball momentum, and Error Feedback. In particular, for non-convex smooth distributed problems with clients having arbitrarily heterogeneous data, we prove that Clip21-SGDM has optimal convergence rate and also optimal (local-)DP neighborhood. Our numerical experiments on non-convex logistic regression and training of neural networks highlight the superiority of Clip21-SGDM over baselines in terms of the optimization performance for a given DP-budget.

## 1 INTRODUCTION

Federated Learning (Konečný et al., 2016; McMahan et al., 2017a) is a modern training paradigm where multiple (possibly heterogeneous) clients aim to jointly train a machine learning model without sacrificing the privacy of their own data. This setup presents several noticeable challenges in terms of algorithm design affecting different aspects of training, including communication efficiency, partial participation of clients, data heterogeneity, security, and privacy (Kairouz et al., 2021; Wang et al., 2021). As a result, numerous optimization methods for Federated Learning (FL) have been introduced in recent years. However, despite extensive research in the field, achieving both strong optimization convergence and robust differential privacy (DP) guarantees (Dwork et al., 2014) simultaneously in an FL algorithm remains challenging due to the conflicting nature of these objectives. Indeed, most of the results in the field of DP are obtained by adding noise (e.g. Gaussian noise) to the method's update (Abadi et al., 2016; Chen et al., 2020) in order to protect the client's data that could be potentially reconstructed from the updates. Unfortunately, this approach results in less accurate updates, which negatively affects the convergence. Moreover, to ensure DP, this mechanism should be applied to the method with bounded updates, which is typically achieved via *gradient clipping* (Pascanu et al., 2013).

Further complicating the issue, naïve distributed Clipped Gradient Descent (Clip-GD) is not guaranteed to converge (Khirirat et al., 2023) when clients have heterogeneous data (even in the absence of any additive DP-noise), which is a common scenario in FL. To address this issue Khirirat et al. (2023) apply the EF21 mechanism – originally developed by Richtárik et al. (2021) for contractive compression operators to improve the standard Error Feedback (Seide et al., 2014) – to Clip-GD, resulting in a method known as Clip21-GD. Khirirat et al. (2023) show that in contrast to Clip-GD, Clip21-GD converges with $\mathcal{O}(1/T)$ rate for smooth non-convex problems with arbitrary heterogeneous data on clients. However, their analysis is limited to the case of full-batched gradients and does not work with DP-noise. This leads us to the natural question:

> *Is it possible to design a method that combines both strong optimization performance*
> *and DP guarantees in a stochastic setting?*

**Our contribution.** In this paper, we provide a positive answer to the above question by introducing a new method, named Clip21-SGDM, which incorporates clipping, error feedback and heavy-ball

momentum (Polyak, 1964) in a novel way. For smooth non-convex distributed optimization problems, we show that Clip21-SGDM (i) converges with optimal $\mathcal{O}(1/T)$ rate when the workers compute full gradients, (ii) converges with optimal $\widetilde{\mathcal{O}}(1/\sqrt{nT})$ high-probability convergence rate when the workers use stochastic gradients with sub-Gaussian noise, and (iii) has optimal local DP-error when DP-noise is added to the clients' updates. We also prove that Clip21-SGD is not guaranteed to converge in the stochastic case, underscoring the need for changes in the algorithm. Our experiments on logistic regression and neural networks highlight the robustness of Clip21-SGDM to the choice of clipping level and indicate Clip21-SGDM's superiority over Clip-SGD and Clip21-SGD in terms of optimization performance for a given DP-budget.

## 1.1 Problem Formulation and Assumptions

We consider the optimization problem of the form

$$\min_{x \in \mathbb{R}^d} \left[ f(x) := \frac{1}{n} \sum_{i=1}^{n} f_i(x) \right] \tag{1}$$

that typically appears in many machine learning applications and is standard for Federated Learning. Here $x$ denotes the parameters of a model, $f_i$ represents the loss associated with the local dataset $\mathcal{D}_i$ of worker $i \in [n]$, and $f$ is an average loss across all workers participating in the training process.

We make two main assumptions on the problem. The first one is smoothness, which is standard for non-convex optimization (Carmon et al., 2020; Danilova et al., 2022). In addition, we also assume that $f(x)$ is uniformly lower bounded since otherwise, problem (1) is intractable.

**Assumption 1.** *We assume that each individual loss function $f_i$ is $L$-smooth, i.e., for any $x, y \in \mathbb{R}^d$ and $i \in [n]$ we have*

$$\|\nabla f_i(x) - \nabla f_i(y)\| \leq L\|x - y\|. \tag{2}$$

*Moreover, we assume that $f^* := \inf_{x \in \mathbb{R}^d} f(x) > -\infty$.*

We also note that our analysis can be easily generalized to the case when $L$ depends on $f_i$.

Next, since computation of the full gradients is expensive in many practical applications, it is natural to consider the case when clients compute stochastic gradients. We make the following assumption on the stochastic noise of these gradients.

**Assumption 2.** *We assume that each worker $i$ has access to a $\sigma$-sub-Gaussian unbiased estimator $\nabla f_i(x, \xi)$ of a local gradient $\nabla f_i(x)$, i.e., for some[1] $\sigma \geq 0$ and any $x \in \mathbb{R}^d$ and $\forall i \in [n]$ we have*

$$\mathbb{E}\left[\nabla f_i(x, \xi)\right] = \nabla f_i(x), \quad \mathbb{E}\left[\exp\left(\|\theta_i^t\|^2/\sigma^2\right)\right] \leq \exp(1), \tag{3}$$

*where $\xi$ denotes the source of the stochasticity and $\theta_i := \nabla f_i(x, \xi) - \nabla f_i(x)$.*

Although this assumption is stronger than bounded variance, it is standard for the high-probability[2] analysis of SGD-type methods with polylogarithmic dependence on the confidence level (Nemirovski et al., 2009; Ghadimi & Lan, 2012). The second part of (3) is equivalent to $\Pr\left(\|\theta_i^t\| \geq b\right) \leq 2\exp\left(-b^2/(2\sigma^2)\right)$ up to a constant factor in $\sigma^2$ (Vershynin, 2018). We also note that it is possible to show high-probability bounds for SGD-type methods with polylogarithmic dependence on the confidence level when the noise has sub-Weibull tails (Madden et al., 2024), i.e., the noise can be even heavier but it affects the polylogarithmic factors.

Finally, we provide two important definitions for this work. The first one is the definition of the clipping operator, which is a non-linear map from $\mathbb{R}^d$ to $\mathbb{R}^d$ parameterized by the clipping threshold/level $\tau > 0$ and defined as

$$\text{clip}_\tau(x) := \begin{cases} \frac{\tau}{\|x\|}x, & \text{if } \|x\| > \tau, \\ x, & \text{if } \|x\| \leq \tau. \end{cases} \tag{4}$$

Next, we will use the following classical definition of $(\varepsilon, \delta)$-Differential Privacy, which introduces plausible deniability into the output of a learning algorithm.

---

[1]For simplicity, we define $0/0 := 0$. Then, (3) with $\sigma = 0$ implies $\nabla f_i(x, \xi) = \nabla f_i(x)$ almost surely.

[2]We elaborate on the reasons why we focus on high-probability analysis in Section 3.2.

**Definition 1** (($\varepsilon, \delta$)-Differential Privacy (Dwork et al., 2014))**.** A randomized method $\mathcal{M} : \mathcal{D} \to \mathcal{R}$ satisfies ($\varepsilon, \delta$)-Differential Privacy (($\varepsilon, \delta$)-DP) if for any adjacent $D, D' \in \mathcal{D}$ (e.g., if $D$ and $D'$ are datasets, then the adjacency means that $D$ and $D'$ differ in 1 sample) and for any $S \subseteq \mathcal{R}$

$$\Pr\left(\mathcal{M}(D) \in S\right) \leq e^{\varepsilon} \Pr\left(\mathcal{M}(D') \in S\right) + \delta. \tag{5}$$

In this definition, the smaller $\varepsilon, \delta$ are, the more private the method is. Intuitively, if inequality (5) holds with small values of $\varepsilon$ and $\delta$, it becomes difficult to infer the specific data point that differs between two similar datasets based solely on the output of $\mathcal{M}$.

## 1.2 RELATED WORK

**Differential Privacy.** The most common approach to obtaining DP guarantees is to clip each client's update, i.e., by bounding their $\ell_2$ norm, and adding a calibrated amount of Gaussian noise to each update or the average. This is typically sufficient to obscure the influence of any single client (McMahan et al., 2017b). Commonly, two scenarios of the DP model are considered: *the central model* and *the local model*. In the first setting, central privacy, a trusted server collects updates and adds noise only before updating the server-side model. This ensures that client data remains private from external parties. In the second setting, local privacy, client data is protected even from the server by clipping and adding noise to updates locally before sending them to the server, ensuring privacy from both the server and other clients (Kasiviswanathan et al., 2011; Allouah et al., 2024). The local privacy setting offers stronger privacy against untrusted servers but results in poorer learning performance due to the need for more noise to obscure individual updates (Chan et al., 2012; Duchi et al., 2018). This can be improved by using a secure shuffler (Erlingsson et al., 2019; Balle et al., 2019), which permutes updates, or a secure aggregator (Bonawitz et al., 2017), which sums updates before sending them to the server. These methods anonymize updates and enhance privacy while maintaining reasonable learning performance, even without a fully trusted server. Finally, (Chaudhuri et al., 2022; Hegazy et al., 2024) show that when DP is required, one can also achieve compression of updates for free.

In this work, we adopt the local DP model by injecting Gaussian noise into each client's update. However, the average noise can also be viewed as noise added to the average update. Therefore, Clip21-SGDM is compatible with all the aforementioned techniques and can also be applied to the central DP model with a smaller amount of noise.

**Distributed methods with clipping.** In the single-node regime, Clip-SGD has been analyzed under various assumptions by many authors (Zhang et al., 2020b;c;a; Gorbunov et al., 2020a; Cutkosky & Mehta, 2021; Sadiev et al., 2023; Liu et al., 2023). Of course, these results can be generalized to the multi-node case if clipping is applied to the aggregated (e.g. averaged) vector, although mini-batching requires a refined analysis when the noise is heavy-tailed(Kornilov et al., 2024). However, to get DP, clipping has to be applied to the vectors communicated by clients to the server. In this regime, Clip-SGD is not guaranteed to converge even without any stochastic noise in the gradients (Chen et al., 2020; Khirirat et al., 2023). There exist several approaches to bypass this limitation that can be split into two lines of work. The first one relies on explicit or implicit assumptions about bounded heterogeneity. More precisely, Liu et al. (2022) analyze a version of Local-SGD/FedAvg (Mangasarian, 1995; McMahan et al., 2017a) with gradient clipping for homogeneous data case assuming that the stochastic gradients have symmetric distribution around their mean and Wei et al. (2020) consider Local-SGD with clipping of the models and analyze its convergence under bounded heterogeneity assumption. Moreover, the boundedness of the stochastic gradient is another assumption used in the literature but it implies the boundedness of gradients' heterogeneity of clients as well. This assumption is used in numerous works, including: i) Zhang et al. (2022) in the analysis of a version of FedAvg with clipping of model difference (also empirically studied by Geyer et al. (2017)), ii) Noble et al. (2022) who propose and analyze a version of SCAFFOLD (Karimireddy et al., 2020) with gradient clipping (DP-SCAFFOLD), iii) Li & Chi (2023) who propose and analyze a version of BEER (Li et al., 2021) with gradient clipping (PORTER) under bounded gradient and/or bounded data heterogeneity assumption, and iv) Allouah et al. (2024) who study a version of Gossip-SGD (Nedic & Ozdaglar, 2009) with gradient clipping (DECOR). Although most of the mentioned works have rigorous DP guarantees, the corresponding methods are not guaranteed to converge for arbitrary heterogeneous problems.

The second line of work focuses on the clipping of shifted (stochastic) gradient. In particular, Khirirat et al. (2023) proposed and analyzed Clip21-GD, which is based on the application of EF21 (Richtárik et al., 2021) to the clipping operator, and Gorbunov et al. (2024) develop and analyze methods that apply clipping to the difference of stochastic gradients and learnable shift – an idea that was initially proposed by Mishchenko et al. (2019) to handle data heterogeneity in the Distributed Learning with unbiased communication compression. However, the analysis from (Khirirat et al., 2023) is limited to the noiseless regime, i.e., full-batched gradients are computed on workers, and both of the mentioned works do not provide[3] DP guarantees. We also note that clipping of gradient differences is helpful in tolerating Byzantine attacks in the partial participation regime (Malinovsky et al., 2023).

**Error Feedback.** Error Feedback (EF) (Seide et al., 2014) is a popular technique for incorporating communication compression into Distributed/Federated Learning. However, for non-convex smooth problems, the existing analysis of EF is provided either for the single-node case or relies on restrictive assumptions such as boundedness of the gradient/compression error or boundedness of the data heterogeneity (gradient dissimilarity) (Stich et al., 2018; Stich & Karimireddy, 2019; Karimireddy et al., 2019; Koloskova et al., 2019; Beznosikov et al., 2023; Tang et al., 2019; Xie et al., 2020; Sahu et al., 2021). Moreover, the convergence bounds for EF also depend on the data heterogeneity, which is not an artifact of the analysis as illustrated in the experiments on strongly convex problems Gorbunov et al. (2020b). Richtárik et al. (2021) address this limitation and propose a new version of Error Feedback called EF21. However, the existing analysis of EF21-SGD requires the usage of large batch sizes to achieve any predefined accuracy (Fatkhullin et al., 2021). It turns out that the large batch size requirement is unavoidable for EF21-SGD to converge, but this issue can be fixed using momentum (Fatkhullin et al., 2024). Momentum is also helpful in the decentralized extensions of Error Feedback (Yau & Wai, 2022; Huang et al., 2023; Islamov et al., 2024).

## 2   NON-CONVERGENCE OF Clip-SGD AND Clip21-SGD

We start with a discussion of the key limitations of Clip-SGD (Algortihm 1) and Clip21-SGD (Algorithm 2) – their potential non-convergence.

| **Algorithm 1** Clip-SGD (Abadi et al., 2016) | **Algorithm 2** Clip21-SGD (Khirirat et al., 2023) |
|---|---|
| **Input:** $x^0 \in \mathbb{R}^d$, stepsize $\gamma > 0$, clipping parameter $\tau > 0$ | **Input:** $x^0, g^0 \in \mathbb{R}^d$, stepsize $\gamma > 0$, clipping parameter $\tau > 0$ |
| | 1: Initialize $g_i^0 = g^0$ for all $i \in [n]$ |
| 1: **for** $t = 0, \ldots, T - 1$ **do** | 2: **for** $t = 0, \ldots, T - 1$ **do** |
| | 3:     $x^{t+1} = x^t - \gamma g^t$ |
| 2:     **for** $i = 1, \ldots, n$ in parallel **do** | 4:     **for** $i = 1, \ldots, n$ in parallel **do** |
| | 5:         $c_i^{t+1} = \text{clip}_\tau(\nabla f_i(x^{t+1,\xi_i^{t+1}}) - g_i^t)$ |
| 3:         $g_i^t = \text{clip}_\tau(\nabla f_i(x^t, \xi_i^t))$ | 6:         $g_i^{t+1} = g_i^t + c_i^{t+1}$ |
| 4:     **end for** | 7:     **end for** |
| 5:     $g^t = \frac{1}{n} \sum_{i=1}^n g_i^t$ | 8:     $g^{t+1} = g^t + \frac{1}{n} \sum_{i=1}^n c_i^{t+1}$ |
| 6:     $x^{t+1} = x^t - \gamma g^t$ | |
| 7: **end for** | 9: **end for** |

We start by restating the example from (Chen et al., 2020) illustrating the potential non-convergence of Clip-SGD even when full gradients are computed on clients (Clip-GD).

**Example 1** (Non-Convergence of Clip-GD (Chen et al., 2020))**.** Let $n = 2$, $d = 1$, and $f_1(x) = \frac{1}{2}(x - 3)^2$, $f_2(x) = \frac{1}{2}(x + 3)^2$ in problem (1) having a unique solution $x^* = 0$. Consider Clip-GD with $\tau = 1$ applied to this problem. If for some $t_0$ we have $x^{t_0} \in [-2, 2]$ in Clip-GD, then $g^t = 0$ and $x^t = x^{t_0}$ for any $t \geq t_0$, which can be seen via direct calculations. In particular, for any $x^0 \in [-2, 2]$, the method does not move away from $x^0$.

---

[3]The proof of the DP guarantee by Khirirat et al. (2023) relies on the condition for some $C > 1$ and $\nu, \sigma_\omega \geq 0$ that implies $\min\{\nu^2, \sigma_\omega^2\} \geq C \max\{\nu^2, \sigma_\omega^2\}$. The latter one holds if and only if $\nu = \sigma_\omega = 0$, which means that no noise is added to the method since $\sigma_\omega^2$ is the variance of DP-noise.

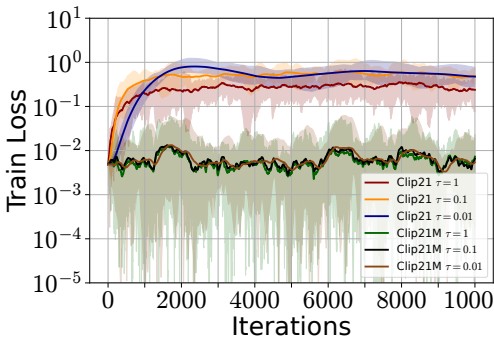 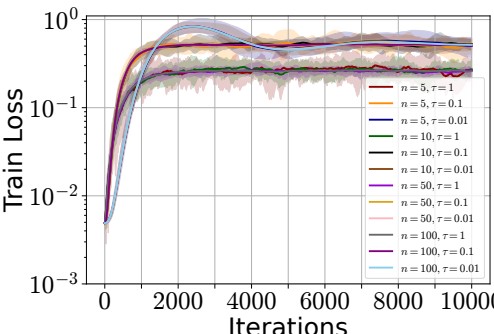

Figure 1: **Left:** behavior of stochastic Clip21-SGD and Clip21-SGDM without DP noise (see Algorithm 3) initialized at $x^0 = (0, -0.07)^\top$, with stepsize $\gamma = 1/\sqrt{T}$ where $T = 10^4$, i.e., close to the solution and small stepsize. We observe that Clip21-SGD escapes the good neighborhood of the solution for the problem from Theorem 1 with $n = 1, L = 2, \sigma = 5$, and varying $\tau \in \{1, 0.1, 0.01\}$. In contrast, Clip21-SGDM remains stable around the solution. **Right:** convergence of Clip21-SGD does not improve with the increase of $n$ for the same problem.

To address the non-convergence of Clip-GD, Khirirat et al. (2023) propose Clip21-GD that applies the clipping operator to the difference between $\nabla f_i(x^{t+1})$ and the shift $g_i^t$, which is designed to approximate $\nabla f_i(x^t)$. In the deterministic case, this strategy ensures that after a certain number of steps, clipping turns off on all clients since $\|\nabla f_i(x^{t+1}) - g_i^t\|$ becomes smaller than $\tau$ for all $i \in [n]$ eventually. However, when workers compute stochastic gradients instead of the full gradients, Clip21-SGD can be non-convergent as well. To illustrate this, we consider the ideal version of Clip21-SGD with stochastic gradients, i.e., instead of $g_i^t$, we use $\nabla f_i(x^{t+1})$ as a shift:

$$x^{t+1} = x^t - \gamma g^t, \quad g^t = \frac{1}{n} \sum_{i=1}^{n} g_i^t, \tag{6}$$

$$g_i^{t+1} = \nabla f_i(x^{t+1}) + \text{clip}_\tau(\nabla f_i(x^{t+1}, \xi_i^{t+1}) - \nabla f_i(x^{t+1})) \tag{7}$$

The next theorem shows that even this (ideal) version of stochastic Clip21-SGD fails to converge even for a simple quadratic problem with sub-Gaussian noise.

**Theorem 1.** *Let $L, \sigma > 0, 0 < \gamma \leq 1/L, n = 1$. There exists a convex, L-smooth problem, clipping parameter $\tau < {}^{3\sigma\sqrt{3}}/10$, and an unbiased stochastic gradient satisfying Assumption 2 such that the method (6) is run with a stepsize $\gamma$ and clipping parameter $\tau$, then for all $x^0 \in \{(0, x_{(2)}^0) \in \mathbb{R}^2 \mid x_{(2)}^0 < 0\}$ we have*

$$\mathbb{E}\left[\|\nabla f(x^T)\|^2\right] \geq \frac{1}{2} \min\left\{\|\nabla f(x^0)\|^2, \frac{\tau^2}{45}\right\}.$$

*Moreover, fix $0 < \varepsilon < {}^{L}/\sqrt{2}$ and $x^0 = (0, -1)^\top$. Let the sub-Gaussian variance of stochastic gradients is bounded by $\sigma^2/B$ where $B$ is a batch size. If $B < {}^{27\sigma^2}/(60\varepsilon^2)$ and $\tau \geq \varepsilon/(3\sqrt{10})$, then we have $\mathbb{E}\left[\|\nabla f(x^T)\|^2\right] > \varepsilon^2$ for all $T > 0$.*

We also illustrate the above result with simple numerical experiments reported in Figure 1. The left figure shows that Clip21-SGD diverges from the initial function sub-optimality level while the right one demonstrates non-improvement with the number of workers $n$ — one of the desired properties of algorithms for FL.

## 3 Clip21-SGDM: New Method and Theoretical Results

This section introduces Clip21-SGDM (Algorithm 3), a novel distributed method with clipping that can be viewed as an enhanced version of Clip21-SGD, integrating momentum and DP-noise. That is, to control the noise coming from the stochastic gradients, we introduce momentum buffers $\{v_i^t\}_{i \in [n]}$ on the clients and clip $\{v_i^{t+1} - g_i^t\}_{i \in [n]}$ in contrast to the stochastic version of Clip21-SGD that

---

**Algorithm 3** Clip21-SGDM

---

1: **Input:** $x^0, g^0, v^0 \in \mathbb{R}^d$ (by default $g^0 = v^0 = 0$), momentum parameter $\beta \in (0, 1]$, stepsize $\gamma > 0$, clipping parameter $\tau > 0$, DP-variance parameter $\sigma_\omega^2 \geq 0$
2: Set $g_i^0 = g^0$ and $v_i^0 = v^0$ for all $i \in [n]$
3: **for** $t = 0, \dots, T - 1$ **do**
4:     $x^{t+1} = x^t - \gamma g^t$
5:     **for** $i = 1, \dots, n$ **do**
6:         $v_i^{t+1} = (1 - \beta)v_i^t + \beta \nabla f_i(x^{t+1}, \xi_i^{t+1})$
7:         $\omega_i^{t+1} \sim \mathcal{N}(0, \sigma_\omega^2 \mathbf{I})$                    only for DP version
8:         $c_i^{t+1} = \mathrm{clip}_\tau(v_i^{t+1} - g_i^t) + \omega_i^{t+1}$
9:         $g_i^{t+1} = g_i^t + c_i^{t+1} - \omega_i^{t+1} = g_i^t + \mathrm{clip}_\tau(v_i^{t+1} - g_i^t)$
10:    **end for**
11:    $g^{t+1} = g^t + \frac{1}{n} \sum_{i=1}^n c_i^{t+1}$
12: **end for**

---

applies clipping to potentially noisier vectors $\{\nabla f_i(x^{t+1}, \xi_i^{t+1}) - g_i^t\}_{i \in [n]}$. Moreover, similarly to Clip21-SGD – which can be seen as EF21 (Richtárik et al., 2021) where the compression operator is replaced by clipping – Clip21-SGDM can also be interpreted as EF21M (Fatkhullin et al., 2024) with the same replacement. However, both EF21 and EF21M rely on the contractiveness property of the compression operator $\mathcal{C}(x)$, i.e., the (randomized) mapping $\mathcal{C} : \mathbb{R}^d \to \mathbb{R}^d$ should satisfy

$$\mathbb{E}\left[\|\mathcal{C}(x) - x\|^2\right] \leq (1 - \nu)\|x\|^2 \quad \text{for some} \quad \nu \in (0, 1], \tag{8}$$

where the expectation is w.r.t. the randomness of $\mathcal{C}$. As shown and discussed by Khirirat et al. (2023), clipping satisfies a condition that resembles (8) namely

$$\|\mathrm{clip}_\tau(x) - x\|^2 \leq \begin{cases} 0, & \text{if } \|x\| \leq \tau, \\ \left(1 - \frac{\tau}{\|x\|}\right)^2 \|x\|^2, & \text{if } \|x\| > \tau, \end{cases} \tag{9}$$

but there is a significant difference: if $\|x\| > \tau$, the contraction factor is dependent of $x$ and can be arbitrarily close to 1. To circumvent this issue, Khirirat et al. (2023) prove via induction that for all iterates of Clip21-SGD, the vectors $\nabla f_i(x^{t+1}) - g_i^t$ have norms bounded by some constant depending on the starting point. We show that a similar statement holds for Clip21-SGDM when the clients compute full-batched gradients and no DP-noise is added, and we start our analysis with this important special case. We also present the results in the stochastic case with and without DP noise.

### 3.1 ANALYSIS IN THE DETERMINISTIC CASE

The next result derives a convergence rate for Clip21-SGDM when $\nabla f_i(x^{t+1}, \xi_i^{t+1}) \equiv \nabla f_i(x^t)$ almost surely, i.e., Assumption 2 holds with $\sigma = 0$.

**Theorem 2** (Simplified). *Let Assumptions 1 and 2 with $\sigma = 0$ hold. Let $B := \max_i \|\nabla f_i(x^0)\| > 3\tau$ and $\Delta \geq f(x^0) - f^*$. Then there exists a stepsize $\gamma \leq 1/12L$ and momentum parameter $\beta = 4L\gamma$ such that the iterates of Clip21-SGDM (Algorithm 3) converge with the rate*

$$\frac{1}{T} \sum_{t=0}^{T-1} \|\nabla f(x^t)\|^2 \leq \mathcal{O}\left(\frac{L\Delta}{T}\right). \tag{10}$$

*Moreover, after at most $2B/\tau$ iterations, the clipping will eventually be turned off for all workers.*

*Proof sketch.* The proof of Theorem 2 (and all following ones) relies on the same Lyapunov function that is used by Fatkhullin et al. (2024) in the analysis of EF21M:

$$\Phi^t := f(x^t) - f^* + \frac{\gamma}{\eta} \frac{1}{n} \sum_{i=1}^n \|g_i^t - v_i^t\|^2 + \frac{4\gamma\beta}{\eta^2} \frac{1}{n} \sum_{i=1}^n \|v_i^t - \nabla f_i(x^t)\|^2 + \frac{\gamma}{\beta} \|v^t - \nabla f(x^t)\|^2. \tag{11}$$

In the definition of $\Phi^t$, the only parameter that was not introduced earlier in the paper is $\eta$, and it hides the main technical difficulty of the proof. That is, by induction we prove that $\|v_i^{t+1} - $

$g_i^t\| \leq \tau/\eta$ for some $\eta$ defined in the proof. This bound is essential in deriving a descent of each term in the Lyapunov function. In view of (9) and (8), this allows us to consider clipping as a contractive compression operator for vectors $v_i^{t+1} - g_i^t$ generated by the method, and also this allows us to use the same Lyapunov function as in the analysis of EF21M. We defer the detailed proof to Appendix D. □

The above result establishes a $\mathcal{O}(1/T)$ convergence rate that is optimal for non-convex smooth first-order optimization (Carmon et al., 2020; 2021). This result matches the one obtained by Khirirat et al. (2023), and, in particular, similarly to Clip21-SGD, Clip21-SGDM turns off clipping on each client after a finite number of steps $t$ satisfying $\|v_i^{t+1} - g_i^t\| \leq \tau$. We also emphasize that Theorem 2 holds without bounded heterogeneity/gradient assumption. In contrast, even with bounded heterogeneity/gradient assumption, many existing convergence results in the non-convex case (Liu et al., 2022; Zhang et al., 2022; Li & Chi, 2023; Allouah et al., 2024) do not recover the $\mathcal{O}(1/T)$ rate in the noiseless regime.

### 3.2 ANALYSIS IN THE STOCHASTIC CASE WITHOUT DP-NOISE

Next, we turn to the stochastic setting where each worker has access to local gradient estimators satisfying Assumption 2. For simplicity, we first consider the case when no DP noise is added.

**Theorem 3.** *Let Assumptions 1 and 2 hold and $\alpha \in (0,1)$. Let $\widetilde{B} := \max_i \|\nabla f_i(x^0)\| > 3\tau$ and $\Delta \geq \Phi^0$. Then there exists a stepsize $\gamma$ and momentum parameter $\beta$ such that the iterates of* Clip21-SGDM *(Algorithm 3) satisfy with probability at least $1 - \alpha$*

$$\frac{1}{T}\sum_{t=0}^{T-1} \|\nabla f(x^t)\|^2 \leq \widetilde{\mathcal{O}}\left( \frac{L\Delta}{T} + \frac{\sigma(\sqrt{L\Delta} + \widetilde{B} + \sigma)}{\sqrt{Tn}} \right), \tag{12}$$

*where $\widetilde{\mathcal{O}}$ hides constant and logarithmic factors, and higher order terms that decrease in $T$.*

*Proof sketch.* The core of the proof is similar to the one of Theorem 2. However, in contrast to the deterministic case, the vectors $v_i^{t+1} - g_i^t$ are stochastic, meaning that under Assumption 2, they can have arbitrarily large norms. Therefore, we focus on the high-probability analysis and prove by induction that the vectors $v_i^{t+1} - g_i^t$ are bounded *with high probability*, meaning that clipping can be seen as a contractive compressor with high probability for the vectors $v_i^{t+1} - g_i^t$ generated by the method. The proof is also based on a refined estimation of sums of martingale difference sequences; see the details in Appendix G. □

This result demonstrates that Clip21-SGDM achieves an optimal $\mathcal{O}(1/\sqrt{nT})$ (Arjevani et al., 2023) rate in the stochastic setting. In contrast to the previous works establishing similar rates (Liu et al., 2022; Noble et al., 2022; Allouah et al., 2024), our result does not rely on the boundedness of the gradients or data heterogeneity. Moreover, when $\sigma = 0$ (no stochastic noise), the rate from (12) becomes $\mathcal{O}(1/T)$, recovering the one given by Theorem 2.

### 3.3 ANALYSIS IN THE STOCHASTIC CASE WITH DP-NOISE

Finally, we provide the convergence result for Clip21-SGDM with DP-noise.

**Theorem 4.** *Let Assumptions 1 and 2 hold and $\alpha \in (0,1)$. Let $\Delta \geq \Phi^0$. Then there exists a stepsize $\gamma$ and momentum parameter $\beta$ such that the iterates of* Clip21-SGDM *(Algorithm 3) with the DP-noise variance $\sigma_\omega^2$ with probability at least $1 - \alpha$ satisfy*

$$\frac{1}{T}\sum_{t=0}^{T-1} \|\nabla f(x^t)\|^2 \leq \widetilde{\mathcal{O}}\left( \frac{L\Delta\sqrt{d}\sigma_\omega}{\sqrt{Tn}\tau} + \frac{(L\Delta)^{1/6}\sigma^{5/3}}{T^{1/6}n^{5/6}} + \frac{(L\Delta)^{4/9}\sigma^{5/9}d^{5/18}\sigma_\omega^{5/9}}{T^{4/9}n^{5/9}} \right), \tag{13}$$

*where $\widetilde{\mathcal{O}}$ hides constant and logarithmic factors, and higher order terms decreasing in $T$.*

In the special case of local Differential Privacy, the noise level has to be chosen in a specific way. In this setting, we obtain the following privacy-utility trade-off.

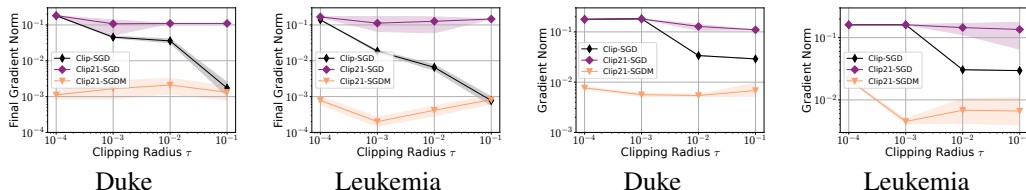

Figure 2: Comparison of tuned Clip-SGD, Clip21-SGD, and Clip21-SGDM on logistic regression with non-convex regularization for various clipping radii $\tau$ with mini-batch (**two left**) and Gaussian-added (**two right**) stochastic gradients. The final gradient norm is averaged over the last 100 iterations.

**Corollary 1.** Let Assumptions 1 and 2 hold and $\alpha \in (0,1)$. Let $\Delta \geq \Phi^0$ and $\sigma_\omega$ be chosen as $\sigma_\omega = \Theta\left(\frac{\tau}{\varepsilon}\sqrt{T \log \frac{1}{\delta}}\right)$. Then there exists a stepsize $\gamma$ and momentum parameter $\beta$ such that the iterates of Clip21-SGDM (Algorithm 3) with probability at least $1 - \alpha$ satisfy local $(\varepsilon, \delta)$-DP and

$$\frac{1}{T}\sum_{t=0}^{T-1} \|\nabla f(x^t)\|^2 \leq \widetilde{\mathcal{O}}\left(\frac{L\Delta\sqrt{d}}{\sqrt{n}\varepsilon}\right), \tag{14}$$

where $\widetilde{\mathcal{O}}$ hides constant and logarithmic factors, and terms decreasing in $T$.

To obtain local $(\varepsilon, \delta)$-DP guarantees we follow Theorem 1 in (Abadi et al., 2016). This privacy-utility trade-off matches the known lower bound for locally private algorithms (Duchi et al., 2018). Overall, Theorems 2 and 3 and Corollary 1 indicate that Clip21-SGDM achieves optimal convergence rates in both deterministic and stochastic regime, and also has an optimal privacy-utility trade-off. These results are derived without assuming the boundedness of the gradients/data heterogeneity.

## 4 EXPERIMENTS

In this section, we provide an empirical evaluation of the proposed algorithm against baselines such as Clip21-SGD (Khirirat et al., 2023) and Clip-SGD. The learning rate and momentum (for Clip21-SGDM) are tuned in all experiments. We refer to Appendix H for the detailed description of tuning.

### 4.1 STOCHASTIC SETTING

First, we test the convergence of Clip-SGD, Clip21-SGD, and the proposed Clip21-SGDM algorithms with stochastic gradients for various clipping radii $\tau$ on several workloads. These results demonstrate the significance of using the momentum technique to achieve better performance.

### 4.1.1 NON-CONVEX LOGISTIC REGRESSION

We demonstrate the performance of all algorithms without adding noise for privacy but with stochastic gradients. We consider two cases: adding Gaussian noise to full local gradient $\nabla f_i(x)$ and mini-batch stochastic gradient. We conduct experiments on logistic regression with non-convex regularization, namely, $f_i(x) = \frac{1}{m}\sum_{j=1}^m \log(1 + \exp(-b_{ij}a_{ij}^\top x)) + \lambda \sum_{l=1}^d \frac{x_l^2}{1+x_l^2}$ which is a typical problem considered in previous works (Khirirat et al., 2023; Li & Chi, 2023). We use the Duke and Leukemia LibSVM (Chang & Lin, 2011) datasets.

We plot the gradient norm averaged across the last 100 iterations and 3 different runs in Figure 2. The results demonstrate the resilience of Clip21-SGDM to the choice of the clipping radius $\tau$: it achieves a smaller or similar gradient norm compared to two other algorithms over all values of $\tau$. This is especially visible when the clipping radius $\tau$ is small. These experimental findings align with the theoretical results presented in this work. Besides, the convergence plots are presented in Figure 7. The results demonstrate faster convergence for Clip21-SGDM than that of Clip21-SGD and Clip-SGD.

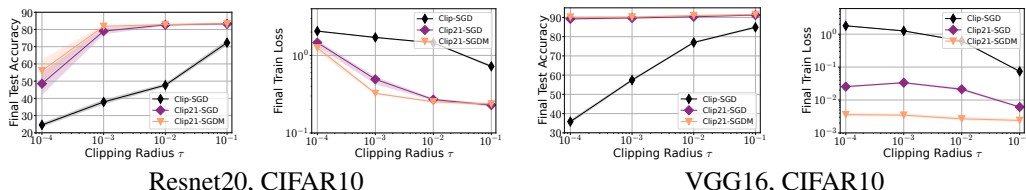

Figure 3: Comparison of tuned Clip-SGD, Clip21-SGD, and Clip21-SGDM on training Resnet20 (**two left**) and VGG16 (**two right**) models on CIFAR10 dataset where the clipping is applied globally.

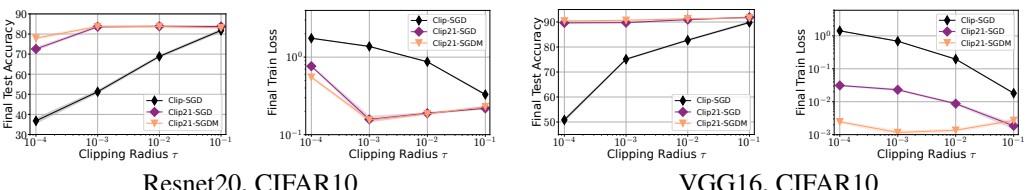

Figure 4: Comparison of tuned Clip-SGD, Clip21-SGD, and Clip21-SGDM on training Resnet20 (**two left**) and VGG16 (**two right**) models on CIFAR10 dataset where the clipping is applied layer-wise.

### 4.1.2 TRAINING RESNET20 AND VGG16

Next, we conduct experiments in training Resnet20 (He et al., 2016) and VGG16 (Simonyan & Zisserman, 2014) models on CIFAR10 dataset (Krizhevsky et al., 2009)[4]. The results are averaged across 3 different random seeds and shown in Figure 3 (the clipping operator is applied on all weights simultaneously) and Figure 4 (the clipping operator is applied layer-wise). We plot the test accuracy and train loss at the last point of the training. The results show that the performance of Clip-SGD consistently deteriorates as the clipping radius $\tau$ decreases, while Clip21-SGD and Clip21-SGDM are more stable to the changes of $\tau$. Moreover, Clip21-SGDM outperforms Clip21-SGD for small values of $\tau$ reaching smaller train loss and larger test accuracy that supports the theoretical claims of this paper. For the convergence curves we refer to Figures 8 to 11.

### 4.2 ADDING GAUSSIAN NOISE FOR DP

In the second set of experiments, we test the performance of algorithms with additive Gaussian noise to preserve privacy. Since DP noise variance $\sigma_\omega$ typically scales with the clipping radius $\tau$ (e.g., see Corollary 1), we conduct the following set of experiments: we fix a noise-clipping ratio from $\{0.1, 1.0, 10\}$ for logistic regression and from $\{0.1, 0.3, 1.0, 3.0, 10.0\}$ for neural networks, and find such $\tau$ that gives the lowest final gradient norm, train loss, or test accuracy depending on the considered workload. The high values of the noise-clipping ratio correspond to stronger DP guarantees, while low values stand for weaker DP guarantees.

### 4.2.1 NON-CONVEX LOGISTIC REGRESSION

We provide the convergence results for non-convex logistic regression in Figure 5 where the gradient norm is averaged over the last 100 iterations and 5 random seeds. We demonstrate that Clip21-SGDM can achieve a smaller gradient norm for all values of the noise-clipping ratio than Clip-SGD. Besides, the performance of Clip21-SGD does not improve even if the noise-clipping ratio is small, demonstrating the importance of the use of momentum.

### 4.2.2 TRAINING NEURAL NETWORKS WITH DP NOISE

Next, we conduct experiments on training CNN and MLP models on MNIST dataset (Deng, 2012) varying the noise-clipping ratio. We highlight that it is a standard experiment setting considered in the literature on differential privacy (Papernot et al., 2020; Li & Chi, 2023; Allouah et al., 2024). The performance results are reported in Figure 6. We observe that no algorithm outperforms others

---

[4]We use the code base from (Horváth & Richtárik, 2020) with small modifications.

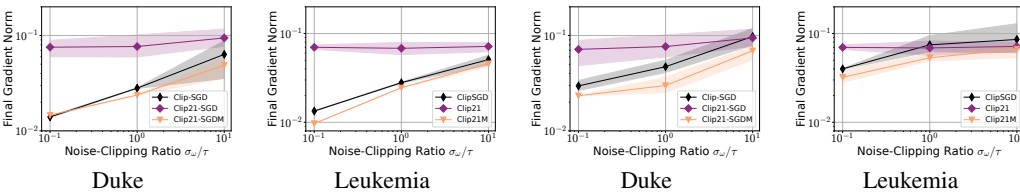

Figure 5: Comparison of tuned Clip-SGD, Clip21-SGD, Clip21-SGDM with mini-batch (**two left**) and Gaussian-added (**two right**) stochastic gradients and with additional DP-noise with variance $\sigma_\omega$ and varying noise-clipping ratio $\sigma_\omega/\tau$ on non-convex logistic regression with non-convex regularization. The final gradient norm is averaged over the last 100 iterations.

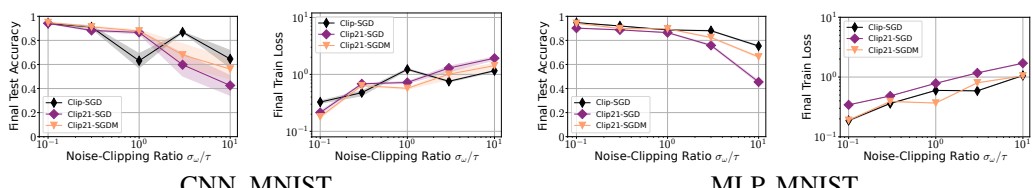

Figure 6: Comparison of tuned Clip-SGD, Clip21-SGD, and Clip21-SGDM on training CNN (**two left**) and MLP (**two right**) models on MNIST dataset varying the noise-clipping ration where the clipping is applied globally.

across all values of the noise-clipping ratio in terms of the train loss. However, Clip-SGD typically attains smaller train loss than Clip21-SGDM for a large value of the noise-clipping ratio while Clip21-SGDM achieves smaller train loss Clip-SGD when that ratio is small.

## 5 CONCLUSION AND FUTURE WORK

In this work, we introduced a new method called Clip21-SGDM and proved that it achieves an optimal convergence rate and optimal privacy-utility trade-off without assuming boundedness of the gradients or boundedness of the data heterogeneity. Notably, several interesting directions remain unexplored. The first one is related to the generalization of the derived results to the case when stochastic gradients have heavy-tailed noise. Next, it would be interesting to study AdaGrad/Adam-type (Streeter & McMahan, 2010; Duchi et al., 2011; Kingma & Ba, 2014) versions of Clip21-SGDM due to their practical superiority over SGD in solving Deep Learning problems. Finally, it is important to extend the current analysis of Clip21-SGDM to the case when generalized smoothness is satisfied (Zhang et al., 2020b).

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

CONTENTS

## A    NOTATION

For shortness, in all proofs, we use the following notation

$$\delta^t := f(x^t) - f^*, \quad \widetilde{V}^t := \frac{1}{n} \sum_{i=1}^n \|g_i^t - v_i^t\|^2,$$

$$\widetilde{P}^t := \frac{1}{n} \sum_{i=1}^n \|v_i^t - \nabla f_i(x^t)\|^2, \quad P^t := \|v^t - \nabla f(x^t)\|^2,$$

$$R^t := \|x^{t+1} - x^t\|^2.$$

We additionally denote $\eta_i^t := \frac{\tau}{\|v_i^t - g_i^{t-1}\|}$ and $\eta := \frac{\tau}{B}$ where $B$ is defined in each section (it is different in deterministic and stochastic settings). Besides, we define $\mathcal{I}_t := \{i \in [n] \mid \|v_i^t - g_i^{t-1}\| > \tau\}$.

We denote $\theta_i^t := \nabla f_i(x^t, \xi_i^t) - \nabla f_i(x^t)$. From Assumption 2, we have that $\theta_i^t$ is zero-centered $\sigma$-sub-Gaussian random vector conditioned at $x^t$, namely

$$\mathbb{E}\left[\theta_i^t \mid x^t\right] = 0, \quad \Pr(\|\theta_i^t\| > b) \leq 2\exp\left(-\frac{b^2}{2\sigma^2}\right) \quad \forall b > 0. \tag{15}$$

Moreover, we define an average of $\theta_i^t$ as $\theta^t := \frac{1}{n} \sum_{i=1}^n \theta_i^t$.

## B    USEFUL LEMMAS

**Lemma 1** (Lemma C.3 in (Gorbunov et al., 2019)). Let $\{\xi_k\}_{k=1}^N$ be the sequence of random vectors with values in $\mathbb{R}^n$ such that

$$\mathbb{E}\left[\xi_k \mid \xi_{k-1}, \ldots, \xi_1\right] = 0 \text{ almost surely}, \forall k \in \{1, \ldots, N\},$$

and set $S_N := \sum_{k=1}^N \xi_k$. Assume that the sequence $\{\xi_k\}_{k=1}^N$ are sub-Gaussian, i.e.

$$\mathbb{E}\left[\exp\left(\|\xi_k\|^2/\sigma_k^2 \mid \xi_{k-1}, \ldots, \xi_1\right)\right] \leq \exp(1) \text{ almost surely}, \forall k \in \{1, \ldots, N\},$$

where $\sigma_2, \ldots, \sigma_N$ are some positive numbers. Then for all $\gamma \geq 0$

$$\Pr\left(\|S_N\| \geq (\sqrt{2} + 2\gamma)\sqrt{\sum_{k=1}^N \sigma_k^2}\right) \leq \exp(-\gamma^2/3). \tag{16}$$

**Lemma 2** (Modification of Lemma 1 in (Li et al., 2021)). Let $\delta^t = f(x^t) - f^*$, $x^{t+1} = x^t - \gamma g^t$, and the stepsize $\gamma \leq \frac{1}{2L}$. Then

$$\delta^{t+1} \leq \delta^t - \frac{\gamma}{2}\|\nabla f(x^t)\|^2 - \frac{1}{4\gamma}\|x^{t+1} - x^t\|^2 + \gamma\frac{1}{n}\sum_{i=1}^n \|g_i^t - v_i^t\|^2 + \gamma\|v^t - \nabla f(x^t)\|^2. \tag{17}$$

**Lemma 3** (Lemma 4.1 in (Khirirat et al., 2023)). The clipping operator satisfies for any $x \in \mathbb{R}^d$

$$\|\operatorname{clip}_\tau(x) - x\| \leq \max\{\|x\| - \tau, 0\}. \tag{18}$$

**Lemma 4** (Property of smooth functions). Let $\phi: \mathbb{R}^d \to \mathbb{R}$ be $L$-smooth and lower bounded by $\phi^* \in \mathbb{R}$, i.e. $\phi(x) \geq \phi^*$ for any $x \in \mathbb{R}^d$. Then we have

$$\|\nabla\phi(x)\|^2 \leq 2L(\phi(x) - \phi^*). \tag{19}$$

*Proof.* It is a standard property of smooth functions. We refer to Theorem 4.23 of (Orabona, 2019).
$\square$

## C   PROOF OF THEOREM 1

*Proof.* **The case** $n = 1$. Let us consider the problem $f(x) = \frac{L}{2}\|x\|^2$. Let vectors $\{z_j\}_{j=1}^3$ are defined as

$$z_1 = \begin{pmatrix} 3 \\ 0 \end{pmatrix} \sqrt{\frac{3\sigma^2}{100}}, \quad z_2 = \begin{pmatrix} 0 \\ 4 \end{pmatrix} \sqrt{\frac{3\sigma^2}{100}}, \quad z_1 = \begin{pmatrix} -3 \\ -4 \end{pmatrix} \sqrt{\frac{3\sigma^2}{100}}.$$

Note that we have

$$\|z_1\|^2 = \frac{27\sigma^2}{100}, \quad \|z_2\|^2 = \frac{24\sigma^2}{50}, \quad \|z_3\|^2 = \frac{3\sigma^2}{4},$$

meaning that $\tau < \|z_i\|$ for all $i \in [3]$. We define the stochastic gradient as $\nabla f(x^t, \xi^t) = \nabla f(x^t) + \xi^t = Lx^t + \xi^t$ where $\xi^t$ is picked uniformly at random from $\{z_1, z_2, z_3\}$. Simple calculations verify that Assumption 2 holds for such noise. Next, the update rule of the method (6) in the case $n = 1$ is

$$x^{t+1} = x^t - \gamma g^t = x^t - \gamma(\nabla f(x^t) + \mathrm{clip}_\tau(\nabla f(x^t, \xi^t) - \nabla f(x^t))) = x^t - L\gamma x^t - \gamma\,\mathrm{clip}_\tau(\xi^t).$$

Since $\tau < \|z_i\|$ for any $i \in \{1, 2, 3\}$ clipping is always active and we have

$$\begin{aligned}
\mathbb{E}\left[\mathrm{clip}_\tau(\xi^t)\right] &= \frac{1}{3}\,\mathrm{clip}_\tau(z_1) + \frac{1}{3}\,\mathrm{clip}_\tau(z_2) + \frac{1}{3}\,\mathrm{clip}_\tau(z_3) \\
&= \frac{1}{3}\frac{\tau}{\|z_1\|}z_1 + \frac{1}{3}\frac{\tau}{\|z_2\|}z_2 + \frac{1}{3}\frac{\tau}{\|z_3\|}z_3 \\
&= \frac{1}{3}\frac{\tau}{\frac{3\sqrt{3}\sigma}{10}}\frac{\sigma\sqrt{3}}{10}\begin{pmatrix} 3 \\ 0 \end{pmatrix} + \frac{1}{3}\frac{\tau}{\frac{4\sqrt{3}\sigma}{10}}\frac{\sigma\sqrt{3}}{10}\begin{pmatrix} 0 \\ 4 \end{pmatrix} + \frac{1}{3}\frac{\tau}{\frac{5\sqrt{3}\sigma}{10}}\frac{\sigma\sqrt{3}}{10}\begin{pmatrix} -3 \\ -4 \end{pmatrix} \\
&= \frac{\tau}{9}\begin{pmatrix} 3 \\ 0 \end{pmatrix} + \frac{\tau}{12}\begin{pmatrix} 0 \\ 4 \end{pmatrix} + \frac{\tau}{15}\begin{pmatrix} -3 \\ -4 \end{pmatrix} \\
&= \underbrace{\frac{\tau}{15}\begin{pmatrix} 2 \\ 1 \end{pmatrix}}_{:=h}.
\end{aligned}$$

Thus, we obtain

$$\begin{aligned}
\mathbb{E}\left[x^T\right] &= (1 - L\gamma)\mathbb{E}\left[x^{T-1}\right] - \gamma\mathbb{E}\left[\mathrm{clip}_\tau(\xi^t)\right] \\
&= (1 - L\gamma)\mathbb{E}\left[x^{T-1}\right] - \gamma h \\
&= (1 - L\gamma)^T x^0 - \gamma h \sum_{t=0}^{T-1}(1 - L\gamma)^{T-1-t} \\
&= (1 - L\gamma)^T \begin{pmatrix} 0 \\ x_{(2)}^0 \end{pmatrix} - \frac{\tau\gamma}{15}\begin{pmatrix} 2 \\ 1 \end{pmatrix}\frac{1 - (1 - L\gamma)^T}{1 - (1 - L\gamma)} \\
&= (1 - L\gamma)^T \begin{pmatrix} 0 \\ x_{(2)}^0 \end{pmatrix} - \frac{\tau}{15L}\begin{pmatrix} 2 \\ 1 \end{pmatrix}(1 - (1 - L\gamma)^T).
\end{aligned}$$

Therefore, since $x_{(2)}^0 < 0$ we have

$$\begin{aligned}
\mathbb{E}\left[\|\nabla f(x^T)\|^2\right] &= \mathbb{E}\left[\|Lx^T\|^2\right] \\
&= \left\|\mathbb{E}\left[Lx^T\right]\right\|^2 + \mathbb{E}\left[\|Lx^T - \mathbb{E}\left[Lx^T\right]\|^2\right] \\
&\geq \left\|\mathbb{E}\left[Lx^T\right]\right\|^2 \\
&= \frac{4\tau^2}{165}\left(1 - (1 - L\gamma)^T\right)^2 + L^2\left((1 - L\gamma)^T x_{(2)}^0 - \frac{\tau}{15L}\left(1 - (1 - L\gamma)^T\right)\right)^2 \\
&\geq \frac{4\tau^2}{165}\left(1 - (1 - L\gamma)^T\right)^2 + (1 - L\gamma)^{2T}\|Lx^0\|^2 + \frac{\tau^2}{165}(1 - (1 - L\gamma)^T)^2 \\
&= \frac{\tau^2}{45}\left(1 - (1 - L\gamma)^T\right)^2 + (1 - L\gamma)^{2T}\|\nabla f(x^0)\|^2.
\end{aligned}$$

Note that the function $a(1-x)^2 + x^2 b \geq \frac{ab}{a+b}$. Applying this result for $a = \frac{\tau^2}{45}$, $b = \|\nabla f(x^0)\|^2$, and $x = (1 - L\gamma)^T$ we get

$$\mathbb{E}\left[\|\nabla f(x^T)\|^2\right] \geq \frac{\frac{\tau^2}{45}\|\nabla f(x^0)\|^2}{\frac{\tau^2}{45} + \|\nabla f(x^0)\|^2} \geq \frac{1}{2}\min\left\{\|\nabla f(x^0)\|^2, \frac{\tau^2}{45}\right\}.$$

**The case $n > 1$.** If $n > 1$ then we can consider a similar example where each client is quadratic $\frac{L}{2}\|x\|^2$ and the stochastic gradient is constructed as $\nabla f_i(x^t, \xi_i^t) = \nabla f_i(x^t) + \xi_i^t = Lx^t + \xi_i^t$ where $\xi_i^t$ is sampled uniformly at random from vectors $\{z_1, z_2, z_3\}$ such that

$$z_1 = \begin{pmatrix} 3 \\ 0 \end{pmatrix}\sqrt{\frac{3\sigma^2}{100B}}, \quad z_2 = \begin{pmatrix} 0 \\ 4 \end{pmatrix}\sqrt{\frac{3\sigma^2}{100B}}, \quad z_1 = \begin{pmatrix} -3 \\ -4 \end{pmatrix}\sqrt{\frac{3\sigma^2}{100B}}.$$

Then, Assumption 2 is satisfied with $\sigma^2/B$. Therefore, if $x_{(2)}^0 = -1$, $\varepsilon < \frac{L}{\sqrt{2}}$, and $\tau \geq \frac{\varepsilon}{3\sqrt{10}}$, this implies that $B \leq \frac{243\sigma^2}{5\varepsilon^2} < \frac{27\sigma^2}{50\tau^2}$, and

$$\mathbb{E}\left[\|\nabla f(x^T)\|^2\right] \geq \frac{1}{2}\min\left\{\|\nabla f(x^0)\|^2, \frac{\tau^2}{45}\right\} \geq \varepsilon^2.$$

$\square$

## D  PROOF OF THEOREM 2

**Lemma 5.** Let each $f_i$ be $L$-smooth. Then we have the following inequality

$$\|v_i^{t+1} - g_i^t\| \leq \max\left\{0, \|v_i^t - g_i^{t-1}\| - \tau\right\} + \beta L\gamma\|g^t\| + \beta\|\nabla f_i(x^t) - v_i^t\|. \tag{20}$$

*Proof.* We have

$$\begin{aligned}
\|v_i^{t+1} - g_i^t\| &\overset{(i)}{=} \|(1-\beta)v_i^t + \beta\nabla f_i(x^{t+1}) - g_i^t\| \\
&\overset{(ii)}{\leq} \|v_i^t - g_i^t\| + \beta\|\nabla f_i(x^{t+1}) - v_i^t\| \\
&\overset{(iii)}{\leq} \max\left\{0, \|v_i^t - g_i^{t-1}\| - \tau\right\} + \beta\|\nabla f_i(x^{t+1}) - \nabla f_i(x^t)\| + \beta\|\nabla f_i(x^t) - v_i^t\| \\
&\overset{(iv)}{\leq} \max\left\{0, \|v_i^t - g_i^{t-1}\| - \tau\right\} + \beta L\|x^{t+1} - x^t\| + \beta\|\nabla f_i(x^t) - v_i^t\| \\
&\overset{(v)}{=} \max\left\{0, \|v_i^t - g_i^{t-1}\| - \tau\right\} + \beta L\gamma\|g^t\| + \beta\|\nabla f_i(x^t) - v_i^t\|,
\end{aligned}$$

where $(i)$ follows from the update rule of $v_i^t$ in deterministic case; $(ii)$ from triangle inequality; $(iii)$ from the update rule of $g_i^t$, properties of clipping from Lemma 3, and triangle inequality; $(iv)$ from $L$-smoothness of $f_i$; $(v)$ from the update rule of $x^t$. $\square$

**Lemma 6.** Let each $f_i$ be $L$-smooth and $\Delta \geq \Phi^0$. Assume that the following inequalities hold

1. $\|g^{t-1}\| \leq \sqrt{64L\Delta} + 3(B - \tau)$;

2. $\|\nabla f_i(x^{t-1}) - v_i^{t-1}\| \leq \sqrt{4L\Delta} + \frac{3}{2}(B - \tau)$;

3. $\|v_i^t - g_i^{t-1}\| \leq B$;

4. $\gamma \leq \frac{1}{12L}$;

5. $0 \leq \beta \leq \frac{1}{2}$;

6. $\Phi^t \leq \Delta$

Then we have

$$\|g^t\| \leq \sqrt{64L\Delta} + 3(B - \tau). \tag{21}$$

*Proof.* We have

$$\|g^t\| \overset{(i)}{=} \left\| \frac{1}{n} \sum_{i=1}^{n} g_i^{t-1} + \text{clip}_\tau(v_i^t - g_i^{t-1}) \right\|$$

$$= \left\| \frac{1}{n} \sum_{i=1}^{n} \nabla f_i(x^t) + (v_i^t - \nabla f_i(x^t)) + \text{clip}_\tau(v_i^t - g_i^{t-1}) - (v_i^t - g_i^{t-1}) \right\|$$

$$\overset{(ii)}{\leq} \|\nabla f(x^t)\| + \frac{1}{n} \sum_{i=1}^{n} \|v_i^t - \nabla f_i(x^t)\| + \frac{1}{n} \sum_{i=1}^{n} \max\left\{ 0, \|v_i^t - g_i^{t-1}\| - \tau \right\},$$

where $(i)$ follows from the update rule $g_i^t$; $(ii)$ from triangle inequality and clipping properties from Lemma 3. We continue to bound $\|g^t\|$ as follows

$$\|g^t\| \overset{(i)}{\leq} \|\nabla f(x^{t-1})\| + \|\nabla f(x^t) - \nabla f(x^{t-1})\| + \frac{1}{n} \sum_{i=1}^{n} (1-\beta) \|v_i^{t-1} - \nabla f_i(x^t)\| + B - \tau$$

$$\overset{(ii)}{\leq} \|\nabla f(x^{t-1})\| + \|\nabla f(x^t) - \nabla f(x^{t-1})\| + \frac{1}{n} \sum_{i=1}^{n} (1-\beta) \|v_i^{t-1} - \nabla f_i(x^{t-1})\|$$

$$+ \frac{1}{n} \sum_{i=1}^{n} (1-\beta) \|\nabla f_i(x^t) - \nabla f_i(x^{t-1})\| + B - \tau$$

$$\overset{(iii)}{\leq} \sqrt{2L(f(x^{t-1}) - f^*)} + L\gamma(2-\beta)\|g^{t-1}\| + (1-\beta)\frac{1}{n} \sum_{i=1}^{n} \|v_i^{t-1} - \nabla f_i(x^{t-1})\| + B - \tau$$

$$\overset{(iv)}{\leq} \sqrt{2L\Phi^t} + 2L\gamma\|g^{t-1}\| + (1-\beta)\frac{1}{n} \sum_{i=1}^{n} \|v_i^{t-1} - \nabla f_i(x^{t-1})\| + B - \tau$$

$$\overset{(v)}{\leq} \sqrt{2L\Delta} + 2L\gamma \left( \sqrt{64L\Delta} + 3(B-\tau) \right) + \left( \sqrt{4L\Delta} + \frac{3}{2}(B-\tau) \right) + B - \tau$$

$$= \left( \sqrt{2} + 16L\gamma + 2 \right) \sqrt{L\Delta} + (6L\gamma + 1 + 3/2)(B - \tau),$$

where $(i)$ follows from triangle inequality and update of $v_i^t$, and assumption 3 in the statement of the lemma; $(ii)$ from triangle inequality; $(iii)$ from properties of smooth function from Lemma 4 and update rule of $x^t$; $(iv)$ from the definition of $\Phi^t$; $(v)$ from assumption 1, 2, and 6 in the statement of the lemma. Since $\gamma \leq \frac{1}{12L} \leq \frac{6-\sqrt{2}}{16L}$, then $16L\gamma + \sqrt{2} + 2 \leq 8$, and $\gamma \leq \frac{1}{12L}$, then $6L\gamma + 5/2 \leq 3$.  $\square$

**Lemma 7.** Let each $f_i$ be $L$-smooth and $\Delta \geq \Phi^0$. Let the following inequalities hold

1. $4L\gamma = \beta$ and $\gamma \leq \frac{1}{4L}$;

2. $\|\nabla f_i(x^{t-1}) - v_i^{t-1}\| \leq \sqrt{4L\Delta} + \frac{3}{2}(B-\tau)$;

3. $\|g^{t-1}\| \leq \sqrt{64L\Delta} + 3(B-\tau)$.

Then we have

$$\|\nabla f_i(x^t) - v_i^t\| \leq \sqrt{4L\Delta} + \frac{3}{2}(B-\tau) \quad \forall i \in [n]. \tag{22}$$

*Proof.* We have

$$\|\nabla f_i(x^t) - v_i^t\| \overset{(i)}{=} \|\nabla f_i(x^t) - (1-\beta)v_i^{t-1} - \beta\nabla f_i(x^t)\|$$

$$= (1-\beta)\|\nabla f_i(x^t) - v_i^{t-1}\|$$

$$\overset{(ii)}{\leq} (1-\beta)L\gamma\|g^{t-1}\| + (1-\beta)\|\nabla f_i(x^{t-1}) - v_i^{t-1}\|$$

$$\overset{(iii)}{\leq} L\gamma \left( \sqrt{64L\Delta} + 3(B-\tau) \right) + (1-\beta) \left( \sqrt{4L\Delta} + \frac{3}{2}(B-\tau) \right)$$

$$= (8L\gamma + 2(1-\beta))\sqrt{L\Delta} + (3L\gamma + 3(1-\beta)/2)(B-\tau),$$

where $(i)$ follows from the update rule of $v_i^t$; $(ii)$ from triangle inequality, smoothness, and update of $x^t$; $(iii)$ from assumption 2-3 of the statement of the lemma. Since $4L\gamma = \beta$, then $4L\gamma + 2(1-\beta) = 2$ and $3L\gamma + 3(1-\beta)/2 = 3L\gamma + \frac{3}{2}(1 - 4L\gamma) \le \frac{3}{2}$. $\square$

**Lemma 8.** Let each $f_i$ be $L$-smooth, $\Delta \ge \Phi^0$ and $i \in \mathcal{I}_t$. Let the following inequalities hold

1. $\beta = 4L\gamma$ and $\beta \le \frac{1}{2}$;

2. $\gamma \le \frac{\tau}{48L\sqrt{L\Delta}}$;

3. $\gamma \le \frac{\tau}{30L(B-\tau)}$;

4. $\|g^t\| \le \sqrt{64L\Delta} + 3(B - \tau)$;

5. $\|\nabla f_i(x^t) - v_i^t\| \le \sqrt{4L\Delta} + \frac{3}{2}(B - \tau)$.

Then

$$\|v_i^{t+1} - g_i^t\| \le \|v_i^t - g_i^{t-1}\| - \frac{\tau}{2}. \tag{23}$$

*Proof.* Since $i \in \mathcal{I}_t$, then $\|v_i^t - g_i^{t-1}\| > \tau$, thus from Lemma 5 we have

$$\|v_i^{t+1} - g_i^t\| \le \|v_i^t - g_i^{t-1}\| - \tau + \beta L\gamma\|g^t\| + \beta\|\nabla f_i(x^t) - v_i^t\|$$

$$\overset{(i)}{\le} \|v_i^t - g_i^{t-1}\| - \tau + \frac{1}{2}L\gamma\left(\sqrt{64L\Delta} + 3(B-\tau)\right) + \beta\left(\sqrt{4L\Delta} + \frac{3}{2}(B-\tau)\right)$$

$$= \|v_i^t - g_i^{t-1}\| - \tau + (4L\gamma + 2\beta)\sqrt{L\Delta} + (3L\gamma/2 + 3\beta/2)(B - \tau),$$

where $(i)$ follows from assumptions 4-5 of the statement of the lemma. Since $\beta = 4L\gamma$, we have

$$\|v_i^{t+1} - g_i^t\| \le \|v_i^t - g_i^{t-1}\| - \tau + 12L\gamma\sqrt{L\Delta} + \frac{15}{2}L\gamma(B - \tau).$$

Since $\gamma \le \frac{\tau}{48L\sqrt{L\Delta}}$, then $12L\gamma\sqrt{L\Delta} \le \frac{\tau}{4}$, and since $\gamma \le \frac{\tau}{30L(B-\tau)}$, then $\frac{15}{2}L\gamma(B - \tau) \le \frac{\tau}{4}$. Therefore, we have

$$\|v_i^{t+1} - g_i^t\| \le \|v_i^t - g_i^{t-1}\| - \frac{\tau}{2}.$$

$\square$

**Lemma 9.** Let each $f_i$ be $L$-smooth. Then $\widetilde{P}^t$ decreases as

$$\widetilde{P}^{t+1} \le (1 - \beta)\widetilde{P}^t + \frac{3L^2}{\beta}R^t. \tag{24}$$

*Proof.* We have

$$\|v_i^{t+1} - \nabla f_i(x^{t+1})\|^2 \overset{(i)}{=} \|(1 - \beta)v_i^t + \beta\nabla f_i(x^{t+1}) - \nabla f_i(x^{t+1})\|^2$$

$$= (1 - \beta)^2\|\nabla f_i(x^{t+1}) - v_i^t\|^2$$

$$\overset{(ii)}{\le} (1 - \beta)^2(1 + \beta/2)\|v_i^t - \nabla f_i(x^t)\|^2$$

$$+ (1 - \beta)^2(1 + 2/\beta)\|\nabla f_i(x^t) - \nabla f_i(x^{t+1})\|^2$$

$$\overset{(iii)}{\le} (1 - \beta)\|v_i^t - \nabla f_i(x^t)\|^2 + \frac{3L^2}{\beta}\|x^t - x^{t+1}\|^2,$$

where $(i)$ follows from the update rule of $v_i^t$; $(ii)$ from the inequality $\|a + b\|^2 \le (1 + \beta/2)\|a\|^2 + (1 + 2/\beta)\|b\|^2$; $(iii)$ from smoothness. Averaging the inequalities above across $i \in [n]$, we get the statement of the lemma. $\square$

Similarly, we can get the descent of $P^t$.

**Lemma 10.** Let each $f_i$ be $L$-smooth. Then $P^t$ decreases as

$$P^{t+1} \leq (1-\beta)P^t + \frac{3L^2}{\beta}R^t. \tag{25}$$

Now we present the descent of $\widetilde{V}^t$.

**Lemma 11.** Let each $f_i$ be $L$-smooth. Let $\|v_i^t - g_i^{t-1}\| \leq B$ for all $i \in [n]$. Then

$$\|g_i^t - v_i^t\|^2 \leq (1-\eta)\|g_i^{t-1} - v_i^{t-1}\|^2 + \frac{4\beta^2}{\eta}\|v_i^{t-1} - \nabla f_i(x^{t-1})\|^2 + \frac{4\beta^2 L^2}{\eta}R^{t-1}.$$

*Proof.* Since $\|v_i^t - g_i^{t-1}\| \leq B$, we have $\eta_i^t \geq \eta$. Thus, we have

$$
\begin{aligned}
\|g_i^t - v_i^t\|^2 &\stackrel{(i)}{=} \|g_i^{t-1} + \mathrm{clip}_\tau(v_i^t - g_i^{t-1}) - v_i^t\|^2 \\
&\stackrel{(ii)}{\leq} (1-\eta_i^t)^2\|g_i^{t-1} - v_i^t\|^2 \\
&\stackrel{(iii)}{\leq} (1-\eta)^2\|g_i^{t-1} - v_i^t\|^2 \\
&\stackrel{(iv)}{=} (1-\eta)^2\|g_i^{t-1} - (1-\beta)v_i^{t-1} - \beta\nabla f_i(x^t)\|^2 \\
&\stackrel{(v)}{\leq} (1-\eta)^2(1+\rho)\|g_i^{t-1} - v_i^{t-1}\|^2 + (1-\eta)^2(1+\rho^{-1})\beta^2\|v_i^{t-1} - \nabla f_i(x^t)\|^2 \\
&\stackrel{(vi)}{\leq} (1-\eta)^2(1+\rho)\|g_i^{t-1} - v_i^{t-1}\|^2 + 2(1-\eta)^2(1+\rho^{-1})\beta^2\|v_i^{t-1} - \nabla f_i(x^{t-1})\|^2 \\
&\quad + 2(1-\eta)^2(1+\rho^{-1})\beta^2 L^2\|x^{t-1} - x^t\|^2,
\end{aligned}
$$

where $(i)$ follows from the update rule of $g_i^t$; $(ii)$ from properties of clipping from Lemma 3; $(iii)$ from the fact that $\eta_i^t \geq \eta$; $(iv)$ from the update rule of $v_i^t$; $(v)$ from the inequality $\|a+b\|^2 \leq (1+r/2)\|a\|^2 + (1+2/r)\|b\|^2$ for any positive $r$; $(vi)$ from the inequality $\|a+b\|^2 \leq (1+r/2)\|a\|^2 + (1+2/r)\|b\|^2$ for any positive $r$ and smoothness. If we choose $\rho = \eta/2$, we get

$$\|g_i^t - v_i^t\|^2 \leq (1-\eta)\|g_i^{t-1} - v_i^{t-1}\|^2 + \frac{4\beta^2}{\eta}\|v_i^{t-1} - \nabla f_i(x^{t-1})\|^2 + \frac{4\beta^2 L^2}{\eta}R^{t-1}.$$

$\square$

**Theorem 5** (Full statement of Theorem 2). *Let Assumptions 1 holds. Let $B := \max_i \|\nabla f_i(x^0)\| > 3\tau$ and $\Delta \geq \Phi^0$. Assume the following inequalities hold*

1. **stepsize restrictions:** $\gamma \leq \frac{1}{12L}, \gamma \leq \frac{\tau}{48L\sqrt{L\Delta}}, \gamma \leq \frac{\tau}{30L(B-\tau)}$, *and*

$$\frac{2}{3} - \frac{16\beta^2 L^2}{\eta^2}\gamma^2 - \frac{48L^2}{\eta^2}\gamma^2 \geq 0;$$

2. **momentum restrictions:** $\beta = 4L\gamma \leq \frac{1}{2}$.

*Then the Lyapunov function decreases as*

$$\Phi^{t+1} \leq \Phi^t - \frac{\gamma}{2}\|\nabla f(x^t)\|^2,$$

*therefore we obtain*

$$\frac{1}{T}\sum_{t=0}^{T-1}\|\nabla f(x^t)\|^2 \leq \frac{2\Delta}{T} = \mathcal{O}\left(\frac{1}{T}\right). \tag{26}$$

*Moreover, after at most $\frac{2B}{\tau}$ iterations, the clipping operator will be turned off for all workers.*

*Proof.* We prove the main theorem by induction. The conventional choice is

$$\nabla f_i(x^{-1}) = v_i^{-1} = g_i^{-1} = 0, \quad \Phi^{-1} = +\infty.$$

We will show that

1. the Lyapunov function decreases as $\Phi^{t+1} \le \Phi^t - \frac{\gamma}{2}\|\nabla f(x^t)\|^2$;

2. $\|g^t\| \le \sqrt{64L\Delta} + 3(B - \tau)$;

3. $\|v_i^t - \nabla f_i(x^t)\| \le \sqrt{4L\Delta} + \frac{3}{2}(B - \tau)$;

4. and $\|v_i^t - g_i^{t-1}\| \le B - \frac{t\tau}{2}$.

First, we prove that the base of induction holds.

**Base of induction.**

1. $\|v_i^0 - g_i^{-1}\| = \|v_i^0\| = \beta\|\nabla f_i(x^0)\| \le \frac{1}{2}B \le B$ holds;

2. $g^0 = \frac{1}{n}\sum_{i=1}^{n}(g_i^{-1} + \mathrm{clip}_\tau(v_i^0 - g_i^{-1})) = \frac{1}{n}\sum_{i=1}^{n}\mathrm{clip}_\tau(\beta\nabla f_i(x^0))$. Therefore, we have

$$\|g^0\| \le \left\| \frac{1}{n}\sum_{i=1}^{n}\beta\nabla f_i(x^0) + (\mathrm{clip}_\tau(\beta\nabla f_i(x^0)) - \beta\nabla f_i(x^0)) \right\|$$

$$\le \beta\|\nabla f(x^0)\| + \frac{1}{n}\sum_{i=1}^{n}\max\left\{0, \beta\|\nabla f_i(x^0)\| - \tau\right\}$$

$$\le \beta\sqrt{2L(f(x^0) - f^*)} + B - \tau$$

$$\le \sqrt{64L\Delta} + 3(B - \tau).$$

3. We have

$$\|v_i^0 - \nabla f_i(x^0)\| = \|\beta\nabla f_i(x^0) - \nabla f_i(x^0)\|$$

$$\le (1 - \beta)B$$

$$\le \sqrt{4L\Delta} + \frac{3}{2}(B - \tau)$$

4. $\Phi^0 \le \Phi^{-1} - \frac{\gamma}{2}\|\nabla f(x^{-1})\|^2 = \Phi^{-1}$ holds.

**Transition of induction.** Assume that for $K$ we have that for all $t \in [0, K]$

1. $\Phi^t \le \Delta$;

2. $\|g^t\| \le \sqrt{64L\Delta} + 3(B - \tau)$;

3. $\|v_i^t - \nabla f_i(x^t)\| \le \sqrt{4L\Delta} + \frac{3}{2}(B - \tau)$;

4. $\|v_i^t - g_i^{t-1}\| \le B$ for $i \in \mathcal{I}_t$.

CASE $|\mathcal{I}_{K+1}| > 0$. Since all requirements of Lemma 8 are satisfied at iteration $K$ we get for all $i \in \mathcal{I}_{K+1}$

$$\|v_i^{K+1} - g_i^K\| \le \|v_i^K - g_i^{K-1}\| - \frac{\tau}{2} \le B - \frac{\tau}{2}.$$

Similarly due to the assumption of induction, from Lemma 6 we get that

$$\|g^{K+1}\| \le \sqrt{64L\Delta} + 3(B - \tau),$$

and from Lemma 7

$$\|\nabla f_i(x^{K+1}) - v_i^{K+1}\| \le \sqrt{4L\Delta} + \frac{3}{2}(B - \tau).$$

This means that steps 1-3 in the assumption of the induction are also verified for step $K + 1$. The remaining part is the descent of the Lyapunov function. For $\widetilde{V}^{K+1}$ we have Lemma 11 since $\|v_i^{K+1} - g_i^K\| \le B - \frac{\tau}{2}$

$$\widetilde{V}^{K+1} \le (1 - \eta)\widetilde{V}^K + \frac{4\beta^2}{\eta}\widetilde{P}^K + \frac{4\beta^2 L^2}{\eta}R^K.$$

Combining this result with the claims of Lemmas 2, 9 and 10 we get

$$
\begin{aligned}
\Phi^{K+1} &= \delta^{K+1} + \frac{\gamma}{\eta}\widetilde{V}^{K+1} + \frac{4\gamma\beta}{\eta^2}\widetilde{P}^{K+1} + \frac{\gamma}{\beta}P^{K+1} \\
&\le \delta^K - \frac{\gamma}{2}\|\nabla f(x^K)\|^2 - \frac{1}{4\gamma}R^K + \gamma\widetilde{V}^K + \gamma P^K \\
&\quad + \frac{\gamma}{\eta}\left((1-\eta)\widetilde{V}^K + \frac{4\beta^2}{\eta}\widetilde{P}^K + \frac{4\beta^2 L^2}{\eta}R^K\right) \\
&\quad + \frac{4\gamma\beta}{\eta^2}\left((1-\beta)\widetilde{P}^K + \frac{3L^2}{\beta}R^K\right) \\
&\quad + \frac{\gamma}{\beta}\left((1-\beta)P^K + \frac{3L^2}{\beta}R^K\right) \\
&= \delta^K - \frac{\gamma}{2}\|\nabla f(x^K)\|^2 + \frac{\gamma}{\eta}\widetilde{V}^K(1-\eta+\eta) + \frac{4\gamma\beta}{\eta^2}\widetilde{P}^{t^*}(1-\beta+\beta) \\
&\quad + \frac{\gamma}{\beta}P^K(1-\beta+\beta) - \frac{1}{4\gamma}\left(1 - \frac{16\beta^2 L^2}{\eta^2}\gamma^2 - \frac{48L^2}{\eta^2}\gamma^2 - \frac{12L^2}{\beta^2}\gamma^2\right)R^K \\
&\le \Phi^K - \frac{\gamma}{2}\|\nabla f(x^K)\|^2 - \frac{1}{4\gamma}\left(1 - \frac{16\beta^2 L^2}{\eta^2}\gamma^2 - \frac{48L^2}{\eta^2}\gamma^2 - \frac{12L^2}{\beta^2}\gamma^2\right)R^K.
\end{aligned}
$$

Since we choose $\beta^2 = 64L^2\gamma^2$, then $-\frac{1}{\beta^2} = -\frac{1}{64L^2\gamma^2}$ and $-\frac{12L^2}{\beta^2}\gamma^2 = -\frac{12L^2}{64^2L^2\gamma^2}\gamma^2 \ge -\frac{1}{3}$. Therefore,

$$
1 - \frac{16\beta^2 L^2}{\eta^2}\gamma^2 - \frac{48L^2}{\eta^2}\gamma^2 - \frac{12L^2}{\beta^2}\gamma^2 \ge \frac{2}{3} - \frac{16\beta^2 L^2}{\eta^2}\gamma^2 - \frac{48L^2}{\eta^2}\gamma^2 \ge 0,
$$

by the choice of $\gamma$. Thus, we get

$$
\Phi^{K+1} \le \Phi^K - \frac{\gamma}{2}\|\nabla f(x^K)\|^2.
$$

In particular, this implies $\Phi^{K+1} \le \Phi^K \le \Delta$.

CASE $|\mathcal{I}_{K+1}| = 0$. In this case $\eta_i^{K+1} = 1$ for all $i \in [n]$, i.e. $\text{clip}_\tau(v_i^{K+1} - g_i^K) = v_i^{K+1} - g_i^K$ that leads to $g_i^{K+1} = v_i^{K+1}$. Thus, $\widetilde{V}^{K+1} = 0$. We can perform similar steps as before for $\Phi^{K+1}$ and get less restrictive inequality

$$
\Phi^{K+1} \le \Phi^K - \frac{\gamma}{2}\|\nabla f(x^K)\|^2 - \frac{1}{4\gamma}\left(1 - \frac{48L^2}{\eta^2}\gamma^2 - \frac{12L^2}{\beta^2}\gamma^2\right)R^K.
$$

Again, $1 - \frac{48L^2}{\eta^2}\gamma^2 - \frac{12L^2}{\beta^2}\gamma^2 \ge \frac{2}{3} - \frac{48L^2}{\eta}\gamma^2 \ge 0$ which is satisfied by the choice of $\gamma$.

We conclude that in both cases the Lyapunov function decreases as $\Phi^{K+1} \le \Phi^K - \frac{\gamma}{2}\|\nabla f(x^K)\|^2$, and consequently, $\Phi^{K+1} \le \Delta$. This finalizes the induction step. Therefore, we can guarantee that for all iterations $t \in [0, T-1]$ we have

$$
\Phi^{t+1} \le \Phi^t - \frac{\gamma}{2}\|\nabla f(x^t)\|^2 \Rightarrow \frac{1}{T}\sum_{t=0}^{T-1}\|\nabla f(x^t)\|^2 \le \frac{2\Delta}{\gamma T}.
$$

Moreover, the proof shows that the clipping operator will be eventually turned off since $\|v_i^t - g_i^{t-1}\| \le B - \frac{t\tau}{2}$, i.e. after at most $\frac{2B}{\tau}$ iterations. $\qquad\square$

**Remark 1.** With $v_i^{-1} = g_i^{-1} = 0$ we have

$$\Phi^0 = \delta^0 + \frac{\gamma}{\eta}\frac{1}{n}\sum_{i=1}^{n}\|\operatorname{clip}_\tau(\beta\nabla f_i(x^0)) - \beta\nabla f_i(x^0)\|^2 + \frac{4\gamma\beta}{\eta^2}\frac{1}{n}(1-\beta)^2\sum_{i=1}^{n}\|\nabla f_i(x^0)\|^2$$

$$+ \frac{\gamma}{\beta}(1-\beta)^2\left\|\nabla f(x^0)\right\|^2$$

$$\leq \delta^0 + \frac{\gamma}{\eta}\frac{1}{n}\sum_{i=1}^{n}\max\left\{(\|\nabla f(x^0)\| - \tau)^2, 0\right\} + \frac{16L\gamma^2}{\eta^2}\frac{1}{n}(1-\beta)^2\sum_{i=1}^{n}\|\nabla f_i(x^0)\|^2$$

$$+ \frac{1}{4L}(1-\beta)^2\left\|\nabla f(x^0)\right\|^2$$

$$\leq \frac{3}{2}\delta^0 + \frac{\gamma}{\eta}\frac{1}{n}\sum_{i=1}^{n}\max\left\{(\|\nabla f_i(x^0)\| - \tau)^2, 0\right\} + \frac{16L\gamma^2}{\eta^2}\frac{1}{n}(1-\beta)^2\sum_{i=1}^{n}\|\nabla f_i(x^0)\|^2.$$

We have the stepsize restriction

$$\frac{2}{3} - \frac{64L^4\gamma^2}{\eta^2} - \frac{48L^2\gamma^2}{\eta^2} \geq 0. \tag{27}$$

For inequality of the form $a\gamma^2 + b\gamma \leq 1$ the stepsize restriction of the form $\gamma \leq \frac{1}{\sqrt{a}+b}$ is tight up to a constant factor 2, i.e. $\frac{2}{\sqrt{a}+b}$ does not satisfy the inequality (see Lemma 5 in (Richtárik et al., 2021)). Using this lemma in our case we get that the stepsize satisfying Equation (27) should also satisfy

$$L^2\gamma^2 \leq 2 \cdot \frac{\eta}{72/\eta + 4\sqrt{6}}.$$

This implies that $L^2\gamma^2 \leq \frac{\eta}{4\sqrt{6}}$ and $L^2\gamma^2 \leq \frac{\eta^2}{72}$. Consequently, it also satisfies $\frac{\gamma}{\eta} \leq \frac{1}{6L\sqrt{2}}$ (from the last inequality). Therefore, we have

$$\Phi^0 \leq \frac{3}{2}\delta^0 + \frac{1}{6L\sqrt{2}}\frac{1}{n}\sum_{i=1}^{n}\max\left\{(\|\nabla f_i(x^0)\| - \tau)^2, 0\right\} + \frac{2}{9L}\frac{1}{n}\sum_{i=1}^{n}\|\nabla f_i(x^0)\|^2$$

$$\leq \frac{3}{2}\delta^0 + \left(\frac{1}{6L\sqrt{2}} + \frac{2}{9L}\right)\frac{1}{n}\sum_{i=1}^{n}\|\nabla f_i(x^0)\|^2,$$

which is independent of $\tau$, and can be use as a bound for $\Delta$.

# E    PROOF OF THEOREM 4

We define constants $a$, $b$, and $c$ as follows that will be used later in the proofs:

$$a := \left( \sqrt{2} + 2\sqrt{3 \log \frac{6(T+1)}{\alpha}} \right) \sqrt{d} \sigma_\omega \sqrt{T/n},$$

$$b^2 := 2\sigma^2 \log \left( \frac{6(T+1)n}{\alpha} \right), \tag{28}$$

$$c^2 := \left( \sqrt{2} + 2\sqrt{3 \log \frac{6(T+1)}{\alpha}} \right)^2 \sigma^2,$$

where $T$ is the number of iterations, $n$ is the number of workers, $d$ is the dimension of the problem, $\sigma$ is from Assumption 2, $\alpha \in (0,1)$ is a constant, and $\sigma_\omega$ is the variance of DP noise.

**Lemma 12.** Let each $f_i$ be $L$-smooth. Then we have the following inequality with probability 1

$$\|v_i^{t+1} - g_i^t\| \le \max \left\{ 0, \|v_i^t - g_i^{t-1}\| - \tau \right\} + \beta L \gamma \|g^t\| + \beta \|\nabla f_i(x^t) - v_i^t\| + \beta \|\theta_i^{t+1}\|. \tag{29}$$

*Proof.* We have

$$\|v_i^{t+1} - g_i^t\| \stackrel{(i)}{=} \|(1-\beta)v_i^t + \beta \nabla f_i(x^{t+1}, \xi_i^{t+1}) - g_i^t\|$$

$$\stackrel{(ii)}{\le} \|v_i^t - g_i^t\| + \beta \|\nabla f_i(x^{t+1}, \xi_i^{t+1}) - v_i^t\|$$

$$\stackrel{(iii)}{=} \|v_i^t - \mathrm{clip}_\tau(v_i^t - g_i^{t-1}) - g_i^{t-1}\| + \beta \|\nabla f_i(x^{t+1}, \xi_i^{t+1}) - v_i^t\|$$

$$\stackrel{(iv)}{\le} \max \left\{ 0, \|v_i^t - g_i^{t-1}\| - \tau \right\} + \beta \|\nabla f_i(x^{t+1}, \xi_i^{t+1}) - \nabla f_i(x^{t+1})\|$$
$$\quad + \beta \|\nabla f_i(x^{t+1}) - \nabla f_i(x^t)\| + \beta \|\nabla f_i(x^t) - v_i^t\|$$

$$\stackrel{(v)}{\le} \max \left\{ 0, \|v_i^t - g_i^{t-1}\| - \tau \right\} + \beta L \|x^{t+1} - x^t\| + \beta \|\nabla f_i(x^t) - v_i^t\| + \beta \|\theta_i^{t+1}\|$$

$$\stackrel{(vi)}{=} \max \left\{ 0, \|v_i^t - g_i^{t-1}\| - \tau \right\} + \beta L \gamma \|g^t\| + \beta \|\nabla f_i(x^t) - v_i^t\| + \beta \|\theta_i^{t+1}\|,$$

where $(i)$ follows from the update rule of $v_i^t$; $(ii)$ from triangle inequality; $(iii)$ from the update rule of $g_i^t$; $(iv)$ from the properties of the clipping operator from Lemma 3 and triangle inequality; $(v)$ from smoothness; $(vi)$ from the update rule of $x^t$. □

Let us choose $p \in [0.2, 0.8]$. With this choice we have $3x^{1-p} \ge 4x$ for any $x \in (0, 1/12]$.

**Lemma 13.** Let each $f_i$ be $L$-smooth and $\Delta \ge \Phi^0$. Assume that the following inequalities hold

1. $g^0 = \frac{1}{n} \sum_{i=1}^n g_i^0$;

2. $\|g^{t-1}\| \le \sqrt{64L\Delta} + 3(B - \tau) + 3b + 3a$;

3. $\|\nabla f_i(x^{t-1}) - v_i^{t-1}\| \le \sqrt{4L\Delta} + \frac{3}{2}(B - \tau) + 2b + (L\gamma)^p a$ for all $i \in [n]$;

4. $\|v_i^t - g_i^{t-1}\| \le B$ for all $i \in [n]$;

5. $\gamma \le \frac{1}{12L}$;

6. $\|\theta_i^t\| \le b$ for all $i \in [n]$;

7. $\left\| \frac{1}{n} \sum_{l=1}^t \sum_{i=1}^n \omega_i^l \right\| \le a$;

8. $1 \ge \beta \ge 4L\gamma$;

9. $\Phi^{t-1} \le 2\Delta$.

Then we have

$$\|g^t\| \le \sqrt{64L\Delta} + 3(B - \tau) + 3b + 3a. \tag{30}$$

*Proof.* We start as follows

$$
\|g^t\| \overset{(i)}{=} \left\| g^{t-1} + \frac{1}{n}\sum_{i=1}^{n} \text{clip}_\tau(v_i^t - g_i^{t-1}) + \frac{1}{n}\sum_{i=1}^{n} \omega_i^t \right\|
$$

$$
= \left\| \frac{1}{n}\sum_{i=1}^{n} \left[ \nabla f_i(x^t) + (v_i^t - \nabla f_i(x^t)) + \text{clip}_\tau(v_i^t - g_i^{t-1}) - (v_i^t - g_i^{t-1}) \right] \right.
$$

$$
\left. + g^{t-1} - \frac{1}{n}\sum_{i=1}^{n} g_i^{t-1} + \frac{1}{n}\sum_{i=1}^{n} \omega_i^t \right\|
$$

$$
\overset{(ii)}{\le} \|\nabla f(x^t)\| + \frac{1}{n}\sum_{i=1}^{n} \|v_i^t - \nabla f_i(x^t)\| + \frac{1}{n}\sum_{i=1}^{n} \max\left\{0, \|v_i^t - g_i^{t-1}\| - \tau\right\}
$$

$$
+ \left\| g^{t-2} + \frac{1}{n}\sum_{i=1}^{n} \left[ \text{clip}_\tau(v_i^{t-1} - g_i^{t-2}) + \omega_i^{t-1} \right] - \frac{1}{n}\sum_{i=1}^{n} \left[ g_i^{t-2} + \text{clip}_\tau(v_i^{t-1} - g_i^{t-2}) \right] \right.
$$

$$
\left. + \frac{1}{n}\sum_{i=1}^{n} \omega_i^t \right\|
$$

where $(i)$ follows from the update rule of $g^t$; $(ii)$ from the triangle inequality and the properties of the clipping operator from Lemma 3. Cancelling terms inside the norm in the last term above we obtain

$$
\|g^t\| \overset{(i)}{=} \|\nabla f(x^t)\| + \frac{1}{n}\sum_{i=1}^{n} \|v_i^t - \nabla f_i(x^t)\| + \frac{1}{n}\sum_{i=1}^{n} \max\left\{0, \|v_i^t - g_i^{t-1}\| - \tau\right\}
$$

$$
+ \left\| g^{t-2} - \frac{1}{n}\sum_{i=1}^{n} g_i^{t-2} + \frac{1}{n}\sum_{l=t-1}^{t}\sum_{i=1}^{n} \omega_i^l \right\|
$$

$$
\overset{(ii)}{=} \|\nabla f(x^t)\| + \frac{1}{n}\sum_{i=1}^{n} \|v_i^t - \nabla f_i(x^t)\| + \frac{1}{n}\sum_{i=1}^{n} \max\left\{0, \|v_i^t - g_i^{t-1}\| - \tau\right\}
$$

$$
+ \left\| \frac{1}{n}\sum_{l=1}^{t}\sum_{i=1}^{n} \omega_i^l \right\|,
$$

where $(ii)$ follows from performing similar steps as in $(i)$ and having in mind assumption 1 from the statement of the lemma.

We continue to bound $\|g^t\|$ in the following way

$$\|g^t\| \overset{(i)}{\leq} \|\nabla f(x^{t-1})\| + \|\nabla f(x^t) - \nabla f(x^{t-1})\| + \frac{1}{n}\sum_{i=1}^n \|(1-\beta)v_i^{t-1} + \beta\nabla f_i(x^t, \xi_i^t) - \nabla f_i(x^t)\|$$

$$+ B - \tau + \left\|\frac{1}{n}\sum_{l=1}^t\sum_{i=1}^n \omega_i^l\right\|$$

$$\overset{(ii)}{\leq} \|\nabla f(x^{t-1})\| + \|\nabla f(x^t) - \nabla f(x^{t-1})\| + \frac{1}{n}\sum_{i=1}^n \|(1-\beta)v_i^{t-1} + \beta\nabla f_i(x^t) - \nabla f_i(x^t)\|$$

$$+ \frac{\beta}{n}\sum_{i=1}^n \|\theta_i^t\| + B - \tau + \left\|\frac{1}{n}\sum_{l=1}^t\sum_{i=1}^n \omega_i^l\right\|$$

$$\overset{(iii)}{\leq} \|\nabla f(x^{t-1})\| + \|\nabla f(x^t) - \nabla f(x^{t-1})\| + \frac{1}{n}\sum_{i=1}^n (1-\beta)\|v_i^{t-1} - \nabla f_i(x^{t-1})\|$$

$$+ \frac{1}{n}\sum_{i=1}^n (1-\beta)\|\nabla f_i(x^t) - \nabla f_i(x^{t-1})\| + \frac{\beta}{n}\sum_{i=1}^n \|\theta_i^t\| + B - \tau + \left\|\frac{1}{n}\sum_{l=1}^t\sum_{i=1}^n \omega_i^l\right\|,$$

where $(i)$ follows from triangle inequality, the update rule of $v_i^t$, and properties of the clipping operator from Lemma 3; $(ii)$ and $(iii)$ from triangle inequality. Using smoothness of $f$ we continue

$$\|g^t\| \overset{(i)}{\leq} \sqrt{2L(f(x^{t-1}) - f^*)} + L\gamma(2-\beta)\|g^{t-1}\| + (1-\beta)\frac{1}{n}\sum_{i=1}^n \|v_i^{t-1} - \nabla f_i(x^{t-1})\|$$

$$+ \frac{\beta}{n}\sum_{i=1}^n \|\theta_i^t\| + B - \tau + \frac{1}{n}\sum_{i=1}^n \|\omega_i^t\|$$

$$\leq \sqrt{2L\Phi^{t-1}} + 2L\gamma\|g^{t-1}\| + (1-\beta)\frac{1}{n}\sum_{i=1}^n \|v_i^{t-1} - \nabla f_i(x^{t-1})\|$$

$$+ \frac{\beta}{n}\sum_{i=1}^n \|\theta_i^t\| + B - \tau + \left\|\frac{1}{n}\sum_{l=1}^t\sum_{i=1}^n \omega_i^l\right\|$$

$$\overset{(ii)}{\leq} \sqrt{4L\Delta} + 2L\gamma\left(\sqrt{64L\Delta} + 3(B-\tau) + 3b + 3a\right)$$

$$+ (1-\beta)\left(\sqrt{4L\Delta} + \frac{3}{2}(B-\tau) + 2b + (L\gamma)^p a\right) + B - \tau + \beta b + a$$

$$\leq (16L\gamma + 4)\sqrt{L\Delta} + (6L\gamma + 1 + {}^3\!/\!_2)(B-\tau) + b(6L\gamma + 2(1-\beta) + \beta)$$

$$+ a(6L\gamma + (L\gamma)^p(1-\beta) + 1),$$

where $(i)$ follows from Lemma 4 and smoothness; from $(ii)$ from assumptions 2-4 in the statement of the lemma.

The claim of the lemma comes by noticing that since $\gamma \leq \frac{1}{12L} < \frac{1}{4L}$, then $16L\gamma + 4 \leq 8$. Moreover, $6L\gamma + 1 + {}^3\!/\!_2 \leq {}^1\!/\!_2 + 1 + {}^3\!/\!_2 = 3$. Next, we have that

$$6L\gamma + 2(1-\beta) + \beta \leq 3 \Leftrightarrow 6L\gamma \leq 1 + \beta,$$

which is satisfied if $12L\gamma \leq 1$, and

$$6L\gamma + (L\gamma)^p(1-\beta) + 1 \leq \frac{1}{2} + (^1\!/\!_{12})^p + 1 \leq \frac{1}{2} + 1 + 1 < 3,$$

where the last inequality holds for any $p \in [0.2, 0.8]$ since $\beta \leq 1$ and $L\gamma \leq {}^1\!/\!_{12}$. $\qquad\square$

**Lemma 14.** Let each $f_i$ is $L$-smooth and $\Delta \geq \Phi^0$. Assume the following inequalities hold

1. $\gamma \leq \frac{1}{12L}$;

2. $3(L\gamma)^{1-p} = \max\{4L\gamma, 3(L\gamma)^{1-p}\} \le \beta \le 1$[5];

3. $\|\nabla f_i(x^{t-1}) - v_i^{t-1}\| \le \sqrt{4L\Delta} + \frac{3}{2}(B - \tau) + 2b + (L\gamma)^p a$;

4. $\|\theta_i^t\| \le b$;

5. $\|g^{t-1}\| \le \sqrt{64L\Delta} + 3(B - \tau) + 3b + 3a$.

Then we have

$$\|\nabla f_i(x^t) - v_i^t\| \le \sqrt{4L\Delta} + \frac{3}{2}(B - \tau) + 2b + (L\gamma)^p a. \tag{31}$$

*Proof.* We have

$$\|\nabla f_i(x^t) - v_i^t\| \stackrel{(i)}{=} \|\nabla f_i(x^t) - (1 - \beta)v_i^{t-1} - \beta\nabla f_i(x^t, \xi_i^t)\|$$

$$\stackrel{(ii)}{\le} (1 - \beta)\|\nabla f_i(x^t) - v_i^{t-1}\| + \beta\|\nabla f_i(x^t) - \nabla f_i(x^t, \xi_i^t)\|$$

$$\stackrel{(iii)}{\le} (1 - \beta)L\gamma\|g^{t-1}\| + (1 - \beta)\|\nabla f_i(x^{t-1}) - v_i^{t-1}\| + \beta\|\theta_i^t\|$$

$$\stackrel{(iv)}{\le} (1 - \beta)L\gamma\left(\sqrt{64L\Delta} + 3(B - \tau) + 3b + 3a\right)$$

$$+ (1 - \beta)\left(\sqrt{4L\Delta} + \frac{3}{2}(B - \tau) + 2b + (L\gamma)^p a\right) + \beta b$$

$$= (8L\gamma + 2(1 - \beta))\sqrt{L\Delta} + (3L\gamma + {}^{3(1-\beta)}/{}_2)(B - \tau)$$

$$+ (3L\gamma + 2(1 - \beta) + \beta)b + (3L\gamma + (L\gamma)^p(1 - \beta))a,$$

where $(i)$ follows from the update rule of $v_i^t$; $(ii)$ from the triangle inequality; $(iii)$ from triangle inequality, smoothness, and the update rule of $x^t$; $(iv)$ from assumptions 2-4 of the lemma. Since $\beta = 6L\gamma$, then

$$8L\gamma + 2(1 - \beta) \le 2 \Leftrightarrow 4L\gamma \le \beta,$$

$$3L\gamma + \frac{3}{2}(1 - \beta) \le \frac{3}{2} \Leftrightarrow 2L\gamma \le \beta,$$

$$3L\gamma + 2(1 - \beta) + \beta \le 2 \Leftrightarrow 3L\gamma \le \beta,$$

$$3L\gamma + (L\gamma)^p(1 - \beta) \le (L\gamma)^p \Leftrightarrow 3L\gamma \le (L\gamma)^p\beta \Leftrightarrow 3(L\gamma)^{1-p} \le \beta,$$

where the last inequalities in each line hold by the choice of $\beta$. $\qquad\square$

**Lemma 15.** Let each $f_i$ be $L$-smooth, $\Delta \ge \Phi^0$, and $i \in \mathcal{I}_t$. Let the following inequalities hold

1. $12L\gamma \le 1$;

2. $1 \ge \beta \ge \max\{4L\gamma, 3(L\gamma)^{1-p}\} = 3(L\gamma)^{1-p}$;

3. $\beta \le \frac{\tau}{32\sqrt{L\Delta}}$;

4. $\beta \le \frac{\tau}{18(B-\tau)}$;

5. $\beta \le \frac{\tau}{30b}$;

6. $\beta \le \left(\frac{\tau}{48a}\right)^{1-p}$;

7. $\|g^t\| \le \sqrt{64L\Delta} + 3(B - \tau) + 3b + 3a$;

8. $\|\theta_i^{t+1}\| \le b$;

9. $\|\nabla f_i(x^t) - v_i^t\| \le \sqrt{4L\Delta} + \frac{3}{2}(B - \tau) + 2b + (L\gamma)^p a$.

---

[5]For $p \in [1/5, 0.8]$ we have $3x^{1-p} \ge 4x$ for any $x \in [0, 1/12]$.

Then

$$\|v_i^{t+1} - g_i^t\| \le \|v_i^t - g_i^{t-1}\| - \frac{\tau}{2}. \tag{32}$$

*Proof.* Since $i \in \mathcal{I}_t$, then $\|v_i^t - g_i^{t-1}\| > \tau$, thus from Lemma 12 we have

$$
\begin{aligned}
\|v_i^{t+1} - g_i^t\| &\le \|v_i^t - g_i^{t-1}\| - \tau + \beta L\gamma\|g^t\| + \beta\|\nabla f_i(x^t) - v_i^t\| + \beta\|\theta_i^{t+1}\| \\
&\overset{(i)}{\le} \|v_i^t - g_i^{t-1}\| - \tau + L\gamma\left(\sqrt{64L\Delta} + 3(B - \tau) + 3b + 3a\right) \\
&\quad + \beta\left(\sqrt{4L\Delta} + \frac{3}{2}(B - \tau) + 2b + (L\gamma)^p a\right) + \beta b \\
&= \|v_i^t - g_i^{t-1}\| - \tau + (8L\gamma + 2\beta)\sqrt{L\Delta} + (3L\gamma + {}^{3\beta}/_2)(B - \tau) + (3L\gamma + 2\beta)b \\
&\quad + (3L\gamma + (L\gamma)^p\beta)a,
\end{aligned}
$$

where $(i)$ follows from assumptions 7-9 of the lemma. Since $4L\gamma \le \beta$ we have

$$(8L\gamma + 2\beta)\sqrt{L\Delta} \le 4\beta\sqrt{L\Delta} \le \frac{\tau}{8},$$

where $\beta \le \frac{\tau}{32\sqrt{L\Delta}}$. Since $4L\gamma \le \beta$ we have

$$\left(3L\gamma + \frac{3\beta}{2}\right)(B - \tau) \le \frac{9}{4}\beta(B - \tau) \le \frac{\tau}{8},$$

where $\beta \le \frac{\tau}{18(B-\tau)}$. Since $4L\gamma \le \beta$ we have

$$(3L\gamma + 3\beta)\,b \le \frac{15}{4}\beta b \le \frac{\tau}{8},$$

where $\beta \le \frac{\tau}{30b}$. Since $3(L\gamma)^{1-p} \le \beta$ we have

$$(3L\gamma + (L\gamma)^p\beta)\,a \le 6(^{\beta}/_3)^{\frac{1}{1-p}}a \le \frac{\tau}{8},$$

where $\beta \le \left(\frac{\tau}{48a}\right)^{1-p}$. Thus we have

$$\|v_i^{t+1} - g_i^t\| \le \|v_i^t - g_i^{t-1}\| - \tau + 4 \cdot \frac{\tau}{8} = \|v_i^t - g_i^{t-1}\| - \frac{\tau}{2}.$$

$\square$

**Lemma 16.** Let $\|\theta_i^{t+1}\| \le b$ for all $i \in [n]$. Let each $f_i$ be $L$-smooth. Then $\widetilde{P}^t$ decreases as

$$\widetilde{P}^{t+1} \le (1 - \beta)\widetilde{P}^t + \frac{3L^2}{\beta}R^t + \beta^2 b^2 + \frac{2}{n}\beta(1 - \beta)\sum_{i=1}^n \langle v_i^t - \nabla f_i(x^{t+1}), \theta_i^{t+1}\rangle. \tag{33}$$

*Proof.* We have

$$
\begin{aligned}
\|v_i^{t+1} - \nabla f_i(x^{t+1})\|^2 &\overset{(i)}{=} \|(1 - \beta)v_i^t + \beta\nabla f_i(x^{t+1}, \xi_i^{t+1}) - \nabla f_i(x^{t+1})\|^2 \\
&= \|(1 - \beta)(v_i^t - \nabla f_i(x^{t+1})) + \beta(\nabla f_i(x^{t+1}, \xi_i^{t+1}) - \nabla f_i(x^{t+1}))\|^2 \\
&= (1 - \beta)^2\|v_i^t - \nabla f_i(x^{t+1})\|^2 + \beta^2\|\theta_i^{t+1}\|^2 \\
&\quad + 2\beta(1 - \beta)\langle v_i^t - \nabla f_i(x^{t+1}), \theta_i^{t+1}\rangle \\
&\overset{(ii)}{\le} (1 - \beta)^2(1 + {}^{\beta}/_2)\|v_i^t - \nabla f_i(x^t)\|^2 \\
&\quad + (1 - \beta)^2(1 + {}^2/_\beta)\|\nabla f_i(x^t) - \nabla f_i(x^{t+1})\|^2 + \beta^2 b^2 \\
&\quad + 2\beta(1 - \beta)\langle v_i^t - \nabla f_i(x^{t+1}), \theta_i^{t+1}\rangle \\
&\overset{(iii)}{\le} (1 - \beta)\|v_i^t - \nabla f_i(x^t)\|^2 + \frac{3L^2}{\beta}\|x^t - x^{t+1}\|^2 + \beta^2 b^2 \\
&\quad + 2\beta(1 - \beta)\langle v_i^t - \nabla f_i(x^{t+1}), \theta_i^{t+1}\rangle,
\end{aligned}
$$

where $(i)$ follows from the update rule of $v_i^t$; $(ii)$ from $\|x+y\|^2 \le (1+r)\|x\|^2 + (1+r^{-1})\|y\|^2$ for any $x, y \in \mathbb{R}^d$ and $r > 0$; $(iii)$ from the smoothness and inequalities $(1-\beta)^2(1+\beta/2) \le (1-\beta)$ and $(1-\beta)^2(1+2/\beta) \le 3/\beta$. Averaging the inequalities above across all $i \in [n]$ we get the lemma statement. $\qquad\square$

Similarly, we can get the descent of $P^t$.

**Lemma 17.** Let $\|\theta^{t+1}\| \le \frac{c}{\sqrt{n}}$, and each $f_i$ be $L$-smooth. Then $P^t$ decreases as

$$P^{t+1} \le (1-\beta)P^t + \frac{3L^2}{\beta}R^t + \beta^2 b^2 + 2\beta(1-\beta)\langle v^t - \nabla f(x^{t+1}), \theta^{t+1}\rangle.$$

*Proof.* For shortness, we denote $\nabla f(x^t, \xi^t) := \frac{1}{n}\sum_{i=1}^n \nabla f(x^t, \xi_i^t)$ and $\theta^t := \frac{1}{n}\sum_{i=1}^n \nabla f_i(x^t, \xi^t) - \nabla f_i(x^t)$. Then we have

$$\begin{aligned}
\|v^{t+1} - \nabla f(x^t)\|^2 &\overset{(i)}{=} \|(1-\beta)v^t + \beta\nabla f(x^{t+1}, \xi^{t+1}) - \nabla f(x^{t+1})\|^2 \\
&= \|(1-\beta)(v_i^t - \nabla f_i(x^{t+1})) + \beta(\nabla f(x^{t+1}, \xi^{t+1}) - \nabla f(x^{t+1}))\|^2 \\
&= (1-\beta)^2\|v^t - \nabla f(x^{t+1})\|^2 + \beta^2\|\theta^{t+1}\|^2 \\
&\quad + 2\beta(1-\beta)\langle v^t - \nabla f(x^{t+1}), \theta^{t+1}\rangle \\
&\overset{(ii)}{\le} (1-\beta)^2(1+\beta/2)\|v^t - \nabla f(x^t)\|^2 \\
&\quad + (1-\beta)^2(1+2/\beta)\|\nabla f_i(x^t) - \nabla f_i(x^{t+1})\|^2 + \beta^2\frac{c^2}{n} \\
&\quad + 2\beta(1-\beta)\langle v^t - \nabla f(x^{t+1}), \theta_i^{t+1}\rangle \\
&\overset{(iii)}{\le} (1-\beta)\|v^t - \nabla f(x^t)\|^2 + \frac{3L^2}{\beta}\|x^t - x^{t+1}\|^2 + \beta^2\frac{c^2}{n} \\
&\quad + 2\beta(1-\beta)\langle v^t - \nabla f(x^{t+1}), \theta^{t+1}\rangle,
\end{aligned}$$

where $(i)$ follows from the update rule of $v_i^t$; $(ii)$ from $\|x+y\|^2 \le (1+r)\|x\|^2 + (1+r^{-1})\|y\|^2$ for any $x, y \in \mathbb{R}^d$ and $r > 0$; $(iii)$ from the smoothness and inequalities $(1-\beta)^2(1+\beta/2) \le (1-\beta)$ and $(1-\beta)^2(1+2/\beta) \le 3/\beta$. $\qquad\square$

Now we present the descent of $\widetilde{V}^t$.

**Lemma 18.** Let $\|\theta_i^t\| \le b$ for all $i \in [n]$, each $f_i$ be $L$-smooth, and $\|v_i^t - g_i^{t-1}\| \le B$ for all $i \in [n]$. Then

$$\begin{aligned}
\|g_i^t - v_i^t\|^2 &\le (1-\eta)\|g_i^{t-1} - v_i^{t-1}\|^2 + \frac{4\beta^2}{\eta}\|v_i^{t-1} - \nabla f_i(x^{t-1})\|^2 + \frac{4\beta^2 L^2}{\eta}R^{t-1} + \beta^2 b^2 \quad (34) \\
&\quad + 2(1-\eta)^2\beta\langle (g_i^{t-1} - v_i^{t-1}) + \beta(v_i^{t-1} - \nabla f_i(x^{t-1})), \theta_i^t\rangle \\
&\quad + 2(1-\eta)^2\beta\langle \beta(\nabla f_i(x^{t-1}) - \nabla f_i(x^t)), \theta_i^t\rangle.
\end{aligned}$$

Moreover, averaging the inequalities across all $i \in [n]$ we get

$$\widetilde{V}^t \le (1-\eta)\widetilde{V}^{t-1} + \frac{4\beta^2}{\eta}\widetilde{P}^{t-1} + \frac{4\beta^2 L^2}{\eta}R^{t-1} + \beta^2 b^2 \quad (35)$$

$$+ \frac{2}{n}(1-\eta)^2\beta\sum_{i=1}^n \langle (g_i^{t-1} - v_i^{t-1}) + \beta(v_i^{t-1} - \nabla f_i(x^{t-1})) + \beta(\nabla f_i(x^{t-1}) - \nabla f_i(x^t)), \theta_i^t\rangle.$$

*Proof.* Since $\|v_i^t - g_i^{t-1}\| \le B$, we have $\eta_i^t \ge \eta \in (0, 1)$. Thus, we have

$$\begin{aligned}
\|g_i^t - v_i^t\|^2 &\overset{(i)}{=} \|g_i^{t-1} + \text{clip}_\tau(v_i^t - g_i^{t-1}) - v_i^t\|^2 = \|g_i^{t-1} - v_i^t - (v_i^t - g_i^{t-1})\cdot \tau/\|v_i^t - g_i^{t-1}\|\| \\
&\le (1-\eta_i^t)^2\|g_i^{t-1} - v_i^t\|^2 \le (1-\eta)^2\|g_i^{t-1} - v_i^t\|^2,
\end{aligned}$$

where $(i)$ follows from the update rule of $g_i^t$. We can rewrite RHS in the above inequality as follows using the update rule of $v_i^t$

$$\|g_i^t - v_i^t\|^2 \leq (1-\eta)^2\|g_i^{t-1} - (1-\beta)v_i^{t-1} - \beta\nabla f_i(x^t, \xi_i^t)\|^2$$
$$= (1-\eta)^2\|g_i^{t-1} - (1-\beta)v_i^{t-1} - \beta\nabla f_i(x^t) - \beta\theta_i^t\|^2$$
$$= (1-\eta)^2\|g_i^{t-1} - (1-\beta)v_i^{t-1} - \beta\nabla f_i(x^t)\|^2 + (1-\eta)^2\beta^2\|\theta_i^t\|^2$$
$$+ 2(1-\eta)^2\beta\langle g_i^{t-1} - (1-\beta)v_i^{t-1} - \beta\nabla f_i(x^t), \theta_i^t\rangle$$

Therefore, we have

$$\|g_i^t - v_i^t\|^2 \overset{(i)}{\leq} (1-\eta)^2\|g_i^{t-1} - (1-\beta)v_i^{t-1} - \beta\nabla f_i(x^t)\|^2 + \beta^2 b^2$$
$$+ 2(1-\eta)^2\beta\langle g_i^{t-1} - (1-\beta)v_i^{t-1} - \beta\nabla f_i(x^t), \theta_i^t\rangle$$
$$\overset{(ii)}{\leq} (1-\eta)^2(1+\rho)\|g_i^{t-1} - v_i^{t-1}\|^2 + (1-\eta)^2(1+\rho^{-1})\beta^2\|v_i^{t-1} - \nabla f_i(x^t)\|^2$$
$$+ \beta^2 b^2 + 2(1-\eta)^2\beta\langle g_i^{t-1} - (1-\beta)v_i^{t-1} - \beta\nabla f_i(x^t), \theta_i^t\rangle$$
$$\overset{(iii)}{\leq} (1-\eta)^2(1+\rho)\|g_i^{t-1} - v_i^{t-1}\|^2 + 2(1-\eta)^2(1+\rho^{-1})\beta^2\|v_i^{t-1} - \nabla f_i(x^{t-1})\|^2$$
$$+ 2(1-\eta)^2(1+\rho^{-1})\beta^2 L^2\|x^{t-1} - x^t\|^2 + \beta^2 b^2$$
$$+ 2(1-\eta)^2\beta\langle(g_i^{t-1} - v_i^{t-1}) + \beta(v_i^{t-1} - \nabla f_i(x^{t-1})), \theta_i^t\rangle$$
$$+ 2(1-\eta)^2\beta\langle\beta(\nabla f_i(x^{t-1}) - \nabla f_i(x^t)), \theta_i^t\rangle,$$

where $(i)$ follows from the assumption of the lemma; $(ii)$ from the inequality $\|x + y\|^2 \leq (1+r)\|x\|^2 + (1+r^{-1})\|y\|^2$ for any $x, y \in \mathbb{R}^d$ and $r > 0$; from $\|x + y\|^2 \leq 2\|x\|^2 + 2\|y\|^2$ for any $x, y \in \mathbb{R}^d$ and smoothness.

If we choose $\rho = \eta/2$, we get the final bound

$$\|g_i^t - v_i^t\|^2 \leq (1-\eta)\|g_i^{t-1} - v_i^{t-1}\|^2 + \frac{4\beta^2}{\eta}\|v_i^{t-1} - \nabla f_i(x^{t-1})\|^2$$
$$+ \frac{4\beta^2 L^2}{\eta}R^{t-1} + \beta^2 b^2 + 2(1-\eta)^2\beta\langle(g_i^{t-1} - v_i^{t-1}) + \beta(v_i^{t-1} - \nabla f_i(x^{t-1})), \theta_i^t\rangle$$
$$+ 2(1-\eta)^2\beta\langle\beta(\nabla f_i(x^{t-1}) - \nabla f_i(x^t)), \theta_i^t\rangle$$

$\square$

**Theorem 6** (Full statement of Theorem 4). *Let Assumptions 1 and 2 hold, $B := \max_i\{\|\nabla f_i(x^0)\|\} + b > \tau$, probability constant $\alpha \in (0, 1)$, constants $a, b,$ and $c$ be defined as in (28), $p = 0.8$, and $\Delta \geq \Phi^0$. Let us run Algorithm 3 for $T$ iterations with DP noise with variance $\sigma_\omega$. Assume the following inequalities hold*

    *1.* **stepsize restrictions:**

        *i)* $12L\gamma \leq 1$;

        *ii)*

$$\frac{2}{3} - \frac{16\beta^2 L^2}{\eta^2}\gamma^2 - \frac{48L^2}{\eta^2}\gamma^2 \geq 0;$$

    *2.* **momentum restrictions:**

        *i)* $1 \geq \beta \geq \max\{4L\gamma, 3L\gamma)^{1-p}\} = 3(L\gamma)^{1-p}$;

        *ii)* $\beta \leq \frac{\tau}{32\sqrt{L\Delta}}$;

        *iii)* $\beta \leq \frac{\tau}{18(B-\tau)}$;

        *iv)* $\beta \leq \frac{\tau}{30b}$;

        *v)* $\beta \leq \left(\frac{3\tau}{40a}\right)^{1-p}$;

*vi) and momentum restrictions defined in (38), (39), (40), (41), (42), (44), (43), and (45);*

*Then with probability $1 - \alpha$ we have*

$$\frac{1}{T} \sum_{t=0}^{T-1} \|\nabla f(x^t)\|^2 \leq \widetilde{\mathcal{O}} \left( \frac{L^2 \Delta \sqrt{d} \sigma_\omega}{\sqrt{T} n \tau} + \frac{(L\Delta)^{\frac{1}{6}} \sigma^{\frac{5}{3}}}{T^{\frac{1}{6}} n^{\frac{5}{6}}} + \frac{(L\Delta)^{\frac{4}{9}} \sigma^{\frac{5}{9}} d^{\frac{5}{18}} \sigma_\omega^{\frac{5}{9}}}{T^{\frac{4}{9}} n^{\frac{5}{9}}} \right),$$

*where $\widetilde{\mathcal{O}}$ hides constant and logarithmic factors, and higher order terms decreasing in $T$.*

*Proof.* We prove the main theorem by induction. The conventional choice $\nabla f_i(x^{-1}, \xi_i^{-1}) = v_i^{-1} = g_i^{-1} = 0, \Phi^{-1} = \Phi^0$.

Let us define an event $E_t$ for each $t \in \{0, \ldots, T\}$ such that the following inequalities hold for all $k \in \{0, \ldots, t\}$

1. $\|v_i^k - g_i^{k-1}\| \leq B$ for $i \in \mathcal{I}_k$;

2. $\|g^k\| \leq \sqrt{64 L \Delta} + 3(B - \tau) + 3b + 3a$;

3. $\|v_i^k - \nabla f_i(x^k)\| \leq \sqrt{4 L \Delta} + \frac{3}{2}(B - \tau) + 2b + (L\gamma)^p a$;

4. $\|\theta_i^k\| \leq b$ for all $i \in [n]$ and $\|\theta^k\| \leq \frac{c}{\sqrt{n}}$;

5. $\left\| \frac{1}{n} \sum_{l=1}^{k+1} \sum_{i=1}^{n} \omega_i^l \right\| \leq a$;

6. $\Phi^k \leq 2\Delta$;

7.

$$\Delta \geq \frac{2\gamma\beta}{n\eta}(1-\eta)^2 \sum_{l=0}^{k-1} \sum_{i=1}^{n} \langle (g_i^l - v_i^l) + \beta(v_i^l - \nabla f_i(x^l)) + \beta(\nabla f_i(x^l) - \nabla f_i(x^{l+1})), \theta_i^t \rangle$$

$$+ \frac{8\gamma\beta^2}{n\eta^2}(1-\beta) \sum_{l=0}^{k-1} \sum_{i=1}^{n} \langle v_i^l - \nabla f_i(x^l), \theta_i^{l+1} \rangle + 2\gamma(1-\beta) \sum_{l=0}^{k-1} \langle v^l - \nabla f(x^l), \theta^{l+1} \rangle$$

$$+ \frac{8\gamma\beta^2}{n\eta^2}(1-\beta) \sum_{l=0}^{k-1} \sum_{i=1}^{n} \langle \nabla f_i(x^l) - \nabla f_i(x^{l+1}), \theta_i^{l+1} \rangle$$

$$+ 2\gamma(1-\beta) \sum_{l=0}^{k-1} \langle \nabla f(x^l) - \nabla f(x^{l+1}), \theta^{l+1} \rangle.$$

Denote the events $\Theta_i^t, \Theta^t$ and $N^{t+1}$ as

$$\Theta_i^t := \{\|\theta_i^t\| \geq b\}, \quad \Theta^t := \{\|\theta^t\| \geq \frac{c}{\sqrt{n}}\}, \quad \text{and} \quad N^{t+1} := \left\{ \left\| \frac{1}{n} \sum_{l=1}^{t} \sum_{i=1}^{n} \omega_i^l \right\| \geq a \right\} \quad (36)$$

respectively. From Assumption 2 we have

$$\Pr(\Theta_i^t) \leq 2 \exp\left(-\frac{b^2}{2\sigma^2}\right) = \frac{\alpha}{6(T+1)n}$$

where the last equality is by definition of $b^2$. Therefore, $\Pr(\overline{\Theta}_i^t) \geq 1 - \frac{\alpha}{6(T+1)n}$.

Besides, notice that the constant $c$ in (28) can be viewed as

$$c = (\sqrt{2} + 2b_3)\sigma \quad \text{where} \quad b_3^2 = 3 \log \frac{6(T+1)}{\alpha}.$$

Now we can use Lemma 1 to bound $\Pr(\Theta^t)$. Since all $\theta_i^t$ are independent $\sigma$-sub-Gaussian random vectors, then we have

$$\Pr\left(\left\|\sum_{i=1}^n \theta_i^t\right\| \geq c\sqrt{n}\right) = \Pr\left(\|\theta^t\| \geq \frac{c}{\sqrt{n}}\right) \leq \exp(-b_3^2/3) = \frac{\alpha}{6(T+1)}.$$

We also use Lemma 1 to bound $\Pr(N^t)$. Indeed, since all $\omega_i^l$ are independent Gaussian random vectors, then we have

$$\Pr\left(\left\|\sum_{l=1}^t \sum_{i=1}^n \omega_i^l\right\| \geq (\sqrt{2}+2b_2)\sqrt{\sum_{l=1}^t \sum_{i=1}^n \sigma_\omega^2 d}\right) \leq \exp(-b_2^2/3) = \frac{\alpha}{6(T+1)}.$$

with $b_2^2 = 3\log\left(\frac{4(T+1)}{\alpha}\right)$.

This implies that

$$\Pr\left(\left\|\frac{1}{n}\sum_{l=1}^t \sum_{i=1}^n \omega_i^l\right\| \geq a\right) \leq \frac{\alpha}{6(T+1)}$$

due to the choice of $a$ from (28):

$$a = (\sqrt{2}+2b_2)\sigma_\omega\sqrt{d}\sqrt{T/n} \quad \text{where} \quad b_2^2 = 3\log\frac{6(T+1)}{\alpha}.$$

Note that with this choice of $a$ we have that the above is true for any $t \in \{1, T\}$, i.e. $\Pr(N^t) \geq 1 - \frac{\alpha}{6(T+1)}$ for all $t \in \{1, T\}$.

Now we prove that $\Pr(E_t) \geq 1 - \frac{\alpha(t+1)}{T+1}$ for all $t \in \{0, \ldots, T-1\}$. First, we show that the base of induction holds.

**Base of induction.**

1. $\|v_i^0 - g_i^{-1}\| = \|v_i^0\| = \beta\|\nabla f_i(x^0, \xi_i^0)\| = \beta\|\theta_i^0\| + \beta\|\nabla f_i(x^0)\| \leq \frac{1}{2}b + \frac{1}{2}B \leq \frac{1}{2}B + \frac{1}{2}B = B$ holds with probability $1 - \frac{\alpha}{6(T+1)}$. Indeed, we have

$$\Pr(\Theta_i^0) \leq 2\exp\left(-\frac{b^2}{2\sigma^2}\right) = \frac{\alpha}{6(T+1)n}.$$

Therefore, we have

$$\Pr\left(\cap_{i=1}^n \overline{\Theta}_i^0\right) = 1 - \Pr\left(\cup_{i=1}^n \Theta_i^0\right) \geq 1 - \sum_{i=1}^n \Pr(\Theta_i^0) = 1 - n\frac{\alpha}{6(T+1)n} = 1 - \frac{\alpha}{6(T+1)}.$$

Moreover, by concentration inequality we have

$$\Pr(\Theta^0) \leq \frac{\alpha}{6(T+1)}.$$

This means that the probability of the event that each $\left\|\frac{1}{n}\sum_{l=1}^1 \sum_{i=1}^n \omega_i^l\right\| \leq a$, $\|\theta_i^0\| \leq b$, and $\|\theta^0\| \leq \frac{c}{\sqrt{n}}$, and is at least

$$1 - \frac{\alpha}{6(T+1)} - n\frac{\alpha}{6n(T+1)} - \frac{\alpha}{6(T+1)} = 1 - \frac{\alpha}{2(T+1)}.$$

2. We have already shown that

$$\Pr\left(\left\|\frac{1}{n}\sum_{i=1}^n \omega_i^1\right\| \geq a\right) \leq \frac{\alpha}{6(T+1)},$$

implying that $\left\|\frac{1}{n}\sum_{i=1}^n \omega_i^1\right\| \leq a$ with probability at least $1 - \frac{\alpha}{6(T+1)}$.

3. $g^0 = \frac{1}{n}\sum_{i=1}^{n}(g_i^{-1} + \text{clip}_\tau(v_i^0 - g_i^{-1})) = \frac{1}{n}\sum_{i=1}^{n}\text{clip}_\tau(\beta\nabla f_i(x^0, \xi_i^0))$. Therefore, we have

$$\|g^0\| \le \left\| \frac{1}{n}\sum_{i=1}^{n}\beta\nabla f_i(x^0) + \beta\theta_i^0 + (\text{clip}_\tau(\beta\nabla f_i(x^0, \xi_i^0)) - \beta\nabla f_i(x^0, \xi_i^0)) \right\|$$

$$\le \beta\|\nabla f(x^0)\| + \frac{\beta}{n}\sum_{i=1}^{n}\|\theta_i^0\| + \frac{1}{n}\sum_{i=1}^{n}\max\left\{0, \beta\|\nabla f_i(x^0, \xi_i^0)\| - \tau\right\}$$

$$\le \beta\sqrt{2L(f(x^0) - f(x^*))} + \frac{\beta}{n}\sum_{i=1}^{n}\|\theta_i^0\| + \frac{1}{n}\sum_{i=1}^{n}\max\left\{0, \beta\|\nabla f_i(x^0)\| + \beta\|\theta_i^0\| - \tau\right\}$$

$$\le \frac{1}{2}\sqrt{2L\Phi^0} + \frac{2\beta}{n}\sum_{i=1}^{n}\|\theta_i^0\| + \frac{\beta}{n}\sum_{i=1}^{n}\|\nabla f_i(x^0)\| - \tau$$

$$\le \sqrt{64L\Delta} + 2\beta b + \beta B - \tau$$

$$\le \sqrt{64L\Delta} + \frac{3}{2}B - \tau + b \le \sqrt{64L\Delta} + 3(B - \tau) + 2b + (L\gamma)^p a.$$

The inequalities above again hold in $\cap_{i=1}^{n}\overline{\Theta}_i^0$, i.e. with probability at least $1 - \frac{\alpha}{6(T+1)}$.

4. We have
$$\|v_i^0 - \nabla f_i(x^0)\| = \|\nabla f_i(x^0, \xi_i^0) - \nabla f_i(x^0)\| \le b.$$

This bound holds with probability at least $1 - \frac{\alpha}{6(T+1)}$ because it holds in $\cap_{i=1}^{n}\overline{\Theta}_i^0$.

5. Inequalities 5 obviously also hold, as $\Phi^0 \le 2\Phi^0 \le 2\Delta$ by the choice of $\Delta$.

Therefore, we conclude that the inequalities 1-7 hold with a probability at least

$$\Pr\left(\Theta^0 \cap \left(\cap_{i=1}^{n}\overline{\Theta}_i^0\right) \cap \overline{N}^t\right) \ge 1 - \Pr(\Theta^0) - \sum_{i=1}^{n}\Pr(\Theta_i^0) - \Pr(N^0)$$

$$\ge 1 - \frac{\alpha}{6(T+1)} - n\cdot\frac{\alpha}{6n(T+1)} - \frac{\alpha}{6(T+1)}$$

$$= 1 - \frac{\alpha}{2(T+1)} > 1 - \frac{\alpha}{T+1},$$

i.e. $\Pr(E_0) \ge 1 - \frac{\alpha}{T+1}$ holds. This is the base of the induction.

**Transition step of induction.**

CASE $|\mathcal{I}_{K+1}| > 0$. Assume that all events $\overline{\Theta}^{K+1}, \overline{\Theta}_i^{K+1}$ and $\overline{N}^{K+1}$ take place, i.e. $\|\theta_i^{K+1}\| \le b, \|\theta^{K+1}\| \le \frac{c}{\sqrt{n}}$ for all $i \in [n]$ and $\left\|\frac{1}{n}\sum_{l=1}^{t+1}\sum_{i=1}^{n}\omega_i^l\right\| \le a$. For that we need to work in $\overline{\Theta}^{K+1} \cap \left(\cap_{i=1}^{n}\overline{\Theta}_i^{K+1}\right) \cap \overline{N}^{K+1}$. Then, by the assumptions of the induction, from Lemma 15 we get for all $i \in \mathcal{I}_{K+1}$
$$\|v_i^{K+1} - g_i^K\| \le \|v_i^K - g_i^{K-1}\| - \frac{\tau}{2} \le B - \frac{\tau}{2}.$$

Therefore, from Lemma 13 we get that
$$\|g^{K+1}\| \le \sqrt{64L\Delta} + 3(B - \tau) + 3b + 3a,$$

and from Lemma 14
$$\|\nabla f_i(x^{K+1}) - v_i^{K+1}\| \le \sqrt{4L\Delta} + \frac{3}{2}(B - \tau) + 2b + (L\gamma)^p a.$$

This means that 1-5 in the induction assumption are also verified for the step $K + 1$.

Since we have for all $t \in \{0, \ldots, K+1\}$ that inequalities 1-5 are verified, then we can write for each $t \in \{0, K\}$ by Lemmas 2 and 16 to 18 the following

$$
\begin{aligned}
\Phi^{t+1} &= \delta^{t+1} + \frac{\gamma}{\eta}\widetilde{V}^{t+1} + \frac{4\gamma\beta}{\eta^2}\widetilde{P}^{t+1} + \frac{\gamma}{\beta}P^{t+1} \\
&\leq \delta^t - \frac{\gamma}{2}\|\nabla f(x^t)\|^2 - \frac{1}{4\gamma}R^t + \gamma\widetilde{V}^t + \gamma P^t \\
&\quad + \frac{\gamma}{\eta}\left((1-\eta)\widetilde{V}^t + \frac{4\beta^2}{\eta}\widetilde{P}^t + \frac{4\beta^2 L^2}{\eta}R^t + \beta^2 b^2\right. \\
&\qquad \left. + \frac{2}{n}\beta(1-\eta^2)\sum_{i=1}^{n}\langle(g_i^t - v_i^t) + \beta(v_i^t - \nabla f_i(x^t)) + \beta(\nabla f_i(x^t) - \nabla f_i(x^{t+1})), \theta_i^{t+1}\rangle\right) \\
&\quad + \frac{4\gamma\beta}{\eta^2}\left((1-\beta)\widetilde{P}^t + \frac{3L^2}{\beta}R^t + \beta^2 b^2 + \frac{2}{n}\beta(1-\beta)\sum_{i=1}^{n}\langle v_i^t - \nabla f_i(x^{t+1}), \theta_i^{t+1}\rangle\right) \\
&\quad + \frac{\gamma}{\beta}\left((1-\beta)P^t + \frac{3L^2}{\beta}R^t + \beta^2\frac{c^2}{n} + 2\beta(1-\beta)\langle v^t - \nabla f(x^{t+1}), \theta^{t+1}\rangle\right)
\end{aligned}
$$

Rearranging terms we get

$$
\begin{aligned}
\Phi^{t+1} &\leq \delta^t - \frac{\gamma}{2}\|\nabla f(x^t)\|^2 + \frac{\gamma}{\eta}\widetilde{V}^t(\eta + 1 - \eta) + \frac{4\gamma\beta}{\eta^2}\widetilde{P}^t(\beta + 1 - \beta) + \frac{\gamma}{\beta}P^t(\beta + 1 - \beta) \\
&\quad - \frac{1}{4\gamma}R^t\left(1 - \frac{16L^2\beta^2}{\eta^2}\gamma^2 - \frac{48L^2}{\eta^2}\gamma^2 - \frac{12L^2}{\beta^2}\gamma^2\right) + b^2\left(\frac{\beta^2\gamma}{\eta} + \frac{4\gamma\beta^3}{\eta^2}\right) + c^2\frac{\gamma\beta}{n} \\
&\quad + \frac{2\gamma\beta}{n\eta}(1-\eta)^2\sum_{i=1}^{n}\langle(g_i^t - v_i^t) + \beta(v_i^t - \nabla f_i(x^t)) + \beta(\nabla f_i(x^t) - \nabla f_i(x^{t+1})), \theta_i^{t+1}\rangle \\
&\quad + \frac{8\gamma\beta^2}{n\eta^2}(1-\beta)\sum_{i=1}^{n}\langle v_i^t - \nabla f_i(x^t), \theta_i^{t+1}\rangle + 2\gamma(1-\beta)\langle v^t - \nabla f(x^t), \theta^{t+1}\rangle \\
&\quad + \frac{8\gamma\beta^2}{n\eta^2}(1-\beta)\sum_{i=1}^{n}\langle\nabla f_i(x^t) - \nabla f_i(x^{t+1}), \theta_i^{t+1}\rangle \\
&\quad + 2\gamma(1-\beta)\langle\nabla f(x^t) - \nabla f(x^{t+1}), \theta^{t+1}\rangle.
\end{aligned}
$$

Using stepsize restriction $(vi)$ we get rid of the term with $R^t$ and obtain

$$
\begin{aligned}
\Phi^{t+1} &\leq \Phi^t - \frac{\gamma}{2}\|\nabla f(x^t)\|^2 + b^2\left(\frac{\beta^2\gamma}{\eta} + \frac{4\gamma\beta^3}{\eta^2}\right) + c^2\frac{\gamma\beta}{n} \\
&\quad + \frac{2\gamma\beta}{n\eta}(1-\eta)^2\sum_{i=1}^{n}\langle(g_i^t - v_i^t) + \beta(v_i^t - \nabla f_i(x^t)) + \beta(\nabla f_i(x^t) - \nabla f_i(x^{t+1})), \theta_i^{t+1}\rangle \\
&\quad + \frac{8\gamma\beta^2}{n\eta^2}(1-\beta)\sum_{i=1}^{n}\langle v_i^t - \nabla f_i(x^t), \theta_i^{t+1}\rangle + 2\gamma(1-\beta)\langle v^t - \nabla f(x^t), \theta^{t+1}\rangle \\
&\quad + \frac{8\gamma\beta^2}{n\eta^2}(1-\beta)\sum_{i=1}^{n}\langle\nabla f_i(x^t) - \nabla f_i(x^{t+1}), \theta_i^{t+1}\rangle \\
&\quad + 2\gamma(1-\beta)\langle\nabla f(x^t) - \nabla f(x^{t+1}), \theta^{t+1}\rangle.
\end{aligned}
$$

Now we sum all the inequalities above for $t \in \{0, \ldots, K\}$ and get

$$\Phi^{K+1} \leq \Phi^0 - \frac{\gamma}{2} \sum_{t=0}^{K} \|\nabla f(x^t)\|^2 + Kb^2 \left( \frac{\beta^2 \gamma}{\eta} + \frac{4\gamma\beta^3}{\eta^2} \right) + Kc^2 \frac{\gamma\beta}{n}$$

$$+ \frac{2\gamma\beta}{n\eta} (1-\eta)^2 \sum_{t=0}^{K} \sum_{i=1}^{n} \langle (g_i^t - v_i^t) + \beta(v_i^t - \nabla f_i(x^t)) + \beta(\nabla f_i(x^t) - \nabla f_i(x^{t+1})), \theta_i^{t+1} \rangle$$

$$+ \frac{8\gamma\beta^2}{n\eta^2} (1-\beta) \sum_{t=0}^{K} \sum_{i=1}^{n} \langle v_i^t - \nabla f_i(x^t), \theta_i^{t+1} \rangle + 2\gamma(1-\beta) \sum_{t=0}^{K} \langle v^t - \nabla f(x^t), \theta^{t+1} \rangle$$

$$+ \frac{8\gamma\beta^2}{n\eta^2} (1-\beta) \sum_{t=0}^{K} \sum_{i=1}^{n} \langle \nabla f_i(x^t) - \nabla f_i(x^{t+1}), \theta_i^{t+1} \rangle$$

$$+ 2\gamma(1-\beta) \sum_{t=0}^{K} \langle \nabla f(x^t) - \nabla f(x^{t+1}), \theta^{t+1} \rangle. \tag{37}$$

Rearranging terms we get

$$\frac{\gamma}{2} \sum_{t=0}^{K} \|\nabla f(x^t)\|^2 \leq \Phi^0 - \Phi^{K+1} + Kb^2 \left( \frac{\beta^2 \gamma}{\eta} + \frac{4\gamma\beta^3}{\eta^2} \right) + Kc^2 \frac{\gamma\beta}{n}$$

$$+ \frac{2\gamma\beta}{n\eta} (1-\eta)^2 \sum_{t=0}^{K} \sum_{i=1}^{n} \langle (g_i^t - v_i^t) + \beta(v_i^t - \nabla f_i(x^t)) + \beta(\nabla f_i(x^t) - \nabla f_i(x^{t+1})), \theta_i^{t+1} \rangle$$

$$+ \frac{8\gamma\beta^2}{n\eta^2} (1-\beta) \sum_{t=0}^{K} \sum_{i=1}^{n} \langle v_i^t - \nabla f_i(x^t), \theta_i^{t+1} \rangle + 2\gamma(1-\beta) \sum_{t=0}^{K} \langle v^t - \nabla f(x^t), \theta^{t+1} \rangle$$

$$+ \frac{8\gamma\beta^2}{n\eta^2} (1-\beta) \sum_{t=0}^{K} \sum_{i=1}^{n} \langle \nabla f_i(x^t) - \nabla f_i(x^{t+1}), \theta_i^{t+1} \rangle$$

$$+ 2\gamma(1-\beta) \sum_{t=0}^{K} \langle \nabla f(x^t) - \nabla f(x^{t+1}), \theta^{t+1} \rangle.$$

Taking into account that $\frac{\gamma}{2} \sum_{t=0}^{K} \|\nabla f(x^t)\|^2 \geq 0$, we get that the event $E_K \cap \left( \cap_{i=1}^{n} \overline{\Theta}_i^{K+1} \right) \cap \overline{N}^t$ implies

$$\Phi^{K+1} \leq \Phi^0 + Kb^2 \left( \frac{\beta^2 \gamma}{\eta} + \frac{4\gamma\beta^3}{\eta^2} \right) + Kc^2 \frac{\gamma\beta}{n}$$

$$+ \frac{2\gamma\beta}{n\eta} (1-\eta)^2 \sum_{t=0}^{K} \sum_{i=1}^{n} \langle (g_i^t - v_i^t) + \beta(v_i^t - \nabla f_i(x^t)) + \beta(\nabla f_i(x^t) - \nabla f_i(x^{t+1})), \theta_i^{t+1} \rangle$$

$$+ \frac{8\gamma\beta^2}{n\eta^2} (1-\beta) \sum_{t=0}^{K} \sum_{i=1}^{n} \langle v_i^t - \nabla f_i(x^t), \theta_i^{t+1} \rangle + \frac{2\gamma(1-\beta)}{n} \sum_{t=0}^{K} \sum_{i=1}^{n} \langle v^t - \nabla f(x^t), \theta_i^{t+1} \rangle$$

$$+ \frac{8\gamma\beta^2}{n\eta^2} (1-\beta) \sum_{t=0}^{K} \sum_{i=1}^{n} \langle \nabla f_i(x^t) - \nabla f_i(x^{t+1}), \theta_i^{t+1} \rangle$$

$$+ \frac{2\gamma(1-\beta)}{n} \sum_{t=0}^{K} \sum_{i=1}^{n} \langle \nabla f(x^t) - \nabla f(x^{t+1}), \theta_i^{t+1} \rangle.$$

Now we define the following random vectors:

$$\zeta_{1,i}^t := \begin{cases} g_i^t - v_i^t, & \text{if } \|g_i^t - v_i^t\| \le B \\ 0, & \text{otherwise} \end{cases},$$

$$\zeta_{2,t}^t := \begin{cases} v_i^t - \nabla f_i(x^t), & \text{if } \|v_i^t - \nabla f_i(x^t)\| \le \sqrt{4L\Delta} + \frac{3}{2}(B-\tau) + 2b + (L\gamma)^p a \\ 0, & \text{otherwise} \end{cases},$$

$$\zeta_{3,i}^t := \begin{cases} \nabla f_i(x^t) - \nabla f_i(x^{t+1}), & \text{if } \|\nabla f_i(x^t) - \nabla f_i(x^{t+1})\| \le L\gamma\left(\sqrt{64L\Delta} + 3(B-\tau) + 3b + 3a\right) \\ 0, & \text{otherwise} \end{cases},$$

$$\zeta_4^t := \begin{cases} v^t - \nabla f(x^t), & \text{if } \|v^t - \nabla f(x^t)\| \le \sqrt{4L\Delta} + \frac{3}{2}(B-\tau) + 2b + (L\gamma)^p a \\ 0, & \text{otherwise} \end{cases},$$

$$\zeta_5^t := \begin{cases} \nabla f(x^t) - \nabla f(x^{t+1}), & \text{if } \|\nabla f(x^t) - \nabla f(x^{t+1})\| \le L\gamma\left(\sqrt{64L\Delta} + 3(B-\tau) + 3b + 3a\right) \\ 0, & \text{otherwise} \end{cases}.$$

By definition, all introduced random vectors $\zeta_{l,i}^t, l \in [3], i \in [n], \zeta_{4,5}^t$ are bounded with probability 1. Moreover, by the definition of $\Phi^t$ and definition of $E_t$ we get that the event $E_K \cap \overline{\Theta}^{K+1} \cap \left(\cap_{i=1}^n \overline{\Theta}_i^{K+1}\right) \cap \overline{N}^{K+1}$ implies

$$\zeta_{1,i}^t = g_i^t - v_i^t, \quad \zeta_{2,i}^t = v_i^t - \nabla f_i(x^t), \quad \zeta_{3,i}^t = \nabla f_i(x^t) - \nabla f_i(x^{t+1}),$$
$$\zeta_4^t = v^t - \nabla f(x^t), \quad \zeta_5^t = \nabla f(x^t) - \nabla f(x^{t+1}).$$

Therefore, the event $E_K \cap \overline{\Theta}^{K+1} \cap \left(\cap_{i=1}^n \overline{\Theta}_i^{K+1}\right) \cap \overline{N}^{K+1}$ implies

$$\Phi^{K+1} \le \Phi^0 + \underbrace{Kb^2\left(\frac{\beta^2\gamma}{\eta} + \frac{4\gamma\beta^3}{\eta^2}\right) + Kc^2\frac{\gamma\beta}{n}}_{①} + \underbrace{\frac{2\gamma\beta}{n\eta}(1-\eta)^2\sum_{t=0}^K\sum_{i=1}^n\langle\zeta_{1,i}^t, \theta_i^{t+1}\rangle}_{②}$$

$$+ \underbrace{\frac{2\gamma\beta^2}{n\eta}(1-\eta)^2\sum_{t=0}^K\sum_{i=1}^n\langle\zeta_{2,i}^t, \theta_i^{t+1}\rangle}_{③} + \underbrace{\frac{2\gamma\beta^2}{n\eta}(1-\eta)^2\sum_{t=0}^K\sum_{i=1}^n\langle\zeta_{3,i}^t, \theta_i^{t+1}\rangle}_{④}$$

$$+ \underbrace{\frac{8\gamma\beta^2}{n\eta^2}(1-\beta)\sum_{t=0}^K\sum_{i=1}^n\langle\zeta_{2,i}^t, \theta_i^{t+1}\rangle}_{⑤} + \underbrace{\frac{2\gamma(1-\beta)}{n}\sum_{t=0}^K\sum_{i=1}^n\langle\zeta_4^t, \theta_i^{t+1}\rangle}_{⑥}$$

$$+ \underbrace{\frac{8\gamma\beta^2}{n\eta^2}(1-\beta)\sum_{t=0}^K\sum_{i=1}^n\langle\zeta_{3,i}^t, \theta_i^{t+1}\rangle}_{⑦} + \underbrace{\frac{2\gamma(1-\beta)}{n}\sum_{t=0}^K\sum_{i=1}^n\langle\zeta_5^t, \theta_i^{t+1}\rangle}_{⑧}.$$

BOUND OF THE TERM ①. For the term ① we have

$$Kb^2\left(\frac{\beta^2\gamma}{\eta} + \frac{4\gamma\beta^3}{\eta^2}\right) + Kc^2\frac{\gamma\beta}{n} \le Kb^2\left(\frac{9(\beta/3)^{\frac{3-2p}{1-p}}}{L\eta} + \frac{108(\beta/3)^{\frac{4-3p}{1-p}}}{L\eta^2}\right) + 3Kc^2\frac{(\beta/3)^{\frac{2-p}{1-p}}}{Ln}.$$

By choosing $\gamma$ such that

$$\beta \le \min\left\{3\left(\frac{L\Delta\eta}{216Tb^2}\right)^{\frac{1-p}{3-2p}}, 3\left(\frac{L\Delta\eta^2}{48Tb^2}\right)^{\frac{1-p}{4-3p}}, 3\left(\frac{L\Delta n}{72Tc^2}\right)^{\frac{1-p}{2-p}}\right\} \tag{38}$$

we get that

$$Kb^2\left(\frac{\beta^2\gamma}{\eta} + \frac{4\gamma\beta^3}{\eta^2}\right) + Kc^2\frac{\gamma\beta}{n} \le 3\cdot\frac{\Delta}{24} = \frac{\Delta}{8}.$$

This bound holds with probability 1. Note that the worst dependency in the restriction on $\beta$ in $T$ is $\mathcal{O}(1/T^{\frac{1-p}{2-p}})$ that comes from the last term in $\min$.

BOUND OF THE TERM ②. For term ②, let us enumerate random variables as

$$\langle\zeta^0_{1,1},\theta^1_1\rangle,\ldots,\langle\zeta^0_{1,n},\theta^1_n\rangle,\langle\zeta^1_{1,1},\theta^2_1\rangle,\ldots,\langle\zeta^1_{1,n},\theta^2_n\rangle,\ldots\langle\zeta^K_{1,1},\theta^{K+1}_1\rangle,\ldots,\langle\zeta^K_{1,n},\theta^{K+1}_n\rangle,$$

i.e. first by index $i$, then by index $t$. Then we have that the event $E_K \cap \left(\cap^n_{i=1}\overline{\Theta}^{K+1}_i\right)$ implies

$$\mathbb{E}\left[\frac{2\gamma\beta}{n\eta}(1-\eta)^2\langle\zeta^l_{1,i},\theta^{l+1}_i\rangle \mid \langle\zeta^l_{1,i-1},\theta^{l+1}_{i-1}\rangle,\ldots,\langle\zeta^l_{1,1},\theta^{l+1}_1\rangle,\ldots,\langle\zeta^0_{1,1},\theta^1_1\rangle\right] = 0,$$

because $\{\theta^{l+1}_i\}^n_{i=1}$ are independent. Let

$$\sigma^2_2 := \frac{4\gamma^2\beta^2}{n^2\eta^2}\cdot B^2\cdot\sigma^2.$$

Since $\theta^{l+1}_i$ is $\sigma$-sub-Gaussian random vector, we have

$$\mathbb{E}\left[\exp\left(\left|\frac{1}{\sigma^2_2}\frac{4\gamma^2\beta^2}{n^2\eta^2}(1-\eta)^4\langle\zeta^l_{1,i},\theta^{l+1}_i\rangle^2\right|\right)\mid l,i-1\right]$$

$$\leq \mathbb{E}\left[\exp\left(\frac{1}{\sigma^2_1}\frac{4\gamma^2\beta^2}{n^2\eta^2}\|\zeta^l_{1,i}\|^2\cdot\|\theta^{l+1}_i\|^2\right)\mid l,i-1\right]$$

$$\leq \mathbb{E}\left[\exp\left(\frac{1}{\sigma^2_2}\frac{4\gamma^2\beta^2}{n^2\eta^2}\cdot B^2\|\theta^{l+1}_i\|^2\right)\mid l,i-1\right]$$

$$\leq \mathbb{E}\left[\exp\left(\frac{n^2\eta^2}{4\gamma^2\beta^2\cdot B^2\cdot\sigma^2}\frac{4\gamma^2\beta^2}{n^2\eta^2}\cdot B^2\|\theta^{l+1}_i\|^2\right)\mid l,i-1\right]$$

$$= \mathbb{E}\left[\exp\left(\frac{\|\theta^{l+1}_i\|^2}{\sigma^2}\mid l,i-1\right)\right] \leq \exp(1).$$

Here $\mathbb{E}\left[\cdot \mid l,i-1\right]$ means

$$\mathbb{E}\left[\cdot \mid \langle\zeta^l_{1,i-1},\theta^{l+1}_{i-1}\rangle,\ldots,\langle\zeta^l_{1,1},\theta^{l+1}_1\rangle,\ldots,\langle\zeta^0_{1,1},\theta^1_1\rangle\right] = 0,$$

Therefore, we have by Lemma 1 with $\sigma^2_k \equiv \sigma^2_2$ that

$$\Pr\left(\frac{2\gamma\beta}{n\eta}(1-\eta)^2\left\|\sum^K_{t=0}\sum^n_{i=1}\langle\zeta^t_{1,i},\theta^{t+1}_i\rangle\right\| \geq (\sqrt{2}+\sqrt{2}b_1)\sqrt{\sum^K_{t=0}\sum^n_{i=1}\frac{4B^2\gamma^2\beta^2\sigma^2}{n^2\eta^2}}\right)$$

$$\leq \exp(-b^2_1/3)$$

$$= \frac{\alpha}{14(T+1)}$$

with $b^2_1 = 3\log\left(\frac{14(T+1)}{\alpha}\right)$. Note that

$$(\sqrt{2}+\sqrt{2}b_1)\sqrt{\sum^K_{t=0}\sum^n_{i=1}\frac{4B^2\gamma^2\beta^2\sigma^2}{n^2\eta^2}} \leq (\sqrt{2}+\sqrt{2}b_1)\sqrt{\sum^K_{t=0}\sum^n_{i=1}\frac{36B^2(\beta/3)^{\frac{4-2p}{1-p}}\sigma^2}{L^2n^2\eta^2}}$$

$$= (\sqrt{2}+\sqrt{2}b_1)\frac{6B(\beta/3)^{\frac{2-p}{1-p}}\sigma}{Ln\eta}\sqrt{(K+1)n}$$

$$\leq \frac{\Delta}{8},$$

because we choose

$$\beta \leq 3\left(\frac{L\Delta\sqrt{n}\eta}{48\sqrt{2}(1+b_1)B\sigma\sqrt{T}}\right)^{\frac{1-p}{2-p}}, \quad \text{and} \quad K+1 \leq T. \tag{39}$$

This implies that

$$\Pr\left(\frac{2\gamma\beta}{n\eta}(1-\eta)^2\left\|\sum^K_{t=0}\sum^n_{i=1}\langle\zeta^t_{1,i},\theta^{t+1}_i\rangle\right\| \geq \frac{\Delta}{8}\right) \leq \frac{\alpha}{14(T+1)}$$

with this choice of momentum parameter. The dependency on $T$ is $\widetilde{\mathcal{O}}(1/T^{\frac{1-p}{2(2-p)}})$.

BOUND OF THE TERM ③. The bound in this case is similar to the previous one. Let

$$\sigma_3^2 := \frac{4\gamma^2\beta^4}{n^2\eta^2} \cdot \left(\sqrt{4L\Delta} + \frac{3}{2}(B-\tau) + 2b + (L\gamma)^p a\right)^2 \cdot \sigma^2.$$

$$\mathbb{E}\left[\exp\left(\left|\frac{1}{\sigma_3^2}\frac{4\gamma^2\beta^4}{n^2\eta^2}(1-\eta)^4\langle\zeta_{2,i}^l, \theta_i^{l+1}\rangle^2\right|\right) \mid l, i-1\right]$$

$$\leq \mathbb{E}\left[\exp\left(\frac{1}{\sigma_3^2}\frac{4\gamma^2\beta^4}{n^2\eta^2}\|\zeta_{2,i}^l\|^2 \cdot \|\theta_i^{l+1}\|^2\right)\right]$$

$$\leq \mathbb{E}\left[\exp\left(\frac{1}{\sigma_3^2}\frac{4\gamma^2\beta^4}{n^2\eta^2} \cdot \left(\sqrt{4L\Delta} + \frac{3}{2}(B-\tau) + 2b + (L\gamma)^p a\right)^2 \cdot \|\theta_i^{l+1}\|^2\right) \mid l, i-1\right]$$

$$\leq \mathbb{E}\left[\exp\left(\left[\frac{4\gamma^2\beta^4}{n^2\eta^2} \cdot \left(\sqrt{4L\Delta} + \frac{3}{2}(B-\tau) + 2b + (L\gamma)^p a\right)^2 \cdot \sigma^2\right]^{-1} \cdot \right.\right.$$

$$\left.\left. \frac{4\gamma^2\beta^4}{n^2\eta^2} \cdot \left(\sqrt{4L\Delta} + \frac{3}{2}(B-\tau) + 2b + (L\gamma)^p a\right)^2 \cdot \|\theta_i^{l+1}\|^2\right) \mid l, i-1\right]$$

$$= \mathbb{E}\left[\exp\left(\frac{\|\theta_i^{l+1}\|^2}{\sigma^2}\right) \mid l, i-1\right] \leq \exp(1).$$

Therefore, we have by Lemma 1 that

$$\Pr\left[\frac{2\gamma\beta^2}{n\eta}(1-\eta)^2\left\|\sum_{t=0}^{K}\sum_{i=1}^{n}\langle\zeta_{2,i}^t, \theta_i^{t+1}\rangle\right\|\right.$$

$$\left. \geq (\sqrt{2} + \sqrt{2}b_1)\sqrt{\sum_{t=0}^{K}\sum_{i=1}^{n}\frac{4\gamma^2\beta^4\sigma^2}{n^2\eta^2} \cdot \left(\sqrt{4L\Delta} + \frac{3}{2}(B-\tau) + 2b + (L\gamma)^p a\right)^2}\right]$$

$$\leq \exp(-b_1^2/3) = \frac{\alpha}{14(T+1)},$$

Note that

$$(\sqrt{2} + \sqrt{2}b_1)\sqrt{(K+1)n}\frac{2\gamma\beta^2\sigma}{\eta n}\left(\sqrt{4L\Delta} + \frac{3}{2}(B-\tau) + 2b + (L\gamma)^p a\right)$$

$$\leq (\sqrt{2} + \sqrt{2}b_1)\sqrt{(K+1)n}\frac{18(\beta/3)^{\frac{3-2p}{1-p}}\sigma}{L\eta n}\left(\sqrt{4L\Delta} + \frac{3}{2}(B-\tau) + 2b + (\beta/3)^{\frac{p}{1-p}} a\right)$$

$$\leq \frac{\Delta}{8}.$$

because we choose

$$\beta \leq \min\left\{3\left(\frac{L\Delta\eta\sqrt{n}}{288\sqrt{2}(1+b_1)\sigma\sqrt{T}\left(\sqrt{4L\Delta} + \frac{3}{2}(B-\tau) + 2b\right)}\right)^{\frac{1-p}{3-2p}}\right.$$

$$\left. 3\left(\frac{L\Delta\eta\sqrt{n}}{288\sqrt{2}(1+b_1)\sqrt{T}\sigma a}\right)^{\frac{1-p}{3-p}}\right\},$$

$$\text{and} \quad K+1 \leq T. \tag{40}$$

This implies

$$\Pr\left(\frac{2\gamma\beta^2}{n\eta}(1-\eta)^2\left\|\sum_{t=0}^{K}\sum_{i=1}^{n}\langle\zeta_{2,i}^t, \theta_i^{t+1}\rangle\right\| \geq \frac{\Delta}{8}\right) \leq \frac{\alpha}{14(T+1)}.$$

Note that the worst dependency w.r.t. $T$ is $\widetilde{\mathcal{O}}(1/T^{\frac{3(1-p)}{2(3-p)}})$ since $a \approx \sigma_\omega\sqrt{T} \sim T$.

BOUND OF THE TERM ④.    The bound in this case is similar to the previous one. Let

$$\sigma_4^2 := \frac{4L^2\gamma^4\beta^4}{n^2\eta^2}\left(\sqrt{64L\Delta} + 3(B-\tau) + 3b + 3a\right)^2 \cdot \sigma^2.$$

Then we have

$$\mathbb{E}\left[\exp\left(\left|\frac{1}{\sigma_4^2}\frac{4\gamma^2\beta^4}{n^2\eta^2}(1-\eta)^4\langle\zeta_{3,i}^l, \theta_i^{l+1}\rangle^2\right|\right) \mid l, i-1\right]$$

$$\leq \mathbb{E}\left[\exp\left(\frac{1}{\sigma_4^2}\frac{4\gamma^2\beta^4}{n^2\eta^2}\|\zeta_{3,i}^l\|^2 \cdot \|\theta_i^{l+1}\|^2\right) \mid l, i-1\right]$$

$$\leq \mathbb{E}\left[\exp\left(\frac{1}{\sigma_4^2}\frac{4\gamma^2\beta^4}{n^2\eta^2} \cdot L^2\gamma^2\left(\sqrt{64L\Delta} + 3(B-\tau) + 3b + 3a\right)^2 \cdot \|\theta_i^{l+1}\|^2\right) \mid l, i-1\right]$$

$$\leq \mathbb{E}\left[\exp\left(\left[\frac{4L^2\gamma^4\beta^4}{n^2\eta^2}\left(\sqrt{64L\Delta} + 3(B-\tau) + 3b + 3a\right)^2 \cdot \sigma^2\right]^{-1}\right.\right.$$

$$\left.\left.\frac{4L^2\gamma^4\beta^4}{n^2\eta^2} \cdot \frac{16\beta^4}{81}\left(\sqrt{64L\Delta} + 3(B-\tau) + 3b + 3a\right)^2 \cdot \|\theta_i^{l+1}\|^2\right) \mid l, i-1\right]$$

$$= \mathbb{E}\left[\exp\left(\frac{\|\theta_i^{l+1}\|^2}{\sigma^2}\right)\right] \leq \exp(1).$$

Therefore, we have by Lemma 1 that

$$\Pr\left(\frac{2\gamma\beta^2}{n\eta}(1-\eta)^2\left\|\sum_{t=0}^{K}\sum_{i=1}^{n}\langle\zeta_{3,i}^t, \theta_i^{t+1}\rangle\right\|\right.$$

$$\geq (\sqrt{2} + \sqrt{2}b_1)\sqrt{\sum_{t=0}^{K}\sum_{i=1}^{n}\frac{4L^2\gamma^4\beta^4\sigma^2}{n^2\eta^2} \cdot \left(\sqrt{64L\Delta} + 3(B-\tau+b) + 3a\right)^2}\right)$$

$$\leq \exp(-b_1^2/3) = \frac{\alpha}{14(T+1)}.$$

Note that

$$(\sqrt{2} + \sqrt{2}b_1)\sqrt{(K+1)n}\frac{2L\gamma^2\beta^2\sigma}{\eta n}\left(\sqrt{64L\Delta} + 3(B-\tau+b) + 3a\right)$$

$$\leq \sqrt{2}(1+b_1)\sqrt{(K+1)n}\frac{18(\beta/3)^{\frac{4-2p}{1-p}}\sigma}{L\eta n}\left(\sqrt{64L\Delta} + 3(B-\tau+b) + 3a\right)$$

$$\leq \frac{\Delta}{8}.$$

because we choose

$$\beta \leq \min\left\{3\left(\frac{L\Delta\eta\sqrt{n}}{288\sqrt{2}(1+b_1)\sigma\sqrt{T}\left(\sqrt{64L\Delta} + 3(B-\tau+b)\right)}\right)^{\frac{1-p}{4-2p}}\right.$$

$$\left.3\left(\frac{L\Delta\eta\sqrt{n}}{288\sqrt{2}(1+b_1)\sigma\sqrt{T}a}\right)^{\frac{1-p}{4-2p}}\right\},$$

$$\text{and}\quad K+1 \leq T. \tag{41}$$

This implies

$$\Pr\left(\frac{2\gamma\beta^2}{n\eta}(1-\eta)^2\left\|\sum_{t=0}^{K}\sum_{i=1}^{n}\langle\zeta_{2,i}^t, \theta_i^{t+1}\rangle\right\| \geq \frac{\Delta}{8}\right) \leq \frac{\alpha}{14(T+1)},$$

Note that the worst dependency w.r.t. $T$ is $\widetilde{\mathcal{O}}(1/T^{\frac{3(1-p)}{4(2-p)}})$ since $a \sim T$.

BOUND OF THE TERM ⑤.    The bound in this case is similar to the previous one. Let

$$\sigma_5^2 := \frac{64\gamma^2\beta^4}{n^2\eta^4} \cdot \left(\sqrt{4L\Delta} + \frac{3}{2}(B-\tau) + 2b + (L\gamma)^p a\right)^2 \cdot \sigma^2.$$

Then we have

$$\mathbb{E}\left[\exp\left(\left|\frac{1}{\sigma_5^2}\frac{64\gamma^2\beta^4}{n^2\eta^4}(1-\beta)^2\langle\zeta_{2,i}^l,\theta_i^{l+1}\rangle^2\right|\right) \mid l, i-1\right]$$

$$\leq \mathbb{E}\left[\exp\left(\frac{1}{\sigma_5^2}\frac{64\gamma^2\beta^4}{n^2\eta^4}\|\zeta_{2,i}^l\|^2 \cdot \|\theta_i^{l+1}\|^2\right) \mid l, i-1\right]$$

$$\leq \mathbb{E}\left[\exp\left(\frac{1}{\sigma_5^2}\frac{64\gamma^2\beta^4}{n^2\eta^4} \cdot \left(\sqrt{4L\Delta} + \frac{3}{2}(B-\tau) + 2b + (L\gamma)^p a\right)^2 \cdot \|\theta_i^{l+1}\|^2\right) \mid l, i-1\right]$$

$$= \mathbb{E}\left[\exp\left(\left[\frac{64\gamma^2\beta^4}{81L^2n^2\eta^4} \cdot \left(\sqrt{4L\Delta} + \frac{3}{2}(B-\tau) + 2b + (L\gamma)^p a\right)^2 \cdot \sigma^2\right]^{-1}\right.\right.$$

$$\left.\left.\frac{64\gamma^2\beta^4}{n^2\eta^4} \cdot \left(\sqrt{4L\Delta} + \frac{3}{2}(B-\tau) + 2b + (L\gamma)^p a\right)^2 \cdot \|\theta_i^{l+1}\|^2\right) \mid l, i-1\right]$$

$$= \mathbb{E}\left[\exp\left(\frac{\|\theta_i^{l+1}\|^2}{\sigma^2}\right) \mid l, i-1\right] \leq \exp(1).$$

Therefore, we have by Lemma 1 that

$$\Pr\left[\frac{8\gamma\beta^2}{n\eta^2}(1-\beta)\left\|\sum_{t=0}^K\sum_{i=1}^n\langle\zeta_{2,i}^t,\theta_i^{t+1}\rangle\right\|\right.$$

$$\left.\geq (\sqrt{2}+\sqrt{2}b_1)\sqrt{\sum_{t=0}^K\sum_{i=1}^n\frac{64\gamma^2\beta^4\sigma^2}{n^2\eta^4}\left(\sqrt{4L\Delta} + \frac{3}{2}(B-\tau) + 2b + (L\gamma)^p a\right)^2}\right]$$

$$\leq \exp(-b_1^2/3) = \frac{\alpha}{14(T+1)}.$$

Note that

$$(\sqrt{2}+\sqrt{2}b_1)\sqrt{(K+1)n}\frac{8\gamma\beta^2\sigma}{n\eta^2}\left(\sqrt{4L\Delta} + \frac{3}{2}(B-\tau) + 2b + (L\gamma)^p a\right)$$

$$\leq(\sqrt{2}+\sqrt{2}b_1)\sqrt{(K+1)n}\frac{72(\beta/3)^{\frac{3-2p}{1-p}}\sigma}{Ln\eta^2}\left(\sqrt{4L\Delta} + \frac{3}{2}(B-\tau) + 2b + (\beta/3)^{\frac{p}{1-p}}a\right)$$

$$\leq\frac{\Delta}{8}$$

because we choose

$$\beta \leq \min\left\{3\left(\frac{L\Delta\eta^2\sqrt{n}}{1152\sqrt{2}(1+b_1)\sigma\sqrt{T}\left(\sqrt{4L\Delta} + \frac{3}{2}(B-\tau) + 2b\right)}\right)^{\frac{1-p}{3-2p}}\right.$$

$$\left.3\left(\frac{L\Delta\eta^2\sqrt{n}}{1152\sqrt{2}(1+b_1)\sigma\sqrt{T}a}\right)^{\frac{1-p}{3-p}}\right\},$$

$$\text{and } K+1 \leq T. \tag{42}$$

This implies

$$\Pr\left(\frac{8\gamma\beta^2}{n\eta^2}(1-\beta)\left\|\sum_{t=0}^K\sum_{i=1}^n\langle\zeta_{2,i}^t,\theta_i^{t+1}\rangle\right\| \geq \frac{\Delta}{8}\right) \leq \frac{\alpha}{14(T+1)}.$$

Note that the worst dependency w.r.t. $T$ is $\widetilde{\mathcal{O}}(1/T^{\frac{3(1-p)}{2(3-p)}})$ since $a \sim T$.

BOUND OF THE TERM ⑦. The bound in this case is similar to the previous one. Let

$$\sigma_7^2 := \frac{64\gamma^2\beta^4}{n^2\eta^4}\left(\sqrt{64L\Delta} + 3(B - \tau + b) + 3a\right)^2 \cdot \sigma^2.$$

Then we have

$$\mathbb{E}\left[\exp\left(\left|\frac{1}{\sigma_7^2}\frac{64L^2\gamma^4\beta^4}{n^2\eta^4}(1-\beta)^2\langle\zeta_{3,i}^l, \theta_i^{l+1}\rangle^2\right|\right) \mid l, i-1\right]$$

$$\leq \mathbb{E}\left[\exp\left(\frac{1}{\sigma_7^2}\frac{64\gamma^2\beta^4}{n^2\eta^4}\|\zeta_{3,i}^l\|^2 \cdot \|\theta_i^{l+1}\|^2\right) \mid l, i-1\right]$$

$$\leq \mathbb{E}\left[\exp\left(\frac{64\gamma^2\beta^4}{n^2\eta^4} \cdot L^2\gamma^2\left(\sqrt{64L\Delta} + 3(B-\tau+b) + 3a\right)^2 \cdot \|\theta_i^{l+1}\|^2\right) \mid l, i-1\right]$$

$$\leq \mathbb{E}\left[\exp\left(\left[\frac{64L^2\gamma^4\beta^4}{n^2\eta^4}\left(\sqrt{64L\Delta} + 3(B-\tau+b) + 3a\right)^2 \cdot \sigma^2\right]^{-1}\right.\right.$$

$$\left.\left.\frac{64L^2\gamma^4\beta^4}{n^2\eta^4}\left(\sqrt{64L\Delta} + 3(B-\tau+b) + 3a\right)^2 \cdot \|\theta_i^{l+1}\|^2\right) \mid l, i-1\right]$$

$$= \mathbb{E}\left[\exp\left(\frac{\|\theta_i^{l+1}\|^2}{\sigma^2}\right) \mid l, i-1\right] \leq \exp(1).$$

Therefore, we have by Lemma 1 that

$$\Pr\left[\frac{8\gamma\beta^2}{n\eta^2}(1-\beta)\left\|\sum_{t=0}^{K}\sum_{i=1}^{n}\langle\zeta_{3,i}^t, \theta_i^{t+1}\rangle\right\| \geq\right.$$

$$\left.(\sqrt{2} + \sqrt{2}b_1)\sqrt{\sum_{t=0}^{K}\sum_{i=1}^{n}\frac{64L^2\gamma^4\beta^4\sigma^2}{n^2\eta^4} \cdot \left(\sqrt{64L\Delta} + 3(B-\tau+b) + 3a\right)^2}\right]$$

$$\leq \exp(-b_1^2/3) = \frac{\alpha}{14(T+1)}.$$

Note that

$$(\sqrt{2} + \sqrt{2}b_1)\sqrt{(K+1)n}\frac{8L\gamma^2\beta^2\sigma}{\eta^2 n}\left(\sqrt{64L\Delta} + 3(B-\tau+b) + 3a\right)$$

$$\leq(\sqrt{2} + \sqrt{2}b_1)\sqrt{(K+1)n}\frac{72(\beta/3)^{\frac{4-2p}{1-p}}\sigma}{L\eta^2 n}\left(\sqrt{64L\Delta} + 3(B-\tau+b) + 3a\right)$$

$$\leq\frac{\Delta}{8}$$

because we choose

$$\beta \leq \min\left\{\left(\frac{L\Delta\eta^2\sqrt{n}}{1152\sqrt{2}(1+b_1)\sigma\sqrt{T}\left(\sqrt{64L\Delta} + 3(B-\tau+b)\right)}\right)^{\frac{1-p}{4-2p}}\right.$$

$$\left.\left(\frac{L\Delta\eta^2\sqrt{n}}{3456\sqrt{2}(1+b_1)\sigma\sqrt{T}a}\right)^{\frac{1-p}{4-2p}}\right\}$$

$$\text{and} \quad K+1 \leq T. \tag{43}$$

This implies

$$\Pr\left(\frac{8\gamma\beta^2}{n\eta^2}(1-\beta)\left\|\sum_{t=0}^{K}\sum_{i=1}^{n}\langle\zeta_{3,i}^t, \theta_i^{t+1}\rangle\right\| \geq \frac{\Delta}{8}\right) \leq \frac{\alpha}{14(T+1)}.$$

Note that the worst dependency w.r.t. $T$ is $\widetilde{\mathcal{O}}(1/T^{\frac{3(1-p)}{4(2-p)}})$.

BOUND OF THE TERM ⑥.    The bound in this case is similar to the previous one. Let

$$\sigma_6^2 := \frac{4\gamma^2}{n^2}\left(\sqrt{4L\Delta} + \frac{3}{2}(B-\tau) + 2b + (L\gamma)^p a\right)^2 \cdot \sigma^2.$$

Then we have

$$\mathbb{E}\left[\exp\left(\left|\frac{1}{\sigma_6^2}\frac{4\gamma^2}{n^2}(1-\beta)^2\langle\zeta_4^l, \theta_i^{l+1}\rangle^2\right|\right) \mid l, i-1\right]$$

$$\leq \mathbb{E}\left[\exp\left(\frac{1}{\sigma_6^2}\frac{4\gamma^2}{n^2}\|\zeta_4^l\|^2 \cdot \|\theta_i^{l+1}\|^2\right) \mid l, i-1\right]$$

$$\leq \mathbb{E}\left[\exp\left(\frac{1}{\sigma_6^2}\frac{4\gamma^2}{n^2}\left(\sqrt{4L\Delta} + \frac{3}{2}(B-\tau) + 2b + (L\gamma)^p a\right)^2 \cdot \|\theta_i^{l+1}\|^2\right) \mid l, i-1\right]$$

$$\leq \mathbb{E}\left[\exp\left(\left[\frac{4\gamma^2}{n^2}\left(\sqrt{4L\Delta} + \frac{3}{2}(B-\tau) + 2b + (L\gamma)^p a\right)^2 \cdot \sigma^2\right]^{-1}\right.\right.$$

$$\left.\left.\frac{4\gamma^2}{n^2}\left(\sqrt{4L\Delta} + \frac{3}{2}(B-\tau) + 2b + (L\gamma)^p a\right)^2 \cdot \|\theta_i^{l+1}\|^2\right) \mid l, i-1\right]$$

$$= \mathbb{E}\left[\exp\left(\frac{\|\theta_i^{t+1}\|^2}{\sigma^2}\right) \mid l, i-1\right] \leq \exp(1).$$

Therefore, we have by Lemma 1 that

$$\Pr\left[\frac{2\gamma(1-\beta)}{n}\left\|\sum_{t=0}^{K}\sum_{i=1}^{n}\langle\zeta_{4,i}^t, \theta_i^{t+1}\rangle\right\|\right.$$

$$\left.\geq (\sqrt{2} + \sqrt{2}b_1)\sqrt{\sum_{t=0}^{K}\sum_{i=1}^{n}\frac{4\gamma^2}{n^2}\sigma^2 \cdot \left(\sqrt{4L\Delta} + \frac{3}{2}(B-\tau) + 2b + (L\gamma)^p a\right)^2}\right]$$

$$\leq \exp(-b_1^2/3) = \frac{\alpha}{14(T+1)},$$

Note that

$$(\sqrt{2} + \sqrt{2}b_1)\sqrt{(K+1)n} \cdot \frac{2\gamma}{n}\sigma\left(\sqrt{4L\Delta} + \frac{3}{2}(B-\tau) + 2b + (L\gamma)^p a\right)$$

$$\leq(\sqrt{2} + \sqrt{2}b_1)\sqrt{(K+1)n} \cdot \frac{2(\beta/3)^{\frac{1}{1-p}}}{Ln}\sigma\left(\sqrt{4L\Delta} + \frac{3}{2}(B-\tau) + 2b + (\beta/3)^{\frac{p}{1-p}}a\right)$$

$$\leq\frac{\Delta}{8}$$

because we choose

$$\beta \leq \min\left\{3\left(\frac{L\Delta\sqrt{n}}{32\sqrt{2}(1+b_1)\sigma\sqrt{T}\left(\sqrt{4L\Delta} + \frac{3}{2}(B-\tau) + 2b\right)}\right)^{1-p}\right.$$

$$\left.\left(\frac{L\Delta\sqrt{n}}{32\sqrt{2}(1+b_1)\sigma\sqrt{T}a}\right)^{\frac{1-p}{1+p}}\right\},$$

$$\text{and}\quad K+1 \leq T. \tag{44}$$

This implies

$$\Pr\left(\frac{2\gamma(1-\beta)}{n}\left\|\sum_{t=0}^{K}\sum_{i=1}^{n}\langle\zeta_{4,i}^t, \theta_i^{t+1}\rangle\right\| \geq \frac{\Delta}{8}\right) \leq \frac{\alpha}{14(T+1)}.$$

Note that the worst dependency w.r.t. $T$ is $\widetilde{\mathcal{O}}(1/T^{\frac{3(1-p)}{2(1+p)}})$ since $a \sim T$.

BOUND OF THE TERM ⑧. The bound in this case is similar to the previous one. Let

$$\sigma_8^2 := \frac{4L^2\gamma^4}{n^2} \cdot \left(\sqrt{64L\Delta} + 3(B - \tau + b) + 3a\right)^2 \cdot \sigma^2.$$

Then we have

$$\mathbb{E}\left[\exp\left(\left|\frac{1}{\sigma_8^2}\frac{4\gamma^2}{n^2}(1-\beta)^2\langle\zeta_5^l, \theta_i^{l+1}\rangle^2\right|\right) \mid l, i-1\right]$$

$$\leq \mathbb{E}\left[\exp\left(\frac{1}{\sigma_8^2}\frac{4\gamma^2}{n^2}\|\zeta_5^l\|^2 \cdot \|\theta_i^{l+1}\|^2\right) \mid l, i-1\right]$$

$$\leq \mathbb{E}\left[\exp\left(\frac{1}{\sigma_8^2}\frac{4\gamma^2}{n^2}L^2\gamma^2\left(\sqrt{64L\Delta} + 3(B - \tau + b) + 3a\right)^2 \cdot \|\theta_i^{l+1}\|^2\right) \mid l, i-1\right].$$

Since $\theta_i^{l+1}$ is sub-Gaussian with parameter $\sigma^2$, then we can continue the chain of inequalities above using the definition of $\sigma_8^2$

$$\mathbb{E}\left[\exp\left(\left[\frac{4L^2\gamma^4}{n^2} \cdot \left(\sqrt{64L\Delta} + 3(B - \tau + b) + 3a\right)^2 \cdot \sigma^2\right]^{-1}\right.\right.$$

$$\left.\left.\frac{4L^2\gamma^4}{n^2} \cdot \left(\sqrt{64L\Delta} + 3(B - \tau + b) + 3a\right)^2 \cdot \|\theta_i^{l+1}\|^2\right) \mid l, i-1\right]$$

$$= \mathbb{E}\left[\exp\left(\frac{\|\theta_i^{l+1}\|^2}{\sigma^2}\right)\right] \leq \exp(1).$$

Therefore, we have by Lemma 1 that

$$\Pr\left[\frac{2\gamma(1-\beta)}{n}\left\|\sum_{t=0}^{K}\sum_{i=1}^{n}\langle\zeta_{5,i}^t, \theta^{t+1}\rangle\right\|\right.$$

$$\geq (\sqrt{2} + \sqrt{2}b_1)\sqrt{\sum_{t=0}^{K}\sum_{i=1}^{n}\frac{4L^2\gamma^4}{n^2}\sigma^2 \cdot \left(\sqrt{64L\Delta} + 3(B - \tau + b) + 3a\right)^2}\right]$$

$$\leq \exp(-b_1^2/3) = \frac{\alpha}{14(T+1)}.$$

Note that

$$(\sqrt{2} + \sqrt{2}b_1)\sqrt{(K+1)n} \cdot \frac{2L\gamma^2}{n}\sigma\left(\sqrt{64L\Delta} + 3(B - \tau + b) + 3a\right)$$

$$\leq (\sqrt{2} + \sqrt{2}b_1)\sqrt{(K+1)n} \cdot \frac{2(\beta/3)^{\frac{2}{1-p}}}{Ln}\sigma\left(\sqrt{64L\Delta} + 3(B - \tau + b) + 3a\right)$$

$$\leq \frac{\Delta}{8}$$

because we choose

$$\beta \leq \min\left\{\left(\frac{L\Delta\sqrt{n}}{32\sqrt{2}(1+b_1)\sigma\sqrt{T}\left(\sqrt{64L\Delta} + 3(B - \tau + b)\right)}\right)^{\frac{1-p}{2}}\right.$$

$$\left.\left(\frac{L\Delta\sqrt{n}}{96\sqrt{2}(1+b_1)\sigma\sqrt{T}a}\right)^{\frac{1-p}{2}}\right\}, \tag{45}$$

$$\text{and} \quad K + 1 \leq T. \tag{46}$$

This implies

$$\Pr\left(2\gamma(1-\beta)\left\|\sum_{t=0}^{K}\sum_{i=1}^{n}\langle\zeta_{5,i}^t, \theta^{t+1}\rangle\right\| \geq \frac{\Delta}{8}\right) \leq \frac{\alpha}{14(T+1)}.$$

Note that the worst dependency w.r.t $T$ is $\widetilde{\mathcal{O}}(1/T^{\frac{3(1-p)}{4}})$.

**Final probability.** Therefore, the probability event

$$\Omega := E_K \cap \overline{\Theta}^{K+1} \cap \left( \cap_{i=1}^n \overline{\Theta}_i^{K+1} \right) \cap \overline{N}^{K+1} \cap E_① \cap E_② \cap E_③ \cap E_④ \cap E_⑤ \cap E_⑥ \cap E_⑦ \cap E_⑧,$$

where each $E_①$-$E_⑧$ denotes that each of 1-8-th terms is smaller than $\frac{\Delta}{8}$ implies that

$$① + ② + ③ + ④ + ⑤ + ⑥ + ⑦ + ⑧ \leq 8 \cdot \frac{\Phi^0}{8} = \Delta,$$

i.e. 7 in the induction assumption holds. Moreover, this also implies that

$$\Phi^{K+1} \leq \Phi^0 + \Delta \leq \Delta + \Delta = 2\Delta,$$

i.e. 6 in the induction assumption holds. The probability $\Pr(E_{K+1})$ can be lower bounded as follows

$$\Pr(E_{K+1}) \geq \Pr(\Omega)$$
$$= \Pr\left( E_K \cap \overline{\Theta}^{K+1} \cap \left( \cap_{i=1}^n \overline{\Theta}_i^{K+1} \right) \cap \overline{N}^{K+1} \cap E_① \cap E_② \cap E_③ \cap E_④ \cap E_⑤ \cap E_⑥ \right.$$
$$\left. \cap E_⑦ \cap E_⑧ \right)$$
$$= 1 - \Pr\left( \overline{E}_K \cup \Theta^{K+1} \cup \left( \cup_{i=1}^n \Theta_i^{K+1} \right) \cup N^{K+1} \cup \overline{E}_① \cup \overline{E}_② \cup \overline{E}_③ \cup \overline{E}_④ \cup \overline{E}_⑤ \cup \overline{E}_⑥ \right.$$
$$\left. \cup \overline{E}_⑦ \cup \overline{E}_⑧ \right)$$
$$\geq 1 - \Pr(\overline{E}_K) - \Pr(\Theta^{K+1}) - \sum_{i=1}^n \Pr(\Theta_i^{K+1}) - \Pr(N^{K+1}) - \Pr(\overline{E}_①) - \Pr(\overline{E}_②)$$
$$- \Pr(\overline{E}_③) - \Pr(\overline{E}_④) - \Pr(\overline{E}_⑤) - \Pr(\overline{E}_⑥) - \Pr(\overline{E}_⑦) - \Pr(\overline{E}_⑧)$$
$$\geq 1 - \frac{\alpha(K+1)}{T+1} - \frac{\alpha}{6(T+1)} - \sum_{i=1}^n \frac{\alpha}{6n(T+1)} - \frac{\alpha}{6(T+1)} - 0 - 7 \cdot \frac{\alpha}{14(T+1)}$$
$$= 1 - \frac{\alpha(K+2)}{T+1}.$$

This finalizes the transition step of induction. The result of the theorem follows by setting $K = T - 1$. Indeed, from (37) we obtain

$$\frac{\gamma}{2} \sum_{t=0}^K \|\nabla f(x^t)\|^2 \leq \Phi^0 - \Phi^{K+1} + \Delta \leq 2\Delta \Rightarrow \frac{1}{T} \sum_{t=0}^{T-1} \|\nabla f(x^t)\|^2 \leq \frac{4\Delta}{\gamma T}. \qquad (47)$$

**Final rate.** Now we have the following restrictions on the momentum parameter in terms of dependency on $T$ from each bound of terms 1-8 correspondingly

$$\beta \leq \widetilde{\mathcal{O}} \left( \underbrace{\left( \frac{L\Delta n}{T\sigma^2} \right)^{\frac{1-p}{2-p}}}_{\text{from term 1}}, \underbrace{\left( \frac{L\Delta\sqrt{n}\eta}{B\sigma\sqrt{T}} \right)^{\frac{1-p}{2-p}}}_{\text{from term 2}}, \underbrace{\left( \frac{L\Delta\sqrt{n}\eta}{\sqrt{T}\sigma a} \right)^{\frac{1-p}{3-p}}}_{\text{from term 3}}, \underbrace{\left( \frac{L\Delta\eta\sqrt{n}}{\sigma\sqrt{T}a} \right)^{\frac{1-p}{2(2-p)}}}_{\text{from term 4}}, \right.$$

$$\left. \underbrace{\left( \frac{L\Delta\eta^2\sqrt{n}}{\sigma\sqrt{T}a} \right)^{\frac{1-p}{3-p}}}_{\text{from term 5}}, \underbrace{\left( \frac{L\Delta\eta^2\sqrt{n}}{\sigma\sqrt{T}a} \right)^{\frac{1-p}{2(2-p)}}}_{\text{from term 7}}, \underbrace{\left( \frac{L\Delta\sqrt{n}}{\sigma\sqrt{T}a} \right)^{\frac{1-p}{1+p}}}_{\text{from term 6}}, \underbrace{\left( \frac{L\Delta\sqrt{n}}{\sigma\sqrt{T}a} \right)^{\frac{1-p}{2}}}_{\text{from term 8}} \right).$$

Now we need to understand which stepsize restrictions give the worst $T$ complexity. We have

$$\gamma \leq \frac{1}{L} \widetilde{\mathcal{O}} \left( \underbrace{\left( \frac{L\Delta n}{T\sigma^2} \right)^{\frac{1}{2-p}}}_{\text{from term 1}}, \underbrace{\left( \frac{L\Delta\sqrt{n}\eta}{B\sigma\sqrt{T}} \right)^{\frac{1}{2-p}}}_{\text{from term 2}}, \underbrace{\left( \frac{L\Delta\sqrt{n}\eta}{\sqrt{T}\sigma a} \right)^{\frac{1}{3-p}}}_{\text{from term 3}}, \underbrace{\left( \frac{L\Delta\eta\sqrt{n}}{\sigma\sqrt{T}a} \right)^{\frac{1}{2(2-p)}}}_{\text{from term 4}}, \right.$$

$$\left. \underbrace{\left( \frac{L\Delta\eta^2\sqrt{n}}{\sigma\sqrt{T}a} \right)^{\frac{1}{3-p}}}_{\text{from term 5}}, \underbrace{\left( \frac{L\Delta\eta^2\sqrt{n}}{\sigma\sqrt{T}a} \right)^{\frac{1}{2(2-p)}}}_{\text{from term 7}}, \underbrace{\left( \frac{L\Delta\sqrt{n}}{\sigma\sqrt{T}a} \right)^{\frac{1}{1+p}}}_{\text{from term 6}}, \underbrace{\left( \frac{L\Delta\sqrt{n}}{\sigma\sqrt{T}a} \right)^{\frac{1}{2}}}_{\text{from term 8}} \right). \qquad (48)$$

By (28) we get that

$$
\gamma \leq \frac{1}{L} \widetilde{\mathcal{O}} \left( \underbrace{\left( \frac{L\Delta n}{T\sigma^2} \right)^{\frac{1}{2-p}}}_{\text{from term 1}}, \underbrace{\left( \frac{L\Delta\sqrt{n}\eta}{B\sigma\sqrt{T}} \right)^{\frac{1}{2-p}}}_{\text{from term 2}}, \underbrace{\left( \frac{L\Delta n\eta}{T\sigma\sqrt{d}\sigma_\omega} \right)^{\frac{1}{3-p}}}_{\text{from term 3}}, \underbrace{\left( \frac{L\Delta\eta n}{\sigma T\sqrt{d}\sigma_\omega} \right)^{\frac{1}{2(2-p)}}}_{\text{from term 4}}, \right.
$$

$$
\left. \underbrace{\left( \frac{L\Delta\eta^2 n}{\sigma T\sqrt{d}\sigma_\omega} \right)^{\frac{1}{3-p}}}_{\text{from term 5}}, \underbrace{\left( \frac{L\Delta\eta^2 n}{\sigma T\sqrt{d}\sigma_\omega} \right)^{\frac{1}{2(2-p)}}}_{\text{from term 7}}, \underbrace{\left( \frac{L\Delta n}{\sigma T\sqrt{d}\sigma_\omega} \right)^{\frac{1}{1+p}}}_{\text{from term 6}}, \underbrace{\left( \frac{L\Delta n}{\sigma T\sqrt{d}\sigma_\omega} \right)^{\frac{1}{2}}}_{\text{from term 8}} \right). \qquad (49)
$$

If we choose $p = 0.8$, then the worst power of $T$ comes from the term ① and equals to $\frac{1}{6}$. The second worst comes from the term ⑥ and equals to $\frac{4}{9}$. These two terms give the rate of the form

$$
\widetilde{\mathcal{O}} \left( \frac{L\Delta}{T} \left( \frac{T\sigma^2}{L\Delta n} \right)^{\frac{1}{2-p}} + \frac{L\Delta}{T} \left( \frac{\sigma T\sqrt{d}\sigma_\omega}{L\Delta n} \right)^{\frac{1}{1+p}} \right)
$$

$$
= \widetilde{\mathcal{O}} \left( \frac{(L\Delta)^{\frac{1-p}{2-p}}\sigma^{\frac{2}{2-p}}}{T^{\frac{1-p}{2-p}}n^{\frac{1}{2-p}}} + \frac{(L\Delta)^{\frac{p}{1+p}}\sigma^{\frac{1}{1+p}}d^{\frac{1}{2(1+p)}}\sigma_\omega^{\frac{1}{1+p}}}{T^{\frac{p}{1+p}}n^{\frac{1}{1+p}}} \right)
$$

$$
\stackrel{p=0.8}{=} \widetilde{\mathcal{O}} \left( \frac{(L\Delta)^{\frac{1}{6}}\sigma^{\frac{5}{3}}}{T^{\frac{1}{6}}n^{\frac{5}{6}}} + \frac{(L\Delta)^{\frac{4}{9}}\sigma^{\frac{5}{9}}d^{\frac{5}{18}}\sigma_\omega^{\frac{5}{9}}}{T^{\frac{4}{9}}n^{\frac{5}{9}}} \right).
$$

Besides, we have the momentum restriction of the form $\beta \leq \left( \frac{\tau}{48a} \right)^{1-p}$ that translates to

$$
\gamma \leq \widetilde{\mathcal{O}} \left( \frac{\tau}{La} \right),
$$

and therefore, gives an additional term in the rate of the form

$$
\widetilde{\mathcal{O}} \left( \frac{L\Delta}{T} \frac{\sqrt{d}\sqrt{T/n}\sigma_\omega}{\tau} \right) = \widetilde{\mathcal{O}} \left( \frac{L\Delta\sqrt{d}\sigma_\omega}{\sqrt{T}n\tau} \right).
$$

To conclude, we obtain with probability at least $1 - \alpha$ that

$$
\frac{1}{T} \sum_{t=0}^{T-1} \|\nabla f(x^t)\|^2 \leq \widetilde{\mathcal{O}} \left( \frac{L\Delta\sqrt{d}\sigma_\omega}{\sqrt{T}n\tau} + \frac{(L\Delta)^{\frac{1}{6}}\sigma^{\frac{5}{3}}}{T^{\frac{1}{6}}n^{\frac{5}{6}}} + \frac{(L\Delta)^{\frac{4}{9}}\sigma^{\frac{5}{9}}d^{\frac{5}{18}}\sigma_\omega^{\frac{5}{9}}}{T^{\frac{4}{9}}n^{\frac{5}{9}}} \right).
$$

CASE $\mathcal{I}_{K+1} = 0$. This case is even easier. The only change will be with the term next to $R^t$. We will get

$$
1 - \frac{48L^2}{\eta^2}\gamma^2 - \frac{12L^2}{\beta^2}\gamma^2 \geq \frac{2}{3} - \frac{48L^2}{\eta}\gamma^2 \geq 0
$$

instead of

$$
1 - \frac{16\beta^2 L^2}{\eta^2}\gamma^2 - \frac{48L^2}{\eta^2}\gamma^2 - \frac{12L^2}{\beta^2}\gamma^2 \geq 0
$$

as in the previous case. This difference comes from Lemma 18 because $\widetilde{V}^{K+1} = 0$. The rest is a repetition of the previous derivations.

$\square$

## F    PROOF OF COROLLARY 1

**Corollary 1.** Let Assumptions 1 and 2 hold and $\alpha \in (0, 1)$. Let $\Delta \geq \Phi^0$ and $\sigma_\omega$ be chosen as $\sigma_\omega = \Theta\left(\frac{\tau}{\varepsilon}\sqrt{T \log \frac{1}{\delta}}\right)$. Then there exists a stepsize $\gamma$ and momentum parameter $\beta$ such that the iterates of Clip21-SGDM (Algorithm 3) with probability at least $1 - \alpha$ satisfy local $(\varepsilon, \delta)$-DP and

$$\frac{1}{T}\sum_{t=0}^{T-1}\|\nabla f(x^t)\|^2 \leq \widetilde{\mathcal{O}}\left(\frac{L\Delta\sqrt{d}}{\sqrt{n}\varepsilon}\right), \tag{14}$$

where $\widetilde{\mathcal{O}}$ hides constant and logarithmic factors, and terms decreasing in $T$.

*Proof.* We need to plug in the value of $\sigma_\omega$ inside (13). Indeed, we have that

$$\widetilde{\mathcal{O}}\left(\frac{L\Delta\sqrt{d}}{\sqrt{T}n\tau}\frac{\tau}{\varepsilon}\sqrt{T} + \frac{(L\Delta)^{1/6}\sigma^{5/3}}{T^{1/6}n^{5/6}} + \frac{(L\Delta)^{4/9}\sigma^{5/9}d^{5/18}}{T^{4/9}n^{5/9}}\frac{\tau}{\varepsilon}\sqrt{T}\right)$$

$$= \widetilde{\mathcal{O}}\left(\frac{L\Delta\sqrt{d}}{\sqrt{T}n\tau}\frac{\tau}{\varepsilon}\sqrt{T} + \frac{(L\Delta)^{1/6}\sigma^{5/3}}{T^{1/6}n^{5/6}} + \frac{(L\Delta)^{4/9}\sigma^{5/9}d^{5/18}}{T^{4/9}n^{5/9}}\left(\frac{\tau}{\varepsilon}\sqrt{T}\right)^{5/9}\right)$$

$$= \widetilde{\mathcal{O}}\left(\frac{L\Delta\sqrt{d}}{\sqrt{n}\varepsilon} + \frac{(L\Delta)^{1/6}\sigma^{5/3}}{T^{1/6}n^{5/6}} + \frac{(L\Delta)^{4/9}\sigma^{5/9}d^{5/18}\tau^{5/9}}{T^{1/6}n^{5/9}\varepsilon^{5/9}}\right)/$$

Leaving only the terms that do not improve with $T$ we get the result.

$$\square$$

## G    PROOF OF THEOREM 3

We highlight that the proof of Theorem 3 mainly follows that of Theorem 4. The main difference comes from the fact that stepsize and momentum restrictions become less demanding as in purely stochastic setting (without DP noise) $a = 0$. In particular, we can choose $p = 1$. Therefore, we only list the modified lemmas without the proofs.

**Lemma 19.** Let each $f_i$ be $L$-smooth. Then we have the following inequality with probability 1

$$\|v_i^{t+1} - g_i^t\| \leq \max\left\{0, \|v_i^t - g_i^{t-1}\| - \tau\right\} + \beta L\gamma\|g^t\| + \beta\|\nabla f_i(x^t) - v_i^t\| + \beta\|\theta_i^{t+1}\|. \tag{50}$$

**Lemma 20.** Let each $f_i$ be $L$-smooth and $\Delta \geq \Phi^0$. Assume that the following inequalities hold

1. $g^0 = \frac{1}{n}\sum_{i=1}^n g_i^0$;

2. $\|g^{t-1}\| \leq \sqrt{64L\Delta} + 3(B - \tau) + 3b$;

3. $\|\nabla f_i(x^{t-1}) - v_i^{t-1}\| \leq \sqrt{4L\Delta} + \frac{3}{2}(B - \tau) + 2b$ for all $i \in [n]$;

4. $\|v_i^t - g_i^{t-1}\| \leq B$ for all $i \in [n]$;

5. $\gamma \leq \frac{1}{12L}$;

6. $\|\theta_i^t\| \leq b$ for all $i \in [n]$;

7. $1 \geq \beta \geq 4L\gamma$;

8. $\Phi^{t-1} \leq 2\Delta$.

Then we have

$$\|g^t\| \leq \sqrt{64L\Delta} + 3(B - \tau) + 3b. \tag{51}$$

**Lemma 21.** Let each $f_i$ is $L$-smooth and $\Delta \geq \Phi^0$. Assume the following inequalities hold

1. $\gamma \leq \frac{1}{12L}$;

2. $4L\gamma \leq \beta \leq 1$;

3. $\|\nabla f_i(x^{t-1}) - v_i^{t-1}\| \leq \sqrt{4L\Delta} + \frac{3}{2}(B - \tau) + 2b$;

4. $\|\theta_i^t\| \leq b$;

5. $\|g^{t-1}\| \leq \sqrt{64L\Delta} + 3(B - \tau) + 3b$.

Then we have

$$\|\nabla f_i(x^t) - v_i^t\| \leq \sqrt{4L\Delta} + \frac{3}{2}(B - \tau) + 2b. \tag{52}$$

**Lemma 22.** Let each $f_i$ be $L$-smooth, $\Delta \geq \Phi^0$, and $i \in \mathcal{I}_t$. Let the following inequalities hold

1. $12L\gamma \leq 1$;

2. $1 \geq \beta \geq 4L\gamma$;

3. $\beta \leq \frac{\tau}{32\sqrt{L\Delta}}$;

4. $\beta \leq \frac{\tau}{18(B-\tau)}$;

5. $\beta \leq \frac{\tau}{30b}$;

6. $\|g^t\| \leq \sqrt{64L\Delta} + 3(B - \tau) + 3b$;

7. $\|\theta_i^{t+1}\| \leq b$;

8. $\|\nabla f_i(x^t) - v_i^t\| \leq \sqrt{4L\Delta} + \frac{3}{2}(B - \tau) + 2b$.

Then

$$\|v_i^{t+1} - g_i^t\| \leq \|v_i^t - g_i^{t-1}\| - \frac{\tau}{2}. \tag{53}$$

**Lemma 23.** Let $\|\theta_i^{t+1}\| \leq b$ for all $i \in [n]$. Let each $f_i$ be $L$-smooth. Then $\widetilde{P}^t$ decreases as

$$\widetilde{P}^{t+1} \leq (1 - \beta)\widetilde{P}^t + \frac{3L^2}{\beta}R^t + \beta^2 b^2 + \frac{2}{n}\beta(1 - \beta)\sum_{i=1}^n \langle v_i^t - \nabla f_i(x^{t+1}), \theta_i^{t+1} \rangle. \tag{54}$$

Similarly, we can get the descent of $P^t$.

**Lemma 24.** Let $\|\theta^{t+1}\| \leq \frac{c}{\sqrt{n}}$, and each $f_i$ be $L$-smooth. Then $P^t$ decreases as

$$P^{t+1} \leq (1 - \beta)P^t + \frac{3L^2}{\beta}R^t + \beta^2 b^2 + 2\beta(1 - \beta)\langle v^t - \nabla f(x^{t+1}), \theta^{t+1} \rangle.$$

Now we present the descent of $\widetilde{V}^t$.

**Lemma 25.** Let $\|\theta_i^t\| \leq b$ for all $i \in [n]$, each $f_i$ be $L$-smooth, and $\|v_i^t - g_i^{t-1}\| \leq B$ for all $i \in [n]$. Then

$$\|g_i^t - v_i^t\|^2 \leq (1 - \eta)\|g_i^{t-1} - v_i^{t-1}\|^2 + \frac{4\beta^2}{\eta}\|v_i^{t-1} - \nabla f_i(x^{t-1})\|^2 + \frac{4\beta^2 L^2}{\eta}R^{t-1} + \beta^2 b^2 \tag{55}$$

$$+ 2(1 - \eta)^2\beta\langle(g_i^{t-1} - v_i^{t-1}) + \beta(v_i^{t-1} - \nabla f_i(x^{t-1})), \theta_i^t\rangle$$
$$+ 2(1 - \eta)^2\beta\langle\beta(\nabla f_i(x^{t-1}) - \nabla f_i(x^t)), \theta_i^t\rangle.$$

Moreover, averaging the inequalities across all $i \in [n]$ we get

$$\widetilde{V}^t \leq (1 - \eta)\widetilde{V}^{t-1} + \frac{4\beta^2}{\eta}\widetilde{P}^{t-1} + \frac{4\beta^2 L^2}{\eta}R^{t-1} + \beta^2 b^2 \tag{56}$$

$$+ \frac{2}{n}(1 - \eta)^2\beta\sum_{i=1}^n \langle(g_i^{t-1} - v_i^{t-1}) + \beta(v_i^{t-1} - \nabla f_i(x^{t-1})) + \beta(\nabla f_i(x^{t-1}) - \nabla f_i(x^t)), \theta_i^t\rangle.$$

**Theorem 7** (Full statement of Theorem 3). *Let Assumptions 1 and 2 hold, $B :=$ $\max_i\{\|\nabla f_i(x^0)\|\} + b > \tau$, probability constant $\alpha \in (0,1)$, and $\Delta \geq \Phi^0$. Let us run Algorithm 3 for $T$ iterations. Assume the following inequalities hold*

    *1.* **stepsize restrictions:**

        *i)* $12L\gamma \leq 1$;

        *ii)*

$$\frac{2}{3} - \frac{16\beta^2 L^2}{\eta^2}\gamma^2 - \frac{48L^2}{\eta^2}\gamma^2 \geq 0;$$

    *2.* **momentum restrictions:**

        *i)* $1 \geq \beta \geq 4L\gamma$;

        *ii)* $\beta \leq \frac{\tau}{32\sqrt{L\Delta}}$;

        *iii)* $\beta \leq \frac{\tau}{18(B-\tau)}$;

        *iv)* $\beta \leq \frac{\tau}{30b}$;

        *v)* *and momentum restrictions defined in (59), (60), (61), (62), (63), (65), (64), and (66);*

*Then with probability $1 - \alpha$ we have*

$$\frac{1}{T}\sum_{t=0}^{T-1}\|\nabla f(x^t)\|^2 \leq \widetilde{\mathcal{O}}\left(\frac{\sigma(\sqrt{L\Delta} + B + \sigma)}{\sqrt{Tn}}\right),$$

*where $\widetilde{\mathcal{O}}$ hides constant and logarithmic factors, and higher order terms decreasing in $T$.*

*Proof.* We prove the main theorem by induction. The conventional choice $\nabla f_i(x^{-1}, \xi_i^{-1}) = v_i^{-1} = g_i^{-1} = 0, \Phi^{-1} = \Phi^0$.

Let us define an event $E_t$ for each $t \in \{0, \ldots, T\}$ such that the following inequalities hold for all $k \in \{0, \ldots, t\}$

    1. $\|v_i^k - g_i^{k-1}\| \leq B$ for $i \in \mathcal{I}_k$;

    2. $\|g^k\| \leq \sqrt{64L\Delta} + 3(B - \tau) + 3b$;

    3. $\|v_i^k - \nabla f_i(x^k)\| \leq \sqrt{4L\Delta} + \frac{3}{2}(B - \tau) + 2b$;

    4. $\|\theta_i^k\| \leq b$ for all $i \in [n]$ and $\|\theta^k\| \leq \frac{c}{\sqrt{n}}$;

    5. $\Phi^k \leq 2\Delta$;

    6.

$$\Delta \geq \frac{2\gamma\beta}{n\eta}(1-\eta)^2 \sum_{l=0}^{k-1}\sum_{i=1}^{n}\langle(g_i^l - v_i^l) + \beta(v_i^l - \nabla f_i(x^l)) + \beta(\nabla f_i(x^l) - \nabla f_i(x^{l+1})), \theta_i^t\rangle$$

$$+ \frac{8\gamma\beta^2}{n\eta^2}(1-\beta)\sum_{l=0}^{k-1}\sum_{i=1}^{n}\langle v_i^l - \nabla f_i(x^l), \theta_i^{l+1}\rangle + 2\gamma(1-\beta)\sum_{l=0}^{k-1}\langle v^l - \nabla f(x^l), \theta^{l+1}\rangle$$

$$+ \frac{8\gamma\beta^2}{n\eta^2}(1-\beta)\sum_{l=0}^{k-1}\sum_{i=1}^{n}\langle\nabla f_i(x^l) - \nabla f_i(x^{l+1}), \theta_i^{l+1}\rangle$$

$$+ 2\gamma(1-\beta)\sum_{l=0}^{k-1}\langle\nabla f(x^l) - \nabla f(x^{l+1}), \theta^{l+1}\rangle.$$

Denote the events $\Theta_i^t$ and $\Theta^t$ as

$$\Theta_i^t := \{\|\theta_i^t\| \geq b\}, \quad \text{and} \quad \Theta^t := \{\|\theta^t\| \geq \frac{c}{\sqrt{n}}\} \tag{57}$$

respectively. From Assumption 2 we have

$$\Pr(\Theta_i^t) \leq 2\exp\left(-\frac{b^2}{2\sigma^2}\right) = \frac{\alpha}{6(T+1)n}$$

where the last equality is by definition of $b^2$. Therefore, $\Pr(\overline{\Theta}_i^t) \geq 1 - \frac{\alpha}{6(T+1)n}$.

Besides, notice that the constant $c$ in (28) can be viewed as

$$c = (\sqrt{2} + 2b_3)\sigma \quad \text{where} \quad b_3^2 = 3\log\frac{6(T+1)}{\alpha}.$$

Now we can use Lemma 1 to bound $\Pr(\Theta^t)$. Since all $\theta_i^t$ are independent $\sigma$-sub-Gaussian random vectors, then we have

$$\Pr\left(\left\|\sum_{i=1}^n \theta_i^t\right\| \geq c\sqrt{n}\right) = \Pr\left(\|\theta^t\| \geq \frac{c}{\sqrt{n}}\right) \leq \exp(-b_3^2/3) = \frac{\alpha}{6(T+1)}.$$

Now we prove that $\Pr(E_t) \geq 1 - \frac{\alpha(t+1)}{T+1}$ for all $t \in \{0, \ldots, T-1\}$. First, we show that the base of induction holds.

**Base of induction.**

1. $\|v_i^0 - g_i^{-1}\| = \|v_i^0\| = \beta\|\nabla f_i(x^0, \xi_i^0)\| = \beta\|\theta_i^0\| + \beta\|\nabla f_i(x^0)\| \leq \frac{1}{2}b + \frac{1}{2}B \leq \frac{1}{2}B + \frac{1}{2}B = B$ holds with probability $1 - \frac{\alpha}{6(T+1)}$. Indeed, we have

$$\Pr(\Theta_i^0) \leq 2\exp\left(-\frac{b^2}{2\sigma^2}\right) = \frac{\alpha}{6(T+1)n}.$$

   Therefore, we have

$$\Pr\left(\cap_{i=1}^n \overline{\Theta}_i^0\right) = 1 - \Pr\left(\cup_{i=1}^n \Theta_i^0\right) \geq 1 - \sum_{i=1}^n \Pr(\Theta_i^0) = 1 - n\frac{\alpha}{6(T+1)n} = 1 - \frac{\alpha}{6(T+1)}.$$

   Moreover, by concentration inequality we have

$$\Pr(\Theta^0) \leq \frac{\alpha}{6(T+1)}.$$

   This means that the probability of the event that each $\|\theta_i^0\| \leq b$, and $\|\theta^0\| \leq \frac{c}{\sqrt{n}}$, and is at least

$$1 - n\frac{\alpha}{6n(T+1)} - \frac{\alpha}{6(T+1)} = 1 - \frac{\alpha}{3(T+1)}.$$

2. $g^0 = \frac{1}{n}\sum_{i=1}^n(g_i^{-1} + \text{clip}_\tau(v_i^0 - g_i^{-1})) = \frac{1}{n}\sum_{i=1}^n \text{clip}_\tau(\beta\nabla f_i(x^0, \xi_i^0))$. Therefore, we have

$$\|g^0\| \leq \left\|\frac{1}{n}\sum_{i=1}^n \beta\nabla f_i(x^0) + \beta\theta_i^0 + (\text{clip}_\tau(\beta\nabla f_i(x^0, \xi_i^0)) - \beta\nabla f_i(x^0, \xi_i^0))\right\|$$

$$\leq \beta\|\nabla f(x^0)\| + \frac{\beta}{n}\sum_{i=1}^n \|\theta_i^0\| + \frac{1}{n}\sum_{i=1}^n \max\left\{0, \beta\|\nabla f_i(x^0, \xi_i^0)\| - \tau\right\}$$

$$\leq \beta\sqrt{2L(f(x^0) - f(x^*))} + \frac{\beta}{n}\sum_{i=1}^n \|\theta_i^0\| + \frac{1}{n}\sum_{i=1}^n \max\left\{0, \beta\|\nabla f_i(x^0)\| + \beta\|\theta_i^0\| - \tau\right\}$$

$$\leq \frac{1}{2}\sqrt{2L\Phi^0} + \frac{2\beta}{n}\sum_{i=1}^n \|\theta_i^0\| + \frac{\beta}{n}\sum_{i=1}^n \|\nabla f_i(x^0)\| - \tau$$

$$\leq \sqrt{64L\Delta} + 2\beta b + \beta B - \tau$$

$$\leq \sqrt{64L\Delta} + \frac{3}{2}B - \tau + b \leq \sqrt{64L\Delta} + 3(B - \tau) + 2b.$$

The inequalities above again hold in $\cap_{i=1}^n \overline{\Theta}_i^0$, i.e. with probability at least $1 - \frac{\alpha}{6(T+1)}$.

3. We have
$$\|v_i^0 - \nabla f_i(x^0)\| = \|\nabla f_i(x^0, \xi_i^0) - \nabla f_i(x^0)\| \le b.$$

This bound holds with probability at least $1 - \frac{\alpha}{6(T+1)}$ because it holds in $\cap_{i=1}^n \overline{\Theta}_i^0$.

4. Inequalities 5 obviously also hold, as $\Phi^0 \le 2\Phi^0 \le 2\Delta$ by the choice of $\Delta$.

Therefore, we conclude that the inequalities 1-7 hold with a probability at least

$$\Pr\left(\Theta^0 \cap \left(\cap_{i=1}^n \overline{\Theta}_i^0\right) \cap \overline{N}^t\right) \ge 1 - \Pr(\Theta^0) - \sum_{i=1}^n \Pr(\Theta_i^0) - \Pr(N^0)$$

$$\ge 1 - \frac{\alpha}{6(T+1)} - n \cdot \frac{\alpha}{6n(T+1)} - \frac{\alpha}{6(T+1)}$$

$$= 1 - \frac{\alpha}{2(T+1)} > 1 - \frac{\alpha}{T+1},$$

i.e. $\Pr(E_0) \ge 1 - \frac{\alpha}{T+1}$ holds. This is the base of the induction.

**Transition step of induction.**

CASE $|\mathcal{I}_{K+1}| > 0$. Assume that all events $\overline{\Theta}^{K+1}$ and $\overline{\Theta}_i^{K+1}$ take place, i.e. $\|\theta_i^{K+1}\| \le b, \|\theta^{K+1}\| \le \frac{c}{\sqrt{n}}$ for all $i \in [n]$. For that we need to work in $\overline{\Theta}^{K+1} \cap \left(\cap_{i=1}^n \overline{\Theta}_i^{K+1}\right)$. Then, by the assumptions of the induction, from Lemma 22 we get for all $i \in \mathcal{I}_{K+1}$

$$\|v_i^{K+1} - g_i^K\| \le \|v_i^K - g_i^{K-1}\| - \frac{\tau}{2} \le B - \frac{\tau}{2}.$$

Therefore, from Lemma 20 we get that
$$\|g^{K+1}\| \le \sqrt{64L\Delta} + 3(B - \tau) + 3b,$$

and from Lemma 21
$$\|\nabla f_i(x^{K+1}) - v_i^{K+1}\| \le \sqrt{4L\Delta} + \frac{3}{2}(B - \tau) + 2b.$$

This means that 1-3 in the induction assumption are also verified for the step $K + 1$.

Since we have for all $t \in \{0, \ldots, K+1\}$ that inequalities 1-3 are verified, then we can write for each $t \in \{0, K\}$ by Lemmas 2 and 23 to 25 the following

$$\Phi^{t+1} = \delta^{t+1} + \frac{\gamma}{\eta}\widetilde{V}^{t+1} + \frac{4\gamma\beta}{\eta^2}\widetilde{P}^{t+1} + \frac{\gamma}{\beta}P^{t+1}$$

$$\le \delta^t - \frac{\gamma}{2}\|\nabla f(x^t)\|^2 - \frac{1}{4\gamma}R^t + \gamma\widetilde{V}^t + \gamma P^t$$

$$+ \frac{\gamma}{\eta}\left((1-\eta)\widetilde{V}^t + \frac{4\beta^2}{\eta}\widetilde{P}^t + \frac{4\beta^2 L^2}{\eta}R^t + \beta^2 b^2\right.$$

$$+ \frac{2}{n}\beta(1-\eta^2)\sum_{i=1}^n \langle(g_i^t - v_i^t) + \beta(v_i^t - \nabla f_i(x^t)) + \beta(\nabla f_i(x^t) - \nabla f_i(x^{t+1})), \theta_i^{t+1}\rangle\Big)$$

$$+ \frac{4\gamma\beta}{\eta^2}\left((1-\beta)\widetilde{P}^t + \frac{3L^2}{\beta}R^t + \beta^2 b^2 + \frac{2}{n}\beta(1-\beta)\sum_{i=1}^n \langle v_i^t - \nabla f_i(x^{t+1}), \theta_i^{t+1}\rangle\right)$$

$$+ \frac{\gamma}{\beta}\left((1-\beta)P^t + \frac{3L^2}{\beta}R^t + \beta^2\frac{c^2}{n} + 2\beta(1-\beta)\langle v^t - \nabla f(x^{t+1}), \theta^{t+1}\rangle\right)$$

Rearranging terms we get

$$
\Phi^{t+1} \le \delta^t - \frac{\gamma}{2}\|\nabla f(x^t)\|^2 + \frac{\gamma}{\eta}\widetilde{V}^t(\eta + 1 - \eta) + \frac{4\gamma\beta}{\eta^2}\widetilde{P}^t(\beta + 1 - \beta) + \frac{\gamma}{\beta}P^t(\beta + 1 - \beta)
$$

$$
- \frac{1}{4\gamma}R^t\left(1 - \frac{16L^2\beta^2}{\eta^2}\gamma^2 - \frac{48L^2}{\eta^2}\gamma^2 - \frac{12L^2}{\beta^2}\gamma^2\right) + b^2\left(\frac{\beta^2\gamma}{\eta} + \frac{4\gamma\beta^3}{\eta^2}\right) + c^2\frac{\gamma\beta}{n}
$$

$$
+ \frac{2\gamma\beta}{n\eta}(1-\eta)^2\sum_{i=1}^n\langle(g_i^t - v_i^t) + \beta(v_i^t - \nabla f_i(x^t)) + \beta(\nabla f_i(x^t) - \nabla f_i(x^{t+1})), \theta_i^{t+1}\rangle
$$

$$
+ \frac{8\gamma\beta^2}{n\eta^2}(1-\beta)\sum_{i=1}^n\langle v_i^t - \nabla f_i(x^t), \theta_i^{t+1}\rangle + 2\gamma(1-\beta)\langle v^t - \nabla f(x^t), \theta^{t+1}\rangle
$$

$$
+ \frac{8\gamma\beta^2}{n\eta^2}(1-\beta)\sum_{i=1}^n\langle\nabla f_i(x^t) - \nabla f_i(x^{t+1}), \theta_i^{t+1}\rangle
$$

$$
+ 2\gamma(1-\beta)\langle\nabla f(x^t) - \nabla f(x^{t+1}), \theta^{t+1}\rangle.
$$

Using stepsize restriction $(vi)$ we get rid of the term with $R^t$ and obtain

$$
\Phi^{t+1} \le \Phi^t - \frac{\gamma}{2}\|\nabla f(x^t)\|^2 + b^2\left(\frac{\beta^2\gamma}{\eta} + \frac{4\gamma\beta^3}{\eta^2}\right) + c^2\frac{\gamma\beta}{n}
$$

$$
+ \frac{2\gamma\beta}{n\eta}(1-\eta)^2\sum_{i=1}^n\langle(g_i^t - v_i^t) + \beta(v_i^t - \nabla f_i(x^t)) + \beta(\nabla f_i(x^t) - \nabla f_i(x^{t+1})), \theta_i^{t+1}\rangle
$$

$$
+ \frac{8\gamma\beta^2}{n\eta^2}(1-\beta)\sum_{i=1}^n\langle v_i^t - \nabla f_i(x^t), \theta_i^{t+1}\rangle + 2\gamma(1-\beta)\langle v^t - \nabla f(x^t), \theta^{t+1}\rangle
$$

$$
+ \frac{8\gamma\beta^2}{n\eta^2}(1-\beta)\sum_{i=1}^n\langle\nabla f_i(x^t) - \nabla f_i(x^{t+1}), \theta_i^{t+1}\rangle
$$

$$
+ 2\gamma(1-\beta)\langle\nabla f(x^t) - \nabla f(x^{t+1}), \theta^{t+1}\rangle.
$$

Now we sum all the inequalities above for $t \in \{0, \ldots, K\}$ and get

$$
\Phi^{K+1} \le \Phi^0 - \frac{\gamma}{2}\sum_{t=0}^K\|\nabla f(x^t)\|^2 + Kb^2\left(\frac{\beta^2\gamma}{\eta} + \frac{4\gamma\beta^3}{\eta^2}\right) + Kc^2\frac{\gamma\beta}{n}
$$

$$
+ \frac{2\gamma\beta}{n\eta}(1-\eta)^2\sum_{t=0}^K\sum_{i=1}^n\langle(g_i^t - v_i^t) + \beta(v_i^t - \nabla f_i(x^t)) + \beta(\nabla f_i(x^t) - \nabla f_i(x^{t+1})), \theta_i^{t+1}\rangle
$$

$$
+ \frac{8\gamma\beta^2}{n\eta^2}(1-\beta)\sum_{t=0}^K\sum_{i=1}^n\langle v_i^t - \nabla f_i(x^t), \theta_i^{t+1}\rangle + 2\gamma(1-\beta)\sum_{t=0}^K\langle v^t - \nabla f(x^t), \theta^{t+1}\rangle
$$

$$
+ \frac{8\gamma\beta^2}{n\eta^2}(1-\beta)\sum_{t=0}^K\sum_{i=1}^n\langle\nabla f_i(x^t) - \nabla f_i(x^{t+1}), \theta_i^{t+1}\rangle
$$

$$
+ 2\gamma(1-\beta)\sum_{t=0}^K\langle\nabla f(x^t) - \nabla f(x^{t+1}), \theta^{t+1}\rangle. \tag{58}
$$

Rearranging terms we get

$$\frac{\gamma}{2} \sum_{t=0}^{K} \|\nabla f(x^t)\|^2 \le \Phi^0 - \Phi^{K+1} + Kb^2 \left(\frac{\beta^2\gamma}{\eta} + \frac{4\gamma\beta^3}{\eta^2}\right) + Kc^2 \frac{\gamma\beta}{n}$$

$$+ \frac{2\gamma\beta}{n\eta}(1-\eta)^2 \sum_{t=0}^{K}\sum_{i=1}^{n} \langle (g_i^t - v_i^t) + \beta(v_i^t - \nabla f_i(x^t)) + \beta(\nabla f_i(x^t) - \nabla f_i(x^{t+1})), \theta_i^{t+1}\rangle$$

$$+ \frac{8\gamma\beta^2}{n\eta^2}(1-\beta) \sum_{t=0}^{K}\sum_{i=1}^{n} \langle v_i^t - \nabla f_i(x^t), \theta_i^{t+1}\rangle + 2\gamma(1-\beta)\sum_{t=0}^{K} \langle v^t - \nabla f(x^t), \theta^{t+1}\rangle$$

$$+ \frac{8\gamma\beta^2}{n\eta^2}(1-\beta) \sum_{t=0}^{K}\sum_{i=1}^{n} \langle \nabla f_i(x^t) - \nabla f_i(x^{t+1}), \theta_i^{t+1}\rangle$$

$$+ 2\gamma(1-\beta)\sum_{t=0}^{K} \langle \nabla f(x^t) - \nabla f(x^{t+1}), \theta^{t+1}\rangle.$$

Taking into account that $\frac{\gamma}{2}\sum_{t=0}^{K}\|\nabla f(x^t)\|^2 \ge 0$, we get that the event $E_K \cap \left(\cap_{i=1}^{n}\overline{\Theta}_i^{K+1}\right) \cap \overline{N}^t$ implies

$$\Phi^{K+1} \le \Phi^0 + Kb^2 \left(\frac{\beta^2\gamma}{\eta} + \frac{4\gamma\beta^3}{\eta^2}\right) + Kc^2 \frac{\gamma\beta}{n}$$

$$+ \frac{2\gamma\beta}{n\eta}(1-\eta)^2 \sum_{t=0}^{K}\sum_{i=1}^{n} \langle (g_i^t - v_i^t) + \beta(v_i^t - \nabla f_i(x^t)) + \beta(\nabla f_i(x^t) - \nabla f_i(x^{t+1})), \theta_i^{t+1}\rangle$$

$$+ \frac{8\gamma\beta^2}{n\eta^2}(1-\beta) \sum_{t=0}^{K}\sum_{i=1}^{n} \langle v_i^t - \nabla f_i(x^t), \theta_i^{t+1}\rangle + \frac{2\gamma(1-\beta)}{n} \sum_{t=0}^{K}\sum_{i=1}^{n} \langle v^t - \nabla f(x^t), \theta_i^{t+1}\rangle$$

$$+ \frac{8\gamma\beta^2}{n\eta^2}(1-\beta) \sum_{t=0}^{K}\sum_{i=1}^{n} \langle \nabla f_i(x^t) - \nabla f_i(x^{t+1}), \theta_i^{t+1}\rangle$$

$$+ \frac{2\gamma(1-\beta)}{n} \sum_{t=0}^{K}\sum_{i=1}^{n} \langle \nabla f(x^t) - \nabla f(x^{t+1}), \theta_i^{t+1}\rangle.$$

Now we define the following random vectors:

$$\zeta_{1,i}^t := \begin{cases} g_i^t - v_i^t, & \text{if } \|g_i^t - v_i^t\| \le B \\ 0, & \text{otherwise} \end{cases},$$

$$\zeta_{2,t}^t := \begin{cases} v_i^t - \nabla f_i(x^t), & \text{if } \|v_i^t - \nabla f_i(x^t)\| \le \sqrt{4L\Delta} + \frac{3}{2}(B-\tau) + 2b \\ 0, & \text{otherwise} \end{cases},$$

$$\zeta_{3,i}^t := \begin{cases} \nabla f_i(x^t) - \nabla f_i(x^{t+1}), & \text{if } \|\nabla f_i(x^t) - \nabla f_i(x^{t+1})\| \le L\gamma\left(\sqrt{64L\Delta} + 3(B-\tau) + 3b\right) \\ 0, & \text{otherwise} \end{cases},$$

$$\zeta_4^t := \begin{cases} v^t - \nabla f(x^t), & \text{if } \|v^t - \nabla f(x^t)\| \le \sqrt{4L\Delta} + \frac{3}{2}(B-\tau) + 2ba \\ 0, & \text{otherwise} \end{cases},$$

$$\zeta_5^t := \begin{cases} \nabla f(x^t) - \nabla f(x^{t+1}), & \text{if } \|\nabla f(x^t) - \nabla f(x^{t+1})\| \le L\gamma\left(\sqrt{64L\Delta} + 3(B-\tau) + 3b\right) \\ 0, & \text{otherwise} \end{cases}.$$

By definition, all introduced random vectors $\zeta_{l,i}^t, l \in [3], i \in [n], \zeta_{4,5}^t$ are bounded with probability 1. Moreover, by the definition of $\Phi^t$ and definition of $E_t$ we get that the event $E_K \cap \overline{\Theta}^{K+1} \cap \left(\cap_{i=1}^{n}\overline{\Theta}_i^{K+1}\right) \cap \overline{N}^{K+1}$ implies

$$\zeta_{1,i}^t = g_i^t - v_i^t, \quad \zeta_{2,i}^t = v_i^t - \nabla f_i(x^t), \quad \zeta_{3,i}^t = \nabla f_i(x^t) - \nabla f_i(x^{t+1}),$$
$$\zeta_4^t = v^t - \nabla f(x^t), \quad \zeta_5^t = \nabla f(x^t) - \nabla f(x^{t+1}).$$

Therefore, the event $E_K \cap \overline{\Theta}^{K+1} \cap \left( \cap_{i=1}^n \overline{\Theta}_i^{K+1} \right) \cap \overline{N}^{K+1}$ implies

$$\Phi^{K+1} \le \Phi^0 + \underbrace{Kb^2 \left( \frac{\beta^2 \gamma}{\eta} + \frac{4\gamma \beta^3}{\eta^2} \right) + Kc^2 \frac{\gamma \beta}{n}}_{①} + \underbrace{\frac{2\gamma \beta}{n\eta}(1-\eta)^2 \sum_{t=0}^K \sum_{i=1}^n \langle \zeta_{1,i}^t, \theta_i^{t+1} \rangle}_{②}$$

$$+ \underbrace{\frac{2\gamma \beta^2}{n\eta}(1-\eta)^2 \sum_{t=0}^K \sum_{i=1}^n \langle \zeta_{2,i}^t, \theta_i^{t+1} \rangle}_{③} + \underbrace{\frac{2\gamma \beta^2}{n\eta}(1-\eta)^2 \sum_{t=0}^K \sum_{i=1}^n \langle \zeta_{3,i}^t, \theta_i^{t+1} \rangle}_{④}$$

$$+ \underbrace{\frac{8\gamma \beta^2}{n\eta^2}(1-\beta) \sum_{t=0}^K \sum_{i=1}^n \langle \zeta_{2,i}^t, \theta_i^{t+1} \rangle}_{⑤} + \underbrace{\frac{2\gamma(1-\beta)}{n} \sum_{t=0}^K \sum_{i=1}^n \langle \zeta_4^t, \theta_i^{t+1} \rangle}_{⑥}$$

$$+ \underbrace{\frac{8\gamma \beta^2}{n\eta^2}(1-\beta) \sum_{t=0}^K \sum_{i=1}^n \langle \zeta_{3,i}^t, \theta_i^{t+1} \rangle}_{⑦} + \underbrace{\frac{2\gamma(1-\beta)}{n} \sum_{t=0}^K \sum_{i=1}^n \langle \zeta_5^t, \theta_i^{t+1} \rangle}_{⑧}.$$

BOUND OF THE TERM ①.    For the term ① we have

$$Kb^2 \left( \frac{\beta^2 \gamma}{\eta} + \frac{4\gamma \beta^3}{\eta^2} \right) + Kc^2 \frac{\gamma \beta}{n} \le Kb^2 \left( \frac{\beta^3}{4L\eta} + \frac{\beta^4}{L\eta^2} \right) + Kc^2 \frac{\beta^2}{4Ln}.$$

By choosing $\gamma$ such that

$$\beta \le \min \left\{ \left( \frac{L\Delta\eta}{6Tb^2} \right)^{\frac{1}{3}}, \left( \frac{L\Delta\eta^2}{24Tb^2} \right)^{\frac{1}{4}}, \left( \frac{L\Delta n}{6Tc^2} \right)^{\frac{1}{2}} \right\} \tag{59}$$

we get that

$$Kb^2 \left( \frac{\beta^2 \gamma}{\eta} + \frac{4\gamma \beta^3}{\eta^2} \right) + Kc^2 \frac{\gamma \beta}{n} \le 3 \cdot \frac{\Delta}{24} = \frac{\Delta}{8}.$$

This bound holds with probability 1. Note that the worst dependency in the restriction on $\beta$ in $T$ is $\mathcal{O}(1/T^{\frac{1}{2}})$ that comes from the last term in $\min$.

BOUND OF THE TERM ②.    For term ②, let us enumerate random variables as

$$\langle \zeta_{1,1}^0, \theta_1^1 \rangle, \dots, \langle \zeta_{1,n}^0, \theta_n^1 \rangle, \langle \zeta_{1,1}^1, \theta_1^2 \rangle, \dots, \langle \zeta_{1,n}^1, \theta_n^2 \rangle, \dots \langle \zeta_{1,1}^K, \theta_1^{K+1} \rangle, \dots, \langle \zeta_{1,n}^K, \theta_n^{K+1} \rangle,$$

i.e. first by index $i$, then by index $t$. Then we have that the event $E_K \cap \left( \cap_{i=1}^n \overline{\Theta}_i^{K+1} \right)$ implies

$$\mathbb{E} \left[ \frac{2\gamma \beta}{n\eta}(1-\eta)^2 \langle \zeta_{1,i}^l, \theta_i^{l+1} \rangle \mid \langle \zeta_{1,i-1}^l, \theta_{i-1}^{l+1} \rangle, \dots, \langle \zeta_{1,1}^l, \theta_1^{l+1} \rangle, \dots, \langle \zeta_{1,1}^0, \theta_1^1 \rangle \right] = 0,$$

because $\{\theta_i^{l+1}\}_{i=1}^n$ are independent. Let

$$\sigma_2^2 := \frac{4\gamma^2 \beta^2}{n^2 \eta^2} \cdot B^2 \cdot \sigma^2.$$

Since $\theta_i^{l+1}$ is $\sigma$-sub-Gaussian random vector, we have

$$\mathbb{E}\left[\exp\left(\left|\frac{1}{\sigma_2^2}\frac{4\gamma^2\beta^2}{n^2\eta^2}(1-\eta)^4\langle\zeta_{1,i}^l,\theta_i^{l+1}\rangle^2\right|\right)\mid l, i-1\right]$$

$$\leq \mathbb{E}\left[\exp\left(\frac{1}{\sigma_1^2}\frac{4\gamma^2\beta^2}{n^2\eta^2}\|\zeta_{1,i}^l\|^2\cdot\|\theta_i^{l+1}\|^2\right)\mid l, i-1\right]$$

$$\leq \mathbb{E}\left[\exp\left(\frac{1}{\sigma_2^2}\frac{4\gamma^2\beta^2}{n^2\eta^2}\cdot B^2\|\theta_i^{l+1}\|^2\right)\mid l, i-1\right]$$

$$\leq \mathbb{E}\left[\exp\left(\frac{n^2\eta^2}{4\gamma^2\beta^2\cdot B^2\cdot\sigma^2}\frac{4\gamma^2\beta^2}{n^2\eta^2}\cdot B^2\|\theta_i^{l+1}\|^2\right)\mid l, i-1\right]$$

$$= \mathbb{E}\left[\exp\left(\frac{\|\theta_i^{l+1}\|^2}{\sigma^2}\mid l, i-1\right)\right]\leq \exp(1).$$

Here $\mathbb{E}\left[\cdot\mid l, i-1\right]$ means

$$\mathbb{E}\left[\cdot\mid \langle\zeta_{1,i-1}^l,\theta_{i-1}^{l+1}\rangle,\ldots,\langle\zeta_{1,1}^l,\theta_1^{l+1}\rangle,\ldots,\langle\zeta_{1,1}^0,\theta_1^1\rangle\right]=0,$$

Therefore, we have by Lemma 1 with $\sigma_k^2\equiv\sigma_2^2$ that

$$\Pr\left(\frac{2\gamma\beta}{n\eta}(1-\eta)^2\left\|\sum_{t=0}^K\sum_{i=1}^n\langle\zeta_{1,i}^t,\theta_i^{t+1}\rangle\right\|\geq(\sqrt{2}+\sqrt{2}b_1)\sqrt{\sum_{t=0}^K\sum_{i=1}^n\frac{4B^2\gamma^2\beta^2\sigma^2}{n^2\eta^2}}\right)$$

$$\leq \exp(-b_1^2/3)$$

$$= \frac{\alpha}{14(T+1)}$$

with $b_1^2=3\log\left(\frac{14(T+1)}{\alpha}\right)$. Note that

$$(\sqrt{2}+\sqrt{2}b_1)\sqrt{\sum_{t=0}^K\sum_{i=1}^n\frac{4B^2\gamma^2\beta^2\sigma^2}{n^2\eta^2}}\leq(\sqrt{2}+\sqrt{2}b_1)\sqrt{\sum_{t=0}^K\sum_{i=1}^n\frac{B^2\beta^4\sigma^2}{4L^2n^2\eta^2}}$$

$$= (\sqrt{2}+\sqrt{2}b_1)\frac{B\beta^2\sigma}{2Ln\eta}\sqrt{(K+1)n}$$

$$\leq \frac{\Delta}{8},$$

because we choose

$$\beta\leq\left(\frac{L\Delta\sqrt{n}\eta}{4\sqrt{2}(1+b_1)B\sigma\sqrt{T}}\right)^{\frac{1}{2}},\quad\text{and}\quad K+1\leq T.\tag{60}$$

This implies that

$$\Pr\left(\frac{2\gamma\beta}{n\eta}(1-\eta)^2\left\|\sum_{t=0}^K\sum_{i=1}^n\langle\zeta_{1,i}^t,\theta_i^{t+1}\rangle\right\|\geq\frac{\Delta}{8}\right)\leq\frac{\alpha}{14(T+1)}$$

with this choice of momentum parameter. The dependency on $T$ is $\widetilde{\mathcal{O}}(1/T^{\frac{1}{4}})$.

BOUND OF THE TERM ③.  The bound in this case is similar to the previous one. Let

$$\sigma_3^2 := \frac{4\gamma^2\beta^4}{n^2\eta^2}\cdot\left(\sqrt{4L\Delta}+\frac{3}{2}(B-\tau)+2b\right)^2\cdot\sigma^2.$$

$$\mathbb{E}\left[\exp\left(\left|\frac{1}{\sigma_3^2}\frac{4\gamma^2\beta^4}{n^2\eta^2}(1-\eta)^4\langle\zeta_{2,i}^l,\theta_i^{l+1}\rangle^2\right|\right)\mid l,i-1\right]$$

$$\leq\mathbb{E}\left[\exp\left(\frac{1}{\sigma_2^2}\frac{4\gamma^2\beta^4}{n^2\eta^2}\|\zeta_{2,i}^l\|^2\cdot\|\theta_i^{l+1}\|^2\right)\right]$$

$$\leq\mathbb{E}\left[\exp\left(\frac{1}{\sigma_2^3}\frac{4\gamma^2\beta^4}{n^2\eta^2}\cdot\left(\sqrt{4L\Delta}+\frac{3}{2}(B-\tau)+2b\right)^2\cdot\|\theta_i^{l+1}\|^2\right)\mid l,i-1\right]$$

$$\leq\mathbb{E}\left[\exp\left(\left[\frac{4\gamma^2\beta^4}{n^2\eta^2}\cdot\left(\sqrt{4L\Delta}+\frac{3}{2}(B-\tau)+2b\right)^2\cdot\sigma^2\right]^{-1}\cdot\right.\right.$$
$$\left.\left.\frac{4\gamma^2\beta^4}{n^2\eta^2}\cdot\left(\sqrt{4L\Delta}+\frac{3}{2}(B-\tau)+2b\right)^2\cdot\|\theta_i^{l+1}\|^2\right)\mid l,i-1\right]$$

$$=\mathbb{E}\left[\exp\left(\frac{\|\theta_i^{l+1}\|^2}{\sigma^2}\right)\mid l,i-1\right]\leq\exp(1).$$

Therefore, we have by Lemma 1 that

$$\Pr\left[\frac{2\gamma\beta^2}{n\eta}(1-\eta)^2\left\|\sum_{t=0}^{K}\sum_{i=1}^{n}\langle\zeta_{2,i}^t,\theta_i^{t+1}\rangle\right\|\right.$$
$$\left.\geq(\sqrt{2}+\sqrt{2}b_1)\sqrt{\sum_{t=0}^{K}\sum_{i=1}^{n}\frac{4\gamma^2\beta^4\sigma^2}{n^2\eta^2}\cdot\left(\sqrt{4L\Delta}+\frac{3}{2}(B-\tau)+2b\right)^2}\right]$$
$$\leq\exp(-b_1^2/3)=\frac{\alpha}{14(T+1)},$$

Note that

$$(\sqrt{2}+\sqrt{2}b_1)\sqrt{(K+1)n}\frac{2\gamma\beta^2\sigma}{\eta n}\left(\sqrt{4L\Delta}+\frac{3}{2}(B-\tau)+2b\right)$$

$$\leq(\sqrt{2}+\sqrt{2}b_1)\sqrt{(K+1)n}\frac{\beta^3\sigma}{2L\eta n}\left(\sqrt{4L\Delta}+\frac{3}{2}(B-\tau)+2b\right)$$

$$\leq\frac{\Delta}{8}.$$

because we choose

$$\beta\leq\left(\frac{L\Delta\eta\sqrt{n}}{4\sqrt{2}(1+b_1)\sigma\sqrt{T}\left(\sqrt{4L\Delta}+\frac{3}{2}(B-\tau)+2b\right)}\right)^{\frac{1}{3}}\quad\text{and}\quad K+1\leq T. \quad(61)$$

This implies

$$\Pr\left(\frac{2\gamma\beta^2}{n\eta}(1-\eta)^2\left\|\sum_{t=0}^{K}\sum_{i=1}^{n}\langle\zeta_{2,i}^t,\theta_i^{t+1}\rangle\right\|\geq\frac{\Delta}{8}\right)\leq\frac{\alpha}{14(T+1)}.$$

Note that the worst dependency w.r.t. $T$ is $\widetilde{\mathcal{O}}(1/T^{\frac{1}{6}})$.

BOUND OF THE TERM ④.  The bound in this case is similar to the previous one. Let

$$\sigma_4^2:=\frac{4L^2\gamma^4\beta^4}{n^2\eta^2}\left(\sqrt{64L\Delta}+3(B-\tau)+3b\right)^2\cdot\sigma^2.$$

Then we have

$$
\mathbb{E}\left[\exp\left(\left|\frac{1}{\sigma_4^2}\frac{4\gamma^2\beta^4}{n^2\eta^2}(1-\eta)^4\langle\zeta_{3,i}^l,\theta_i^{l+1}\rangle^2\right|\right)\mid l,i-1\right]
$$

$$
\leq \mathbb{E}\left[\exp\left(\frac{1}{\sigma_4^2}\frac{4\gamma^2\beta^4}{n^2\eta^2}\|\zeta_{3,i}^l\|^2\cdot\|\theta_i^{l+1}\|^2\right)\mid l,i-1\right]
$$

$$
\leq \mathbb{E}\left[\exp\left(\frac{1}{\sigma_4^2}\frac{4\gamma^2\beta^4}{n^2\eta^2}\cdot L^2\gamma^2\left(\sqrt{64L\Delta}+3(B-\tau)+3b+3a\right)^2\cdot\|\theta_i^{l+1}\|^2\right)\mid l,i-1\right]
$$

$$
\leq \mathbb{E}\left[\exp\left(\left[\frac{4L^2\gamma^4\beta^4}{n^2\eta^2}\left(\sqrt{64L\Delta}+3(B-\tau)+3b+3a\right)^2\cdot\sigma^2\right]^{-1}\right.\right.
$$

$$
\left.\left.\frac{4L^2\gamma^4\beta^4}{n^2\eta^2}\cdot\frac{16\beta^4}{81}\left(\sqrt{64L\Delta}+3(B-\tau)+3b+3a\right)^2\cdot\|\theta_i^{l+1}\|^2\right)\mid l,i-1\right]
$$

$$
= \mathbb{E}\left[\exp\left(\frac{\|\theta_i^{l+1}\|^2}{\sigma^2}\right)\right]\leq\exp(1).
$$

Therefore, we have by Lemma 1 that

$$
\Pr\left(\frac{2\gamma\beta^2}{n\eta}(1-\eta)^2\left\|\sum_{t=0}^K\sum_{i=1}^n\langle\zeta_{3,i}^t,\theta_i^{t+1}\rangle\right\|\right.
$$

$$
\geq (\sqrt{2}+\sqrt{2}b_1)\sqrt{\sum_{t=0}^K\sum_{i=1}^n\frac{4L^2\gamma^4\beta^4\sigma^2}{n^2\eta^2}\cdot\left(\sqrt{64L\Delta}+3(B-\tau+b)\right)^2}\right)
$$

$$
\leq \exp(-b_1^2/3) = \frac{\alpha}{14(T+1)}.
$$

Note that

$$
(\sqrt{2}+\sqrt{2}b_1)\sqrt{(K+1)n}\frac{2L\gamma^2\beta^2\sigma}{\eta n}\left(\sqrt{64L\Delta}+3(B-\tau+b)+3a\right)
$$

$$
\leq\sqrt{2}(1+b_1)\sqrt{(K+1)n}\frac{\beta^4\sigma}{8L\eta n}\left(\sqrt{64L\Delta}+3(B-\tau+b)\right)
$$

$$
\leq\frac{\Delta}{8}.
$$

because we choose

$$
\beta\leq\left(\frac{L\Delta\eta\sqrt{n}}{\sqrt{2}(1+b_1)\sigma\sqrt{T}\left(\sqrt{64L\Delta}+3(B-\tau+b)\right)}\right)^{\frac{1}{4}},\quad\text{and}\quad K+1\leq T.\tag{62}
$$

This implies

$$
\Pr\left(\frac{2\gamma\beta^2}{n\eta}(1-\eta)^2\left\|\sum_{t=0}^K\sum_{i=1}^n\langle\zeta_{2,i}^t,\theta_i^{t+1}\rangle\right\|\geq\frac{\Delta}{8}\right)\leq\frac{\alpha}{14(T+1)},
$$

Note that the worst dependency w.r.t. $T$ is $\widetilde{\mathcal{O}}(1/T^{\frac{1}{8}})$.

BOUND OF THE TERM ⑤. The bound in this case is similar to the previous one. Let

$$
\sigma_5^2 := \frac{64\gamma^2\beta^4}{n^2\eta^4}\cdot\left(\sqrt{4L\Delta}+\frac{3}{2}(B-\tau)+2b\right)^2\cdot\sigma^2.
$$

Then we have

$$\mathbb{E}\left[\exp\left(\left|\frac{1}{\sigma_5^2}\frac{64\gamma^2\beta^4}{n^2\eta^4}(1-\beta)^2\langle\zeta_{2,i}^l,\theta_i^{l+1}\rangle^2\right|\right)\mid l,i-1\right]$$

$$\leq\mathbb{E}\left[\exp\left(\frac{1}{\sigma_5^2}\frac{64\gamma^2\beta^4}{n^2\eta^4}\|\zeta_{2,i}^l\|^2\cdot\|\theta_i^{l+1}\|^2\right)\mid l,i-1\right]$$

$$\leq\mathbb{E}\left[\exp\left(\frac{1}{\sigma_5^2}\frac{64\gamma^2\beta^4}{n^2\eta^4}\cdot\left(\sqrt{4L\Delta}+\frac{3}{2}(B-\tau)+2b\right)^2\cdot\|\theta_i^{l+1}\|^2\right)\mid l,i-1\right]$$

$$=\mathbb{E}\left[\exp\left(\left[\frac{64\gamma^2\beta^4}{81L^2n^2\eta^4}\cdot\left(\sqrt{4L\Delta}+\frac{3}{2}(B-\tau)+2b\right)^2\cdot\sigma^2\right]^{-1}\right.\right.$$

$$\left.\left.\frac{64\gamma^2\beta^4}{n^2\eta^4}\cdot\left(\sqrt{4L\Delta}+\frac{3}{2}(B-\tau)+2b\right)^2\cdot\|\theta_i^{l+1}\|^2\right)\mid l,i-1\right]$$

$$=\mathbb{E}\left[\exp\left(\frac{\|\theta_i^{l+1}\|^2}{\sigma^2}\right)\mid l,i-1\right]\leq\exp(1).$$

Therefore, we have by Lemma 1 that

$$\Pr\left[\frac{8\gamma\beta^2}{n\eta^2}(1-\beta)\left\|\sum_{t=0}^K\sum_{i=1}^n\langle\zeta_{2,i}^t,\theta_i^{t+1}\rangle\right\|\right.$$

$$\left.\geq(\sqrt{2}+\sqrt{2}b_1)\sqrt{\sum_{t=0}^K\sum_{i=1}^n\frac{64\gamma^2\beta^4\sigma^2}{n^2\eta^4}\left(\sqrt{4L\Delta}+\frac{3}{2}(B-\tau)+2b+(L\gamma)^pa\right)^2}\right]$$

$$\leq\exp(-b_1^2/3)=\frac{\alpha}{14(T+1)}.$$

Note that

$$(\sqrt{2}+\sqrt{2}b_1)\sqrt{(K+1)n}\frac{8\gamma\beta^2\sigma}{n\eta^2}\left(\sqrt{4L\Delta}+\frac{3}{2}(B-\tau)+2b\right)$$

$$\leq(\sqrt{2}+\sqrt{2}b_1)\sqrt{(K+1)n}\frac{2\beta^3\sigma}{Ln\eta^2}\left(\sqrt{4L\Delta}+\frac{3}{2}(B-\tau)+2b\right)$$

$$\leq\frac{\Delta}{8}$$

because we choose

$$\beta\leq\left(\frac{L\Delta\eta^2\sqrt{n}}{16\sqrt{2}(1+b_1)\sigma\sqrt{T}\left(\sqrt{4L\Delta}+\frac{3}{2}(B-\tau)+2b\right)}\right)^{\frac{1}{3}}\quad\text{and}\quad K+1\leq T.\quad(63)$$

This implies

$$\Pr\left(\frac{8\gamma\beta^2}{n\eta^2}(1-\beta)\left\|\sum_{t=0}^K\sum_{i=1}^n\langle\zeta_{2,i}^t,\theta_i^{t+1}\rangle\right\|\geq\frac{\Delta}{8}\right)\leq\frac{\alpha}{14(T+1)}.$$

Note that the worst dependency w.r.t. $T$ is $\widetilde{\mathcal{O}}(1/T^{\frac{1}{6}})$.

BOUND OF THE TERM ⑦.   The bound in this case is similar to the previous one. Let

$$\sigma_7^2:=\frac{64\gamma^2\beta^4}{n^2\eta^4}\left(\sqrt{64L\Delta}+3(B-\tau+b)\right)^2\cdot\sigma^2.$$

Then we have

$$\mathbb{E}\left[\exp\left(\left|\frac{1}{\sigma_7^2}\frac{64L^2\gamma^4\beta^4}{n^2\eta^4}(1-\beta)^2\langle\zeta_{3,i}^l,\theta_i^{l+1}\rangle^2\right|\right)\mid l,i-1\right]$$

$$\leq \mathbb{E}\left[\exp\left(\frac{1}{\sigma_7^2}\frac{64\gamma^2\beta^4}{n^2\eta^4}\|\zeta_{3,i}^l\|^2\cdot\|\theta_i^{l+1}\|^2\right)\mid l,i-1\right]$$

$$\leq \mathbb{E}\left[\exp\left(\frac{64\gamma^2\beta^4}{n^2\eta^4}\cdot L^2\gamma^2\left(\sqrt{64L\Delta}+3(B-\tau+b)\right)^2\cdot\|\theta_i^{l+1}\|^2\right)\mid l,i-1\right]$$

$$\leq \mathbb{E}\left[\exp\left(\left[\frac{64L^2\gamma^4\beta^4}{n^2\eta^4}\left(\sqrt{64L\Delta}+3(B-\tau+b)\right)^2\cdot\sigma^2\right]^{-1}\right.\right.$$

$$\left.\left.\frac{64L^2\gamma^4\beta^4}{n^2\eta^4}\left(\sqrt{64L\Delta}+3(B-\tau+b)\right)^2\cdot\|\theta_i^{l+1}\|^2\right)\mid l,i-1\right]$$

$$= \mathbb{E}\left[\exp\left(\frac{\|\theta_i^{l+1}\|^2}{\sigma^2}\right)\mid l,i-1\right]\leq\exp(1).$$

Therefore, we have by Lemma 1 that

$$\Pr\left[\frac{8\gamma\beta^2}{n\eta^2}(1-\beta)\left\|\sum_{t=0}^{K}\sum_{i=1}^{n}\langle\zeta_{3,i}^t,\theta_i^{t+1}\rangle\right\|\geq\right.$$

$$\left.(\sqrt{2}+\sqrt{2}b_1)\sqrt{\sum_{t=0}^{K}\sum_{i=1}^{n}\frac{64L^2\gamma^4\beta^4\sigma^2}{n^2\eta^4}\cdot\left(\sqrt{64L\Delta}+3(B-\tau+b)\right)^2}\right]$$

$$\leq \exp(-b_1^2/3)=\frac{\alpha}{14(T+1)}.$$

Note that

$$(\sqrt{2}+\sqrt{2}b_1)\sqrt{(K+1)n}\frac{8L\gamma^2\beta^2\sigma}{\eta^2 n}\left(\sqrt{64L\Delta}+3(B-\tau+b)\right)$$

$$\leq(\sqrt{2}+\sqrt{2}b_1)\sqrt{(K+1)n}\frac{\beta^4\sigma}{2L\eta^2 n}\left(\sqrt{64L\Delta}+3(B-\tau+b)\right)$$

$$\leq\frac{\Delta}{8}$$

because we choose

$$\beta\leq\left(\frac{L\Delta\eta^2\sqrt{n}}{4\sqrt{2}(1+b_1)\sigma\sqrt{T}\left(\sqrt{64L\Delta}+3(B-\tau+b)\right)}\right)^{\frac{1}{4}}\quad\text{and}\quad K+1\leq T.\qquad(64)$$

This implies

$$\Pr\left(\frac{8\gamma\beta^2}{n\eta^2}(1-\beta)\left\|\sum_{t=0}^{K}\sum_{i=1}^{n}\langle\zeta_{3,i}^t,\theta_i^{t+1}\rangle\right\|\geq\frac{\Delta}{8}\right)\leq\frac{\alpha}{14(T+1)}.$$

Note that the worst dependency w.r.t. $T$ is $\widetilde{\mathcal{O}}(1/T^{\frac{1}{8}})$.

BOUND OF THE TERM ⑥.   The bound in this case is similar to the previous one. Let

$$\sigma_6^2 := \frac{4\gamma^2}{n^2}\left(\sqrt{4L\Delta}+\frac{3}{2}(B-\tau)+2b\right)^2\cdot\sigma^2.$$

Then we have

$$\mathbb{E}\left[\exp\left(\left|\frac{1}{\sigma_6^2}\frac{4\gamma^2}{n^2}(1-\beta)^2\langle\zeta_4^l,\theta_i^{l+1}\rangle^2\right|\right)\mid l,i-1\right]$$

$$\leq \mathbb{E}\left[\exp\left(\frac{1}{\sigma_6^2}\frac{4\gamma^2}{n^2}\|\zeta_4^l\|^2\cdot\|\theta_i^{l+1}\|^2\right)\mid l,i-1\right]$$

$$\leq \mathbb{E}\left[\exp\left(\frac{1}{\sigma_6^2}\frac{4\gamma^2}{n^2}\left(\sqrt{4L\Delta}+\frac{3}{2}(B-\tau)+2b\right)^2\cdot\|\theta_i^{l+1}\|^2\right)\mid l,i-1\right]$$

$$\leq \mathbb{E}\left[\exp\left(\left[\frac{4\gamma^2}{n^2}\left(\sqrt{4L\Delta}+\frac{3}{2}(B-\tau)+2b\right)^2\cdot\sigma^2\right]^{-1}\right.\right.$$

$$\left.\left.\frac{4\gamma^2}{n^2}\left(\sqrt{4L\Delta}+\frac{3}{2}(B-\tau)+2b\right)^2\cdot\|\theta_i^{l+1}\|^2\right)\mid l,i-1\right]$$

$$= \mathbb{E}\left[\exp\left(\frac{\|\theta_i^{t+1}\|^2}{\sigma^2}\right)\mid l,i-1\right]\leq\exp(1).$$

Therefore, we have by Lemma 1 that

$$\Pr\left[\frac{2\gamma(1-\beta)}{n}\left\|\sum_{t=0}^{K}\sum_{i=1}^{n}\langle\zeta_{4,i}^t,\theta_i^{t+1}\rangle\right\|\right.$$

$$\geq (\sqrt{2}+\sqrt{2}b_1)\sqrt{\sum_{t=0}^{K}\sum_{i=1}^{n}\frac{4\gamma^2}{n^2}\sigma^2\cdot\left(\sqrt{4L\Delta}+\frac{3}{2}(B-\tau)+2b\right)^2}\right]$$

$$\leq \exp(-b_1^2/3)=\frac{\alpha}{14(T+1)},$$

Note that

$$(\sqrt{2}+\sqrt{2}b_1)\sqrt{(K+1)n}\cdot\frac{2\gamma}{n}\sigma\left(\sqrt{4L\Delta}+\frac{3}{2}(B-\tau)+2b\right)$$

$$\leq(\sqrt{2}+\sqrt{2}b_1)\sqrt{(K+1)n}\cdot\frac{\beta\sigma}{2Ln}\left(\sqrt{4L\Delta}+\frac{3}{2}(B-\tau)+2b\right)$$

$$\leq\frac{\Delta}{8}$$

because we choose

$$\beta\leq\left(\frac{L\Delta\sqrt{n}}{32\sqrt{2}(1+b_1)\sigma\sqrt{T}\left(\sqrt{4L\Delta}+\frac{3}{2}(B-\tau)+2b\right)}\right)\quad\text{and}\quad K+1\leq T.\quad(65)$$

This implies

$$\Pr\left(\frac{2\gamma(1-\beta)}{n}\left\|\sum_{t=0}^{K}\sum_{i=1}^{n}\langle\zeta_{4,i}^t,\theta_i^{t+1}\rangle\right\|\geq\frac{\Delta}{8}\right)\leq\frac{\alpha}{14(T+1)}.$$

Note that the worst dependency w.r.t. $T$ is $\widetilde{\mathcal{O}}(1/T^{1/2})$.

BOUND OF THE TERM ⑧. The bound in this case is similar to the previous one. Let

$$\sigma_8^2:=\frac{4L^2\gamma^4}{n^2}\cdot\left(\sqrt{64L\Delta}+3(B-\tau+b)\right)^2\cdot\sigma^2.$$

Then we have

$$\mathbb{E}\left[\exp\left(\left|\frac{1}{\sigma_8^2}\frac{4\gamma^2}{n^2}(1-\beta)^2\langle\zeta_5^l,\theta_i^{l+1}\rangle^2\right|\right)\mid l,i-1\right]$$

$$\leq\mathbb{E}\left[\exp\left(\frac{1}{\sigma_8^2}\frac{4\gamma^2}{n^2}\|\zeta_5^l\|^2\cdot\|\theta_i^{l+1}\|^2\right)\mid l,i-1\right]$$

$$\leq\mathbb{E}\left[\exp\left(\frac{1}{\sigma_8^2}\frac{4\gamma^2}{n^2}L^2\gamma^2\left(\sqrt{64L\Delta}+3(B-\tau+b)\right)\cdot\|\theta_i^{l+1}\|^2\right)^2\mid l,i-1\right].$$

Since $\theta_i^{l+1}$ is sub-Gaussian with parameter $\sigma^2$, then we can continue the chain of inequalities above using the definition of $\sigma_8^2$

$$\mathbb{E}\left[\exp\left(\left[\frac{4L^2\gamma^4}{n^2}\cdot\left(\sqrt{64L\Delta}+3(B-\tau+b)+3a\right)^2\cdot\sigma^2\right]^{-1}\right.\right.$$

$$\left.\left.\frac{4L^2\gamma^4}{n^2}\cdot\left(\sqrt{64L\Delta}+3(B-\tau+b)\right)^2\cdot\|\theta_i^{l+1}\|^2\right)\mid l,i-1\right]$$

$$=\mathbb{E}\left[\exp\left(\frac{\|\theta_i^{l+1}\|^2}{\sigma^2}\right)\right]\leq\exp(1).$$

Therefore, we have by Lemma 1 that

$$\Pr\left[\frac{2\gamma(1-\beta)}{n}\left\|\sum_{t=0}^K\sum_{i=1}^n\langle\zeta_{5,i}^t,\theta^{t+1}\rangle\right\|\right.$$

$$\left.\geq(\sqrt{2}+\sqrt{2}b_1)\sqrt{\sum_{t=0}^K\sum_{i=1}^n\frac{4L^2\gamma^4}{n^2}\sigma^2\cdot\left(\sqrt{64L\Delta}+3(B-\tau+b)\right)^2}\right]$$

$$\leq\exp(-b_1^2/3)=\frac{\alpha}{14(T+1)}.$$

Note that

$$(\sqrt{2}+\sqrt{2}b_1)\sqrt{(K+1)n}\cdot\frac{2L\gamma^2}{n}\sigma\left(\sqrt{64L\Delta}+3(B-\tau+b)\right)$$

$$\leq(\sqrt{2}+\sqrt{2}b_1)\sqrt{(K+1)n}\cdot\frac{\beta^2}{2Ln}\sigma\left(\sqrt{64L\Delta}+3(B-\tau+b)\right)$$

$$\leq\frac{\Delta}{8}$$

because we choose

$$\beta\leq\left(\frac{L\Delta\sqrt{n}}{4\sqrt{2}(1+b_1)\sigma\sqrt{T}\left(\sqrt{64L\Delta}+3(B-\tau+b)\right)}\right)^{\frac{1}{2}}\quad\text{and}\quad K+1\leq T.\qquad(66)$$

This implies

$$\Pr\left(2\gamma(1-\beta)\left\|\sum_{t=0}^K\sum_{i=1}^n\langle\zeta_{5,i}^t,\theta^{t+1}\rangle\right\|\geq\frac{\Delta}{8}\right)\leq\frac{\alpha}{14(T+1)}.$$

Note that the worst dependency w.r.t $T$ is $\widetilde{\mathcal{O}}(1/T^{\frac{1}{4}})$.

**Final probability.** Therefore, the probability event

$$\Omega:=E_K\cap\overline{\Theta}^{K+1}\cap\left(\cap_{i=1}^n\overline{\Theta}_i^{K+1}\right)\cap\overline{N}^{K+1}\cap E_①\cap E_②\cap E_③\cap E_④\cap E_⑤\cap E_⑥\cap E_⑦\cap E_⑧,$$

where each $E_①$-$E_⑧$ denotes that each of 1-8-th terms is smaller than $\frac{\Delta}{8}$ implies that

$$① + ② + ③ + ④ + ⑤ + ⑥ + ⑦ + ⑧ \leq 8 \cdot \frac{\Phi^0}{8} = \Delta,$$

i.e. 7 in the induction assumption holds. Moreover, this also implies that

$$\Phi^{K+1} \leq \Phi^0 + \Delta \leq \Delta + \Delta = 2\Delta,$$

i.e. 6 in the induction assumption holds. The probability $\Pr(E_{K+1})$ can be lower bounded as follows

$$\begin{aligned}
\Pr(E_{K+1}) &\geq \Pr(\Omega) \\
&= \Pr\left( E_K \cap \overline{\Theta}^{K+1} \cap \left( \cap_{i=1}^n \overline{\Theta}_i^{K+1} \right) \cap \overline{N}^{K+1} \cap E_① \cap E_② \cap E_③ \cap E_④ \cap E_⑤ \cap E_⑥ \right. \\
&\qquad \left. \cap E_⑦ \cap E_⑧ \right) \\
&= 1 - \Pr\left( \overline{E}_K \cup \Theta^{K+1} \cup \left( \cup_{i=1}^n \Theta_i^{K+1} \right) \cup N^{K+1} \cup \overline{E}_① \cup \overline{E}_② \cup \overline{E}_③ \cup \overline{E}_④ \cup \overline{E}_⑤ \cup \overline{E}_⑥ \right. \\
&\qquad \left. \cup \overline{E}_⑦ \cup \overline{E}_⑧ \right) \\
&\geq 1 - \Pr(\overline{E}_K) - \Pr(\Theta^{K+1}) - \sum_{i=1}^n \Pr(\Theta_i^{K+1}) - \Pr(N^{K+1}) - \Pr(\overline{E}_①) - \Pr(\overline{E}_②) \\
&\qquad - \Pr(\overline{E}_③) - \Pr(\overline{E}_④) - \Pr(\overline{E}_⑤) - \Pr(\overline{E}_⑥) - \Pr(\overline{E}_⑦) - \Pr(\overline{E}_⑧) \\
&\geq 1 - \frac{\alpha(K+1)}{T+1} - \frac{\alpha}{6(T+1)} - \sum_{i=1}^n \frac{\alpha}{6n(T+1)} - \frac{\alpha}{6(T+1)} - 0 - 7 \cdot \frac{\alpha}{14(T+1)} \\
&= 1 - \frac{\alpha(K+2)}{T+1}.
\end{aligned}$$

This finalizes the transition step of induction. The result of the theorem follows by setting $K = T - 1$. Indeed, from (58) we obtain

$$\frac{\gamma}{2} \sum_{t=0}^K \|\nabla f(x^t)\|^2 \leq \Phi^0 - \Phi^{K+1} + \Delta \leq 2\Delta \Rightarrow \frac{1}{T} \sum_{t=0}^{T-1} \|\nabla f(x^t)\|^2 \leq \frac{4\Delta}{\gamma T}. \qquad (67)$$

**Final rate.** Now we have the following restrictions on the momentum parameter in terms of dependency on $T$ from each bound of terms 1-8 correspondingly

$$\beta \leq \widetilde{\mathcal{O}} \left( \underbrace{\left( \frac{L\Delta n}{T\sigma^2} \right)^{\frac{1}{2}}}_{\text{from term 1}}, \underbrace{\left( \frac{L\Delta\sqrt{n}\eta}{B\sigma\sqrt{T}} \right)^{\frac{1}{2}}}_{\text{from term 2}}, \underbrace{\left( \frac{L\Delta\sqrt{n}\eta}{\sigma\sqrt{T}(\sqrt{4L\Delta} + (B-\tau) + 2b)} \right)^{\frac{1}{3}}}_{\text{from term 3}}, \right.$$

$$\underbrace{\left( \frac{L\Delta\eta\sqrt{n}}{\sigma\sqrt{T}(\sqrt{64L\Delta} + 3(B-\tau+b))} \right)^{\frac{1}{4}}}_{\text{from term 4}}, \underbrace{\left( \frac{L\Delta\eta^2\sqrt{n}}{\sigma\sqrt{T}(\sqrt{4L\Delta} + {}^3\!/\!2(B-\tau) + 2b)} \right)^{\frac{1}{3}}}_{\text{from term 5}},$$

$$\underbrace{\left( \frac{L\Delta\eta^2\sqrt{n}}{\sigma\sqrt{T}(\sqrt{64L\Delta} + 3(B-\tau+b))} \right)^{\frac{1-p}{2(2-p)}}}_{\text{from term 7}}, \underbrace{\left( \frac{L\Delta\sqrt{n}}{\sigma\sqrt{T}(\sqrt{4L\Delta} + {}^3\!/\!2(B-\tau) + 2b)} \right)}_{\text{from term 6}},$$

$$\left. \underbrace{\left( \frac{L\Delta\sqrt{n}}{\sigma\sqrt{T}(\sqrt{64L\Delta} + 3(B-\tau+b))} \right)^{\frac{1}{2}}}_{\text{from term 8}} \right).$$

Multiplying the above by $\frac{1}{4L}$ gives restrictions on $\gamma$. The worst dependency on $T$ is given by ⑥ term and translates to the rate of the form

$$\widetilde{\mathcal{O}}\left( \frac{L\Delta}{T} \frac{\sigma\sqrt{T}(\sqrt{L\Delta} + B + b)}{L\Delta\sqrt{n}} \right) = \widetilde{\mathcal{O}}\left( \frac{\sigma(\sqrt{L\Delta} + B + b)}{\sqrt{T}n} \right).$$

Therefore, with probability $1 - \alpha$ Clip21-SGDM converges as

$$\frac{1}{T}\sum_{t=0}^{T-1} \|\nabla f(x^t)\|^2 \le \widetilde{\mathcal{O}}\left(\frac{L\Delta}{T}\frac{\sigma\sqrt{T}(\sqrt{L\Delta}+B+b)}{L\Delta\sqrt{n}}\right) = \widetilde{\mathcal{O}}\left(\frac{\sigma(\sqrt{L\Delta}+B+\sigma)}{\sqrt{Tn}}\right), \qquad (68)$$

where $\widetilde{\mathcal{O}}$ hides constant and logarithmic factors, and higher order terms decreasing with $T$.

CASE $\mathcal{I}_{K+1} = 0$.  This case is even easier. The only change will be with the term next to $R^t$. We will get

$$1 - \frac{48L^2}{\eta^2}\gamma^2 - \frac{12L^2}{\beta^2}\gamma^2 \ge \frac{2}{3} - \frac{48L^2}{\eta}\gamma^2 \ge 0$$

instead of

$$1 - \frac{16\beta^2 L^2}{\eta^2}\gamma^2 - \frac{48L^2}{\eta^2}\gamma^2 - \frac{12L^2}{\beta^2}\gamma^2 \ge 0$$

as in the previous case. This difference comes from Lemma 18 because $\widetilde{V}^{K+1} = 0$. The rest is a repetition of the previous derivations. $\qquad\square$

**Remark 2.** With $v_i^{-1} = g_i^{-1} = 0$ we have

$$\Phi^0 = F^0 + \frac{\gamma}{\eta}\frac{1}{n}\sum_{i=1}^n \|\operatorname{clip}_\tau(\beta\nabla f_i(x^0,\xi_i^0)) - \beta\nabla f_i(x^0,\xi_i^0)\|^2 + \frac{4\gamma\beta}{\eta^2}\frac{1}{n}(1-\beta)^2\sum_{i=1}^n \|\nabla f_i(x^0,\xi_i^0)\|^2$$

$$+ \frac{\gamma}{\beta}(1-\beta)^2 \left\|\nabla f(x^0,\xi^0)\right\|^2$$

$$\le F^0 + \frac{\gamma}{\eta}\frac{1}{n}\sum_{i=1}^n \max\left\{(\|\nabla f(x^0,\xi_i^0)\| - \tau)^2, 0\right\} + \frac{16L\gamma^2}{\eta^2}\frac{1}{n}(1-\beta)^2\sum_{i=1}^n \|\nabla f_i(x^0,\xi_i^0)\|^2$$

$$+ \frac{1}{4L}(1-\beta)^2 \left\|\nabla f(x^0,\xi^0)\right\|^2.$$

We have the stepsize restriction

$$\frac{2}{3} - \frac{64L^4\gamma^2}{\eta^2} - \frac{48L^2\gamma^2}{\eta^2} \ge 0. \qquad (69)$$

For inequality of the form $a\gamma^2 + b\gamma \le 1$ the stepsize restriction of the form $\gamma \le \frac{1}{\sqrt{a}+b}$ is tight up to a constant factor 2, i.e. $\frac{2}{\sqrt{a}+b}$ does not satisfy the inequality (see Lemma 5 in (Richtárik et al., 2021)). Using this lemma in our case we get that the stepsize satisfying Equation (69) should also satisfy

$$L^2\gamma^2 \le 2 \cdot \frac{\eta}{72/\eta + 4\sqrt{6}}.$$

This implies that $L^2\gamma^2 \le \frac{\eta}{4\sqrt{6}}$ and $L^2\gamma^2 \le \frac{\eta^2}{72}$. Consequently, it also satisfies $\frac{\gamma}{\eta} \le \frac{1}{6L\sqrt{2}}$ (from the last inequality). Therefore, we have

$$\Phi^0 \le F^0 + \frac{1}{6L\sqrt{2}}\frac{1}{n}\sum_{i=1}^n \max\left\{(\|\nabla f_i(x^0,\xi_i^0)\| - \tau)^2, 0\right\} + \frac{2}{9L}\frac{1}{n}\sum_{i=1}^n \|\nabla f_i(x^0,\xi_i^0)\|^2$$

$$+ \frac{1}{4L}(1-\beta)^2 \left\|\nabla f(x^0,\xi^0)\right\|^2$$

$$\le F^0 + \frac{1}{6L\sqrt{2}}\frac{1}{n}\sum_{i=1}^n \max\left\{\|\nabla f_i(x^0,\xi_i^0)\|^2, 0\right\} + \frac{2}{9L}\frac{1}{n}\sum_{i=1}^n \|\nabla f_i(x^0,\xi_i^0)\|^2$$

$$+ \frac{1}{4L}(1-\beta)^2 \left\|\nabla f(x^0,\xi^0)\right\|^2$$

which is independent of $\tau$, and can be use as a bound for $\Delta$. Terms containing $\|\nabla f_i(x^0,\xi_i^0)\|^2$ can be bounded by $B^2$ and $\|\nabla f_i(x^0)\|^2$ with high probability, i.e. $\Delta$ is again independent of $\tau$.

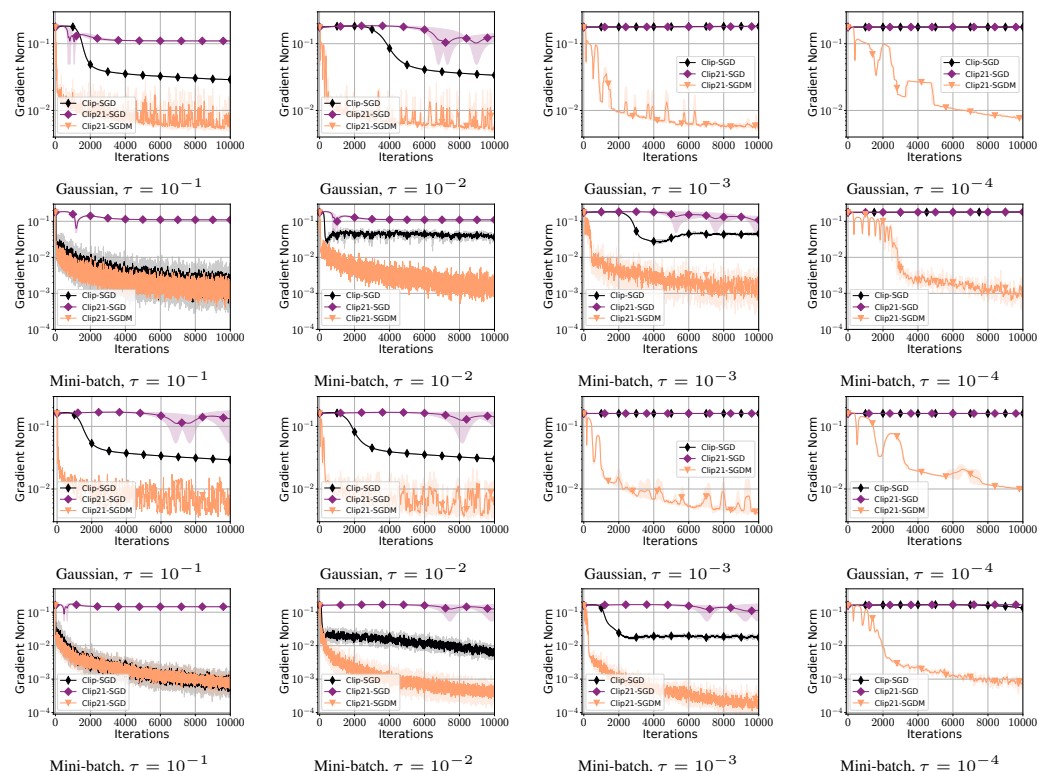

Figure 7: Comparison of Clip-SGD, Clip21-SGD, and Clip21-SGDM on logistic regression with non-convex regularization for various the clipping radii $\tau$ with mini-batch and Gaussian-added stochastic gradients on Duke (**two first rows**) and Leukemia (**two last rows**).

# H EXPERIMENTS DETAILS AND MORE

## H.1 EXPERIMENTS WITH LOGISTIC REGRESSION

### H.1.1 STOCHASTIC SETTING VARYING CLIPPING RADIUS

We conduct experiments on non-convex logistic regression with regularization parameter $\lambda = 10^{-3}$ for $10^4$ iterations. We use Duke and Leukemia datasets from LibSVM library and split the dataset into $n = 4$ equal parts. We normalize the row of the feature matrix to demonstrate the differences between algorithms. To simulate the stochastic gradients we either add centered Gaussian noise with variance $\sigma = 0.05$ for the Duke dataset and $\sigma = 0.1$ for the Leukemia dataset, or mini-batch gradients with batch-size of $\frac{1}{3}$ of the whole local dataset for Duke dataset and $\frac{1}{4}$ of the whole local dataset for Leukemia dataset. For Clip21-SGD and Clip-SGD algorithms, we tune the stepsize in $\{2^{-5}, \dots, 2^5\}$ and choose the one that gives the lowest final gradient norm in average across 3 random seeds. For Clip21-SGDM, we tune both the stepsize in $\{2^{-5}, \dots, 2^5\}$ and the momentum parameter in $\{0.1, 0.5, 0.9\}$ and choose the best pair of parameters similarly as before. For completeness, we report the convergence curves in Figure 7. We observe that Clip21-SGDM is more robust to the choice of the clipping radius $\tau$ while Clip-SGD converges well only for large enough $\tau$. Besides, Clip21-SGD does not converge in all cases which is also highlighted by our theory in Theorem 1.

### H.1.2 STOCHASTIC SETTING WITH ADDITIVE DP NOISE

We describe the setting in more detail for completeness. First, note that we use the same set of problem parameters as in Appendix H.1.1 such as $n, \lambda, \sigma$, and batch-size. Next, we fix a ratio between Gaussian DP noise variance $\sigma_\omega$ and the clipping parameter $\tau$ from $\{0.1, 1.0, 10.0\}$. For a given ratio, we tune Clip21-SGD and Clip-SGD algorithms across all possible pairs of the step-

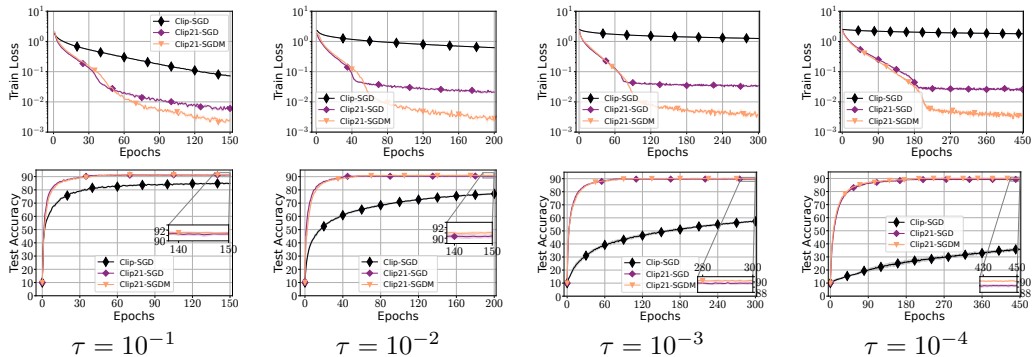

Figure 8: Comparison of Clip-SGD, Clip21-SGD, and Clip21-SGDM on training VGG16 model on CIFAR10 dataset where the clipping is applied globally.

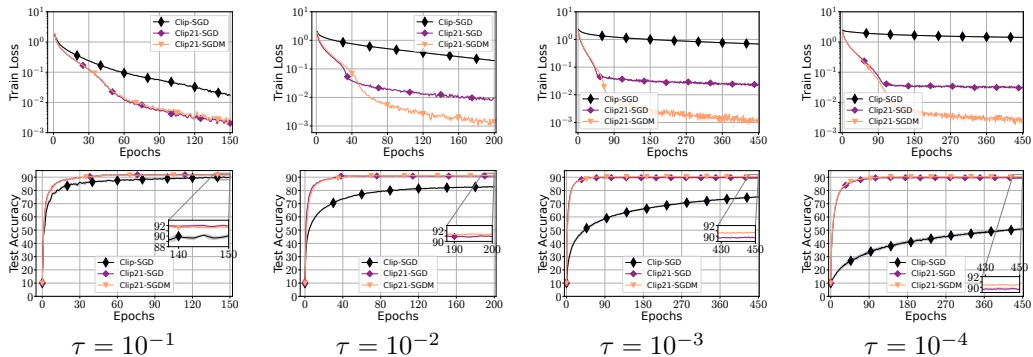

Figure 9: Comparison of Clip-SGD, Clip21-SGD, and Clip21-SGDM on training VGG16 model on CIFAR10 dataset the clipping is applied layer-wise.

size and the clipping radius $\tau$ where the step-size $\gamma$ is taken from $\{2^{-10}, \ldots, 2^0\}$ and $\tau$ — from $\{10^{-4}, \ldots, 10^0\}$, and choose the pair $(\gamma, \tau)$ that gives the smallest final gradient norm averaged over 3 runs. For Clip21-SGDM we perform the same grid search with an additional tuning of the momentum parameter $\beta \in \{0.1, 0.5, 0.9\}$. We report the last final gradient norm reached by each algorithm averaged over 3 runs.

## H.2 EXPERIMENTS WITH NEURAL NETWORKS

### H.2.1 VARYING CLIPPING RADIUS $\tau$

Now we switch to the training of Resnet20 and VGG16 models on CIFAR10 dataset. For all algorithms, we do not use any techniques such as learning rate schedule, warm-up, or weight decay. However, we do tuning of the learning rate for Clip-SGD and Clip21-SGD from $\{10^{-3}, 10^{-2}, 10^{-1}, 10^0\}$ and choose the one that gives the highest test accuracy. For Clip21-SGDM we tune both the learning rate from $\{10^{-3}, 10^{-2}, 10^{-1}, 10^0\}$ and the momentum parameter from $\{0.1, 0.5, 0.9\}$ and choose the pair that reaches the highest test accuracy. The batch size for all algorithms is set to 32. We compare the performance of algorithms in two cases: when the clipping is applied globally on the whole model and layer-wise.

We observe in Figures 8 to 11 that the performance of Clip-SGD gets worsen once the clipping radius is small enough. For Clip21-SGDM is more robust to the choice of $\tau$ and can achieve smaller train loss and test accuracy even when $\tau$ is small.

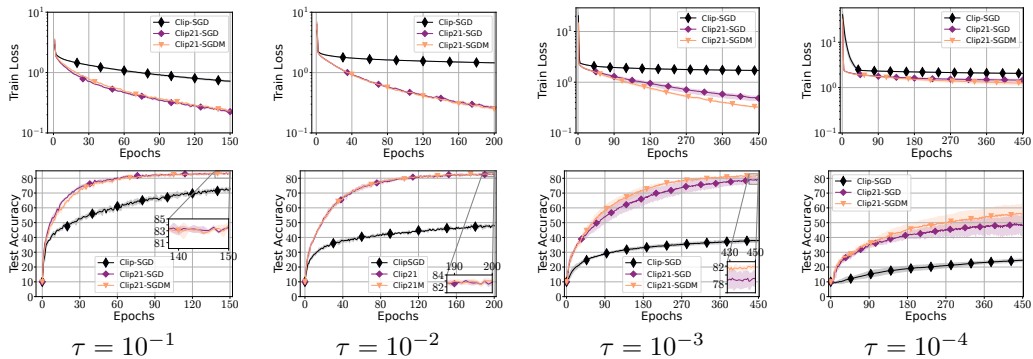

Figure 10: Comparison of Clip-SGD, Clip21-SGD, and Clip21-SGDM on training Resnet20 model on CIFAR10 dataset where the clipping is applied globally.

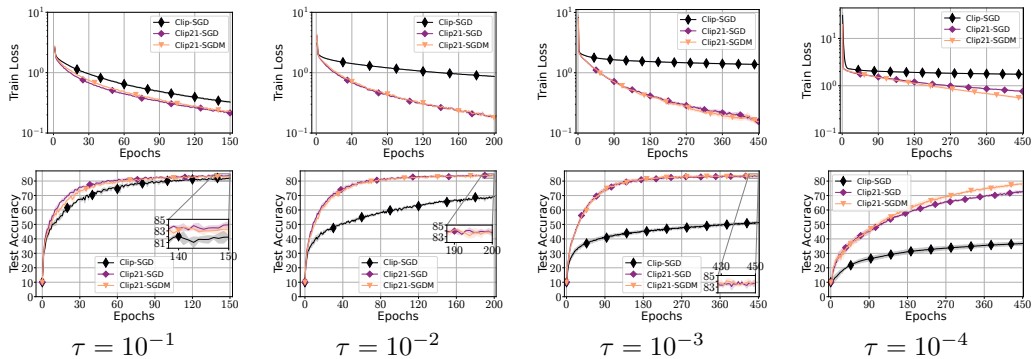

Figure 11: Comparison of Clip-SGD, Clip21-SGD, and Clip21-SGDM on training Resnet20 model on CIFAR10 dataset where the clipping is applied layer-wise.

### H.2.2 ADDING ADDITIVE DP NOISE

We consider the same experiment section described in Appendix H.1.2 but now in the training of MLP and CNN models on MNIST dataset.

We use MLP model with 1 hidden layer of size 256 and Tanh activation function. CNN model has 2 convolution layers with 16 convolutions each and kernel size 5 with one max-pooling layer and Tanh activation function. We perform a grid search over the learning rate from $\{10^{-3}, \ldots, 10^{0}\}$ and the clipping radius from $\{10^{-4}, \ldots, 10^{-1}\}$. The aforementioned tuning is performed for each value of the noise-clipping ratio from $\{0.1, 0.3, 1.0, 3.0, 10.0\}$. he momentum parameter is tuned over $\{0.5, 0.1, 0.01\}$. We highlight that we do not use the techniques such as a learning rate scheduler although it might improve the performance of algorithms. The batch size for all algorithms is set to 64.

In Figures 12 to 15 we demonstrate that Clip-SGD and Clip-SGDM always outperform Clip21-SGD. However, there is no clear separation between Clip21-SGDM and Clip-SGD: in some cases, the latter has better performance, and in some cases — the former.

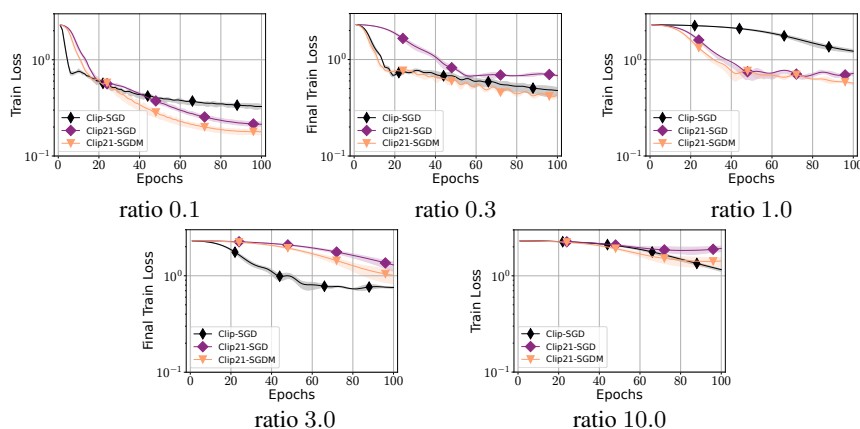

Figure 12: Comparison of Clip-SGD, Clip21-SGD, and Clip21-SGDM on training CNN model on MNIST dataset varying the noise-clipping ratio.

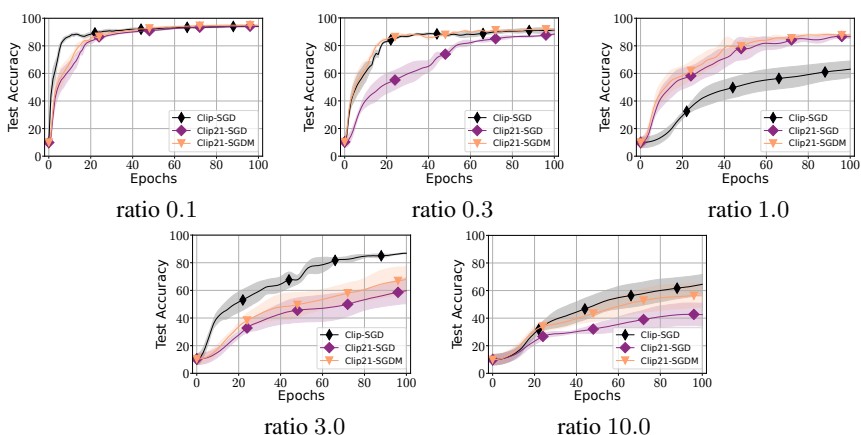

Figure 13: Comparison of Clip-SGD, Clip21-SGD, and Clip21-SGDM on training CNN model on MNIST dataset varying the noise-clipping ratio.

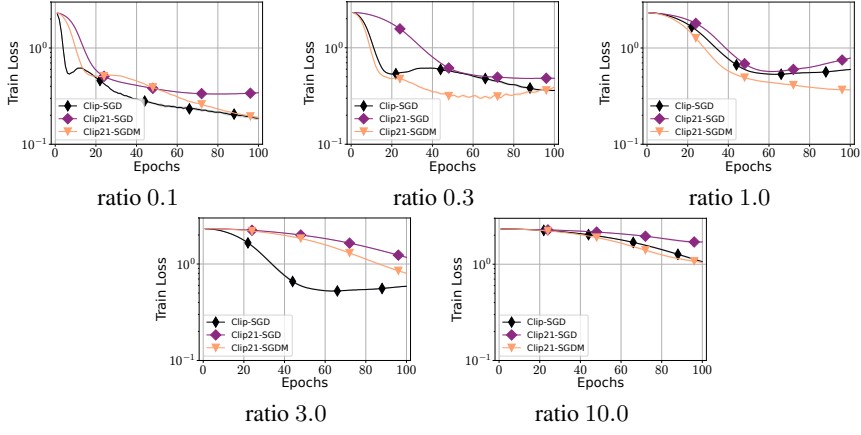

Figure 14: Comparison of Clip-SGD, Clip21-SGD, and Clip21-SGDM on training MLP model on MNIST dataset varying the noise-clipping ratio.

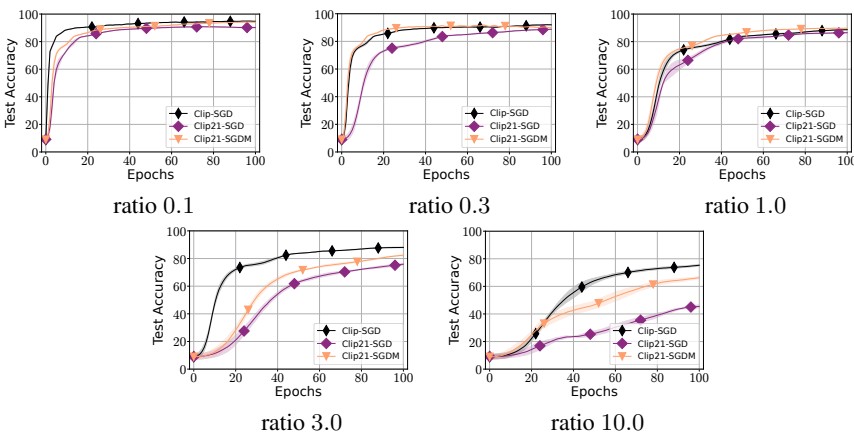

Figure 15: Comparison of Clip-SGD, Clip21-SGD, and Clip21-SGDM on training MLP model on MNIST dataset varying the noise-clipping ratio.

