# OpenReview forum: "Momentum and Error Feedback for Clipping with Fast Rates and Differential Privacy"
_ICLR.cc/2025/Conference — Submitted to ICLR 2025_

### Official Review · Reviewer_LwoD · 2024-10-27

**Soundness:** 2
**Presentation:** 3
**Contribution:** 2
**Rating:** 6
**Confidence:** 4

**Summary:**

This paper basically proposes the heavy-ball momentum version of clipped SGD with error feedback (Richtárik et al., 2021); the authors name this algorithm Clip21-SGDM. A version for DP training is also provided (by suitably adding noise to the updates). The paper also shows that standard clipped SGD + error feedback *without* momentum can fail to converge in some cases. Convergence results of $\mathcal{O}(1/T)$ and $\widetilde{\mathcal{O}}(1/\sqrt{n T})$ are derived for smooth non-convex functions in the deterministic and stochastic cases for Clip21-SGDM -- the good thing about these results is that they don't require any kind of bounded gradient assumption (unlike prior papers). Empirical results verify the theory showing that Clip21-SGDM converges better over a wide range of clipping radii in the non-DP case.

**Strengths:**

Unlike prior papers, the convergence results of this paper don't require any kind of bounded gradient assumption. (However, I'm not up to speed on the newer convergence bounds for clipping-based methods.) Also, I'm surprised by the fact that convergence can be guaranteed for even small clipping radii $\tau$ (looking at Theorems 2 and 3).

The theoretical analysis seems solid and the paper is written pretty well.

**Weaknesses:**

**Major:**

**1.** The $1/\sqrt{n}$ term in the bound of Corollary 1 does not seem optimal. The optimal dependence should be no worse than $1/n$; see e.g., [1]. (Some other works such as [2, 3] obtain even better bounds.) Can the authors explain the reason for this apparent sub-optimality in the dependence on $n$ or if I am missing something here?

**2.** The empirical results on CNN and MLP with DP in Figure 6 -- especially in the high-noise regime (high privacy regime) -- are not very good. The simplest method of Clip-SGD performs the best here. Moreover, results with MNIST are not very insightful; I'd have preferred to see results with CIFAR-10 (similar to Figures 3 and 4). For me, the major application of clipping is in private training; so if the results for private training aren't that good, I'm not sure how valuable the proposed method is to practitioners. Further, the results haven't been shown in terms of $(\varepsilon, \delta)$ (DP parameters) which makes it hard to compare against existing work.

**Minor:**

**1.** In Theorem 1, the authors did not show theoretically that Clip21-SGDM converges for the problem where Clip21-SGD cannot converge.

**2.** In Figure 3, the accuracy values for CIFAR-10 trained on ResNet-20 are much lower than what can be achieved (> 90% at least); see for instance, https://paperswithcode.com/sota/image-classification-on-cifar-10. Somewhere close to 90% should be achievable especially if the clipping radius is large enough. This makes me wonder if the results are with suboptimal hyper-parameters or if training wasn't done for long enough.

-------

[1]: Wang, Di, Minwei Ye, and Jinhui Xu. "Differentially private empirical risk minimization revisited: Faster and more general." Advances in Neural Information Processing Systems 30 (2017).

[2]: Arora, Raman, et al. "Faster rates of convergence to stationary points in differentially private optimization." International Conference on Machine Learning. PMLR, 2023.

[3]: Tran, Hoang, and Ashok Cutkosky. "Momentum aggregation for private non-convex ERM." Advances in Neural Information Processing Systems 35 (2022): 10996-11008.

**Questions:**

In addition to the questions in Weaknesses, please answer the following questions:

**1.** I'm surprised by the fact that convergence can be guaranteed for even small clipping radii $\tau$ (looking at Theorems 2 and 3). Intuitively, I don't understand how this can be true for $\tau \to 0$; the clipping bias should be extremely high in this case. Can the authors explain how convergence is being ensured for small $\tau$?

**2.** In Theorem 3, is $\Phi^0$ the same potential function in the proof sketch of Theorem 2?

**3.** In Theorem 3, I'm assuming $\tilde{O}(.)$ hides log factors in $(1/\alpha)$? I'd suggest adding it to the theorem statement to make the dependence on $\alpha$ clear.

---

### Official Review · Reviewer_xg2D · 2024-11-03

**Soundness:** 2
**Presentation:** 3
**Contribution:** 2
**Rating:** 5
**Confidence:** 2

**Summary:**

This paper addresses the challenge of achieving both strong DP and optimization guarantees in Federated Learning. The authors introduce Clip21-SGDM, a new method that combines clipping, heavy-ball momentum, and error feedback to provide optimal convergence rates. Experiments on non-convex logistic regression and neural network training demonstrate that Clip21-SGDM outperforms baseline methods in optimization performance for a given DP budget.

**Strengths:**

1. The paper is well written and provides sufficient analysis on deterministic, stochastic cases with and without DP-noise.
2. The experiment is carefully designed and details are provided.

**Weaknesses:**

1. This paper improves an unpublished work, Clip21-SGD, thus personally, it is unclear how big the contributions of this paper are.
2. The paper did not compare to other SOTA methods. For example, how is [1] compare to this work for DP case? The paper does not compare with other methods either in the main text, or in the supplementary.

[1] DIFF2: Differential Private Optimization via Gradient Differences for Nonconvex Distributed Learning. ICLR 2023

**Questions:**

How is the privacy budget (epsilon) is set in the experiment with DP noise? I don't see this in Fig. 6. How is your explanation relate to it?

---

### Official Review · Reviewer_W75D · 2024-11-04

**Soundness:** 4
**Presentation:** 3
**Contribution:** 3
**Rating:** 6
**Confidence:** 3

**Summary:**

The paper considers the problem of using (DP) federated learning to minimize a non-convex smooth loss using heterogenous clients. Using the standard Clip-SGD algorithm is not guaranteed to converge to a local minimum when one exists, even when clients' gradients are non-stochastic. The Clip21-SGD algorithm instead clips the difference between a current estimate of the client's gradient and its true gradient, and adds this difference to its current estimate to get a new estimate. When clients have non-stochastic gradients Clip21-SGD converges, but the authors give an example which shows that with stochastic gradients Clip21-SGD is not guaranteed to converge. The authors propose a modification of Clip21-SGD called Clip21-SGDM, where each client reports an exponentially weighted average of its gradients so far (i.e., the update it would use if using SGDM), and Clip21-SGDM clips the difference between this average and its current estimate of the average.  The authors show that unlike Clip21-SGD, Clip21-SGDM converges (in terms of the squared gradient norm)  at the optimal rate of 1/T when gradients are non-stochastic, and at rate $1/T + s/\sqrt{T n}$ when client gradients are s-subGaussian. Finally, the authors show that the algorithm achieves the optimal bound of $\sqrt{d} / \sqrt{n} \epsilon$ under $(\epsilon, \delta)$ local-DP when Gaussian noise is added to the clipped values. Empirically, the authors compare Clip-SGD, Clip21-SGD, and Clip21-SGDM. On logistic regression with a non-convex regularizer, Clip21-SGDM achieves smaller final gradient norm with and without DP. On image recognition tasks of MNIST and CIFAR without DP, it achieves better performance than Clip-SGD and comparable or slightly better performance than Clip21-SGD. With DP, it outperforms Clip21-SGD but is incomparable to Clip-SGD on MNIST tasks.

**Strengths:**

The main strength of the paper is the theoretical contribution. The authors identify a salient issue with the past work, which is that algorithms using clipping do not converge in the stochastic setting. The counterexample the authors construct for Clip21-SGD demonstrates the need for new algorithms to overcome this issue. The modification the authors propose is simple and easily to implement yet leads to optimal rates in a variety of settings, but proving these rates requires a very intricate analysis. Overall the paper gives a clean resolution to a challenging problem of federated non-convex optimization with clipping, and both in terms of results and technical work would be a good contribution to a conference like ICLR.

**Weaknesses:**

* The results on DP deep learning are somewhat limited and not very promising - even though MNIST is one of the most well-behaved deep learning tasks, Clip-SGD sometimes outperforms Clip21-SGDM in the authors results in Figure 6. Since Clip21-SGDM seems to inherently rely on smoothness (as otherwise the differences between gradients that are being clipped can be quite large), on deep learning tasks which are not guaranteed to be non-smooth this is perhaps expected, but it may limit the practicality of the proposed methods. The paper seems to be primarily a theory paper, however, so this did not factor into my final rating significantly (i.e., even if the empirical results only included logistic regression, I think the paper makes a strong contribution). But, I'm also not confident in the presented numbers and think the authors should revisit results in Figure 6 - see Questions below.
* It would be nice to have some discussion, even if brief, of the example in Theorem 1 / intuition for why Clip21-SGD fails in this example, and why Clip21-SGDM would work for this example. Right now the body of the paper doesn't do much to help the reader understand why Clip21-SGDM is the right algorithm for the problem. The Proof Sketch of Theorem 2 is included, but discusses the technical details of the proof which is not particularly helpful for anyone not familiar with the Lyapunov function-based proofs of past work.

**Questions:**

In the leftmost subfigure in Figure 6, do the authors know why Clip-SGD's accuracy is heavily non-monotonic in the noise level? A drop of ~20% accuracy on MNIST at noise multiplier 1 that is almost immediately recovered from at a significantly higher noise multiplier of ~3 seems strange; while I could understand if the accuracy was slightly non-monotonic because of e.g. variance in the experiments, or phenomenon like overfitting, I don't think the resulting change in accuracy would be this large. Based on the other 4 data points, I'd expect this point for Clip-SGD to be comparable to or better than the other methods, which would change the conclusions of the authors significantly.

The authors include error bars which suggest this is reproducible over multiple trials, so it would be nice if the authors could explain this anomaly in the results if they have a convincing explanation, or if not perhaps quickly double-check the code for the experiments (e.g. was a noise multiplier of 10 used instead of 1 here on accident?)

---

### Official Review · Reviewer_n6ya · 2024-11-04

**Soundness:** 4
**Presentation:** 3
**Contribution:** 2
**Rating:** 6
**Confidence:** 3

**Summary:**

The paper provides a momentum and error feedback based clipping SGD mechanism which can satisfy DP by adding appropriate noise.
The authors provide formal proof of convergence of the algorithm in three cases: (i) deterministic case (using exact gradient at each step), (ii) stochastic case (each worker has access to σ-sub-Gaussian unbiased estimator), and (iii) the case where DP noise is added to the clipped term in the stochastic case. The theoretical analysis provides three main results:
$O(1/T)$ convergence rate in the deterministic case
$O(1/\sqrt{nT})$ high-probability convergence rate in the stochastic case with sub-Gaussian noise
Optimal local DP-error bounds when privacy noise is added.
The proof of the algorithm doesn't rely on any assumptions regarding the boundedness of gradients or data heterogeneity and only requires Assumption 1 (L-smoothness and lower boundedness) and Assumption 2 (sub-Gaussian property of the stochastic gradients). They provide experiments on nonconvex logistic regression, CIFAR10 with ResNet20 and VGG16, and MNIST dataset with MLP and CNN.

**Strengths:**

The paper combines clipping, momentum, and error feedback techniques. This combination is a connection between non-DP error feedback momentum-based SGD methods and private error feedback SGD without momentum. The convergence analysis does not rely on common assumptions about bounded gradients and data heterogeneity that were required in previous methods.
The theoretical results of the paper are: (i) $O(1/T)$ convergence rate in the deterministic case, (ii) $O(1/\sqrt{nT})$ high-probability convergence rate in the stochastic case, and (iii) optimal privacy-utility trade-off for local differential privacy.
Theoretically, achieving optimal convergence rate (for first-order methods in non-convex objective functions) without requiring bounded gradient assumptions, and maintaining differential privacy guarantees is interesting. Practically, the algorithm is robust and achieves optimality even with small values of the clipping parameter in the nonconvex optimization case, which makes it suitable for the nonconvex setup compared to other clipping methods.

**Weaknesses:**

The main weakness lies in the experimental section of the paper:
The paper only compares against clipping-based methods (Clip-SGD and Clip21-SGD).
Missing comparisons with:
EF21-SGDM [1] in non-DP settings to isolate the impact of clipping
Other private FL methods like DP-SCAFFOLD or PORTER
Methods using secure aggregation or shuffling for privacy

Additional experimental limitations include:
No experiments to show the effect of data heterogeneity
No discussion of the impact of client sampling/partial participation
The experiments should be based on the $\epsilon$ value for differential privacy
No discussion to show the effect of the number of workers on convergence

To improve the paper, the authors could:
Include comparisons with non-clipping methods and state-of-the-art private distributed learning algorithms
Add experiments with real heterogeneous federated datasets
Provide more detailed privacy analysis with privacy parameter $\epsilon$
Add analysis of computational and communication costs
Explore the impact of partial participation and client sampling
As noted in the paper, examine the effect of gradient clipping with different clipping factors for FL under Byzantine attacks

[1] Fatkhullin, Ilyas, Alexander Tyurin, and Peter Richtárik. "Momentum provably improves error feedback!." Advances in Neural Information Processing Systems 36 (2024).

**Questions:**

Why there is no comparison with the other momentum based methods for instance EF21-SGDM with their experiment in the reference?
Why there is no clip21-sgdm analysis for right side of figure 1 for different clients?
Why were these specific baselines (Clip-SGD and Clip21-SGD) chosen? Could you provide comparisons with EF21-SGDM in non-DP settings?
Could you provide empirical evidence showing the effect of the number of workers (n) on convergence rate?
How does the privacy budget ε affect the convergence rate in practice? Could you provide experiments with different ε values?
What is the computational overhead compared to simpler methods like Clip-SGD?
Could you demonstrate the method's performance on naturally heterogeneous federated datasets?
How does the algorithm perform under partial client participation or Byzantine attacks?

---

### Public Comment · ~Rustem_Islamov1 · 2025-02-21
**Updated version of the paper**

We would like to emphasize that the updated version of the paper is available on arXiv: https://arxiv.org/abs/2502.11682

The key takeaways are:
- the convergence rate in the deterministic and stochastic cases remains unchanged; we added the explicit dependency on the clipping parameter $\tau$ in these cases.
- we added the second momentum which allows to average DP-noise which is needed to reduce its impact.
- we provide the convergence rate in the stochastic with and without DP noise together; the rate in the stochastic case can be obtained by setting the DP-noise variance $\sigma_{\omega}=0$.
- consequently, we provided corrected DP guarantees. The modified algorithm achieves near-optimal privacy-utility tradeoff

---

### Meta-Review · Area_Chair_qGzV · 2024-12-20

**Metareview:**

a) Summary

Prior work identified that clipping stochastic gradients introduces bias, which hampers convergence. The authors demonstrate that error feedback, a commonly used approach to address this bias, fails to resolve convergence issues in certain cases. They propose a novel combination of error feedback (EF21) and momentum averaging, showing that this approach effectively overcomes convergence challenges while achieving optimal convergence rates.

b) Strengths
- A new algorithm, Clip21-SGDM, which combines heavy-ball momentum, error feedback (EF21), and gradient clipping to overcome demonstrated convergence limitations. I especially liked that the approach is well motivated, with clear demonstration that previous approaches fail.
- The paper provides rigorous optimal convergence guarantees for the proposed method in non-convex optimization settings, addressing deterministic, stochastic, and DP noise scenarios.

c) Weaknesses
- The privacy analysis in this paper is **missing**. While the authors define central item-level DP, their approach seems to be motivated by local / user-level DP.  The $v_^t$ term uses heavy ball momentum and so has terms involving multiple datapoints. This is unlike SGD. So standard subsampling and sensitivity analysis does not hold. The authors need to carefully perform privacy accounting for their proposed algorithm.
- Lack of intuition: I share Reviewer LwoD's concern. I'm surprised by the fact that convergence can be guaranteed for arbitarily small clipping radii $\tau$. (Theorems 2 and 3). What is preventing us from setting $\tau = 0$?

d) Reason for *rejection*

While the paper is very intruiging and has promising results, the authors need to address the privacy concerns. Also improved intuitions for their theory would be helpful.

**Additional Comments On Reviewer Discussion:**

There was no discussion since the authors did not respond.

---

### Decision · Program_Chairs · 2025-01-22

Reject